# Lieb-Schultz-Mattis anomalies and web of dualities induced by gauging in quantum spin chains

Ömer Mert Aksoy[1*], Christopher Mudry[1,2], Akira Furusaki[3,4] and Apoorv Tiwari[5†]

**1** CMT Group, Paul Scherrer Institute, CH-5232 Villigen PSI, Switzerland
**2** Institut de Physique, EPF Lausanne, CH-1015 Lausanne, Switzerland
**3** RIKEN Center for Emergent Matter Science, Wako, Saitama, 351-0198, Japan
**4** Condensed Matter Theory Laboratory, RIKEN, Wako, Saitama, 351-0198, Japan
**5** Department of Physics, KTH Royal Institute of Technology, Stockholm, 106 91 Sweden

★ omermertaksoy@gmail.com , † t.apoorv@gmail.com

## Abstract

Lieb-Schultz-Mattis (LSM) theorems impose non-perturbative constraints on the zero-temperature phase diagrams of quantum lattice Hamiltonians (always assumed to be local in this paper). LSM theorems have recently been interpreted as the lattice counterparts to mixed 't Hooft anomalies in quantum field theories that arise from a combination of crystalline and global internal symmetry groups. Accordingly, LSM theorems have been reinterpreted as LSM anomalies. In this work, we provide a systematic diagnostic for LSM anomalies in one spatial dimension. We show that gauging subgroups of the global internal symmetry group of a quantum lattice model obeying an LSM anomaly delivers a dual quantum lattice Hamiltonian such that its internal and crystalline symmetries mix non-trivially through a group extension. This mixing of crystalline and internal symmetries after gauging is a direct consequence of the LSM anomaly, i.e., it can be used as a diagnostic of an LSM anomaly. We exemplify this procedure for a quantum spin-1/2 chain obeying an LSM anomaly resulting from combining a global internal $\mathbb{Z}_2 \times \mathbb{Z}_2$ symmetry with translation or reflection symmetry. We establish a triality of models by gauging a $\mathbb{Z}_2 \subset \mathbb{Z}_2 \times \mathbb{Z}_2$ symmetry in two ways, one of which amounts to performing a Kramers-Wannier duality, while the other implements a Jordan-Wigner duality. We discuss the mapping of the phase diagram of the quantum spin-1/2 $XYZ$ chains under such a triality. We show that the deconfined quantum critical transitions between Neel and dimer orders are mapped to either topological or conventional Landau-Ginzburg transitions. Finally, we extend our results to $\mathbb{Z}_n$ clock models with $\mathbb{Z}_n \times \mathbb{Z}_n$ global internal symmetry, and provide a reinterpretation of the dual internal symmetries in terms of $\mathbb{Z}_n$ charge and dipole symmetries.

# 1 Introduction

## 1.1 Motivation

The Lieb-Schultz-Mattis (LSM) Theorem [1] and its extensions [2–41] are no-go theorems that constrain the low-energy properties of lattice Hamiltonians with certain combinations of internal and crystalline symmetries. While in its original form the LSM Theorem applies to spin-1/2 chains with SO(3) spin-rotation and translation symmetries, many generalizations for general crystalline [11–13, 18, 19] and internal symmetries, for systems with bosonic and

fermionic [24, 35, 41] degrees of freedom, and for spatial dimensions greater than one [7–9, 15, 18, 28, 30, 34, 35, 37] have been proposed.

LSM Theorems rule out a ground state that is trivially gapped and symmetric, i.e., a ground state that is simultaneously gapped, non-degenerate on any closed space manifold, and symmetric under the relevant internal and crystalline symmetries. Conversely, LSM Theorems predict that ground states that are symmetric must support either gapless excitations or topological order.

Global symmetries play a pivotal role in organizing various aspects of quantum systems. In particular, operators and states organize into representations of the symmetry group, while the phase diagrams and dynamics are constrained by symmetries. Global symmetries of quantum systems can be anomalous. A quantum 't Hooft anomaly [42] arises when the partition function coupled to a symmetry background gauge field is not invariant under gauge transformations of this background. Instead, the partition function transforms by a U(1) phase factor that cannot be absorbed by the addition of local terms. While quantum anomalies were initially investigated in continuum quantum field theories with fermions and Lie group symmetries [43–45], in recent years there has been much progress in understanding anomalies in more general contexts involving bosonic systems, finite symmetries [46–48], and lattice quantum systems [39, 49–51]. The low-energy dynamics and phase diagrams of systems with 't Hooft anomalies are strongly constrained in a manner reminiscent of LSM Theorems. The anomaly matching condition [42] requires that any trivially gapped ground state necessarily breaks the full symmetry down to a subgroup that trivializes the anomaly.

This similarity between the constraints imposed at low energies by LSM Theorems for lattice Hamiltonians, on the one hand, and by 't Hooft anomalies on the other hand, suggests a close connection between LSM Theorems and 't Hooft anomalies [15, 17, 23, 28, 39, 52, 53]. More precisely, LSM Theorems can be connected to mixed 't-Hooft anomalies by showing that the long-wavelength continuum descriptions of lattice Hamiltonians, for which an LSM Theorem applies, support mixed 't-Hooft anomalies between symmetries that originate from internal and crystalline symmetries participating in the LSM theorem. This means that, while neither the internal nor the crystalline symmetry are individually anomalous, their combination is. This translates to the fact that, while there is no obstruction to a trivially gapped and symmetric ground state under either internal or crystalline symmetry, any such state cannot be gapped and symmetric under the full symmetry group that participates in the LSM Theorem. Equivalently, the full internal symmetry group cannot be gauged, while preserving the crystalline symmetries. However, there is no obstruction to gauging a non-anomalous subgroup of internal symmetries for which there is no LSM Theorem. Accordingly, LSM Theorems have been reinterpreted as LSM anomalies, a terminology that we will follow in this paper.

In recent years, there has been much progress towards classifying topological phases of matter with crystalline symmetries and understanding the corresponding quantum anomalies [54–62]. Such classifications have often relied on the intuition that in the long wavelength continuum description, some crystalline symmetries appear as internal symmetries. Despite this progress, the lattice understanding of anomalies [50, 51, 63], in particular those involving crystalline symmetries, is very much an evolving subject [39]. Challenges arise because it is often unclear how to probe crystalline symmetries through coupling to crystalline backgrounds [55, 59, 62, 64], as is routinely done with internal symmetries and gauge fields. What is even less clear is how to dynamically gauge a crystalline symmetry by summing over the crystalline backgrounds. These issues make pinpointing LSM anomalies on the lattice a subtle task.

In this work, we circumvent these obstacles in local quantum lattice models by gauging non-anomalous subgroups of their internal symmetries. This approach is always viable since the chosen internal symmetry is non-anomalous, and methods to gauge internal symmetries are well-known from lattice gauge theory.

Gauging global symmetries is a powerful way to manipulate the symmetry structure of a quantum system [65–68]. By starting with a system with a known symmetry structure, like a finite group with certain anomalies, and gauging non-anomalous sub-symmetries, one obtains dual (gauged) theories with novel symmetry structures [51,66–82]. Generalized gauging procedures have recently emerged as effective methods to study generalized symmetry structures in both continuous and lattice systems. For instance, in one spatial dimension, gauging a non-anomalous finite symmetry that participates in a mixed anomaly results in a dual (gauged) theory with a non-anomalous global symmetry that extends the residual symmetries left after gauging. In higher dimensions, this group extension becomes a higher group [66,69]. Interestingly, these gauging procedures have also been used to furnish non-invertible symmetry structures [83].

Another reason to study gauging of finite global symmetries is that such gaugings are realized as dualities in quantum systems. For example, the well-known Kramers-Wannier [84–90] and Jordan-Wigner [91] dualities are essentially gaugings of the $\mathbb{Z}_2$ internal and $\mathbb{Z}_2$ fermion-parity symmetry in one-dimensional lattice models [92–113]. Dualities can be used to provide profound non-perturbative insights into quantum systems and are therefore very valuable.

In this work, we study the gauging of subgroups of internal symmetry which participate in LSM anomalies. More precisely, we choose a subgroup such that neither the gauged subgroup nor the remaining symmetries have an LSM anomaly with the crystalline symmetries, while an LSM anomaly applies for the full internal symmetry group. We track how the crystalline and internal symmetries organize into the symmetry structure of the dual (gauged) theory. We find that, as a direct consequence of an LSM anomaly in the pre-gauged theory, there is necessarily a non-trivial mixing of internal and crystalline symmetries in the dual theory. More concretely, we exemplify this procedure on a local quantum spin-1/2 chain that has a global $\mathbb{Z}_2 \times \mathbb{Z}_2$ internal symmetry in addition to translation and reflection crystalline symmetries [26,31]. The local representatives of the internal symmetry operators satisfy a projective representation of $\mathbb{Z}_2 \times \mathbb{Z}_2$ which, in turn, implies an LSM anomaly involving either translation or reflection symmetry. We gauge a subgroup $\mathbb{Z}_2 \subset \mathbb{Z}_2 \times \mathbb{Z}_2$ of the global internal symmetry $\mathbb{Z}_2 \times \mathbb{Z}_2$ in two ways, which amounts to performing Kramers-Wannier (KW) or Jordan-Wigner (JW) dualities, respectively. We establish a triality of the original model and its duals under KW or JW dualities. After the KW duality, we find that the dual symmetry becomes non-Abelian, more precisely a semi-direct product of the internal and crystalline symmetries. After the JW transformation too, the LSM anomaly gets traded for a symmetry structure that involves a non-trivial fermionic group extension of the internal and crystalline symmetry groups.

Starting with the original LSM Theorem [1], many LSM Theorems have been probed and proven using background gauge fields (or equivalently twisted boundary conditions) of internal symmetries [1, 7, 34, 35, 37, 40]. Our work presents a novel method for probing LSM anomalies based on dynamical gauging of internal sub-symmetries which provides an indirect yet robust way to pin-point the existence of LSM anomalies. We, therefore, confirm that gauging non-anomalous subgroups of finite symmetries with LSM anomalies leads to a non-anomalous group extension in the dual theory, a fact known for finite internal symmetries with mixed anomalies [66].

A deconfined quantum critical point (DQCP) describes a continuous transition between phases with distinct symmetries. Such transitions are driven by deconfinement of point defects of symmetry breaking order parameters such that the defects of the order parameter of one phase bind a non-vanishing expectation value of the order parameter of the other phase and vice versa [114–117]. DQCPs arise naturally in models with symmetries that carry mixed anomalies, where the relationship between defects of the order parameters can be traced back to the mixed anomaly between two subgroups. For instance, the paradigmatic example of DQCP is the conjectured continuous transition between the Neel and valance-bond-solid (VBS)

orders of the Heisenberg antiferromagnet on the square lattice. The former order preserves the crystalline $C_4$ rotation symmetry and breaks the internal SO(3) symmetry, while the latter order breaks the $C_4$ symmetry and preservs the SO(3) symmetry. Indeed, there exists an LSM anomaly between these symmetries, which rules out a trivially gapped ground state that is symmetric under both SO(3) and $C_4$ [18, 21, 28]. Under gauging a non-anomalous subgroup, a DQCP is often mapped to conventional Landau-Ginzburg-type transitions, where symmetries preserved by one phase is a subgroup of the other [51, 118–120]. Motivated by the relation between mixed anomalies and DQCP, we study the phase diagram of the quantum spin-1/2 $XYZ$ chain under the KW and JW dualities. This model features deconfined phase transitions between Neel and dimer ordered phases [121]. We show that as the crystalline and internal symmetries are mixed after gauging, the DQCP between Neel and Dimer ordered phases are mapped to either (i) topological phase transitions between two phases with same symmetries or (ii) conventional Landau-Ginzburg-type symmetry breaking transitions. Therein, we demonstrate how dualities can be utilized to recast DQCPs and also understand the phase diagrams of spin-1/2 and interacting Majorana chains.

## 1.2 Summary of the main results

Common to most LSM Theorems are quantum dynamical degrees of freedom defined on the sites of a lattice with a quantum dynamics that is local and invariant under

- the symmetry group

$$G_{tot} = G_{spa} \times G_{int}, \tag{1.1a}$$

  built from the direct product of a space (crystalline) symmetry group $G_{spa}$ with a global internal symmetry group $G_{int}$

- such that $G_{int}$ cannot be gauged in its entirety, while preserving the symmetry under the full space subgroup $G_{spa}$.[1]

In this work, by way of explicit examples, we study the dualities induced by gauging a subgroup of the internal symmetry $G_{int}$ that does not participate in the LSM anomaly. The paradigmatic example that we shall follow is the case of quantum spin-1/2 degrees of freedom at every site of lattice $\Lambda$ (a chain of even cardinality $|\Lambda|$) with

$$G_{spa} = Z_{|\Lambda|}^t \rtimes \mathbb{Z}_2^r, \tag{1.1b}$$

the space symmetry generated by translation ($t$) and site-centered reflection ($r$), and

$$G_{int} = \mathbb{Z}_2^x \times \mathbb{Z}_2^y, \tag{1.1c}$$

the global internal symmetry generated by $\pi$ rotations along the $x$ and $y$ axes in internal spin-1/2 space, respectively. It is known that both translation and reflection symmetries participate in LSM anomalies with the global internal $\mathbb{Z}_2^x \times \mathbb{Z}_2^y$ symmetry. Our main results are as follows.

1. With the tools reviewed in Sec. 2, we show in Sec. 3 that, under both KW and JW dualities, the LSM anomaly that was present before gauging is no longer operative for the dual symmetries after gauging, instead the dual global symmetry group $G_{tot}^\vee$ is a group extension of the dual internal symmetry group $G_{int}^\vee$ by the dual crystalline symmetry group $G_{spa}^\vee$. For instance, the Abelian global symmetry group $\mathbb{Z}_2^r \times \mathbb{Z}_2^x \times \mathbb{Z}_2^y$ that is generated

---

[1]This is so whenever the internal symmetry group is represented globally by a group homomorphism, while it is represented locally by a non-trivial projective representation of the internal symmetry group. Indeed, whereas it is possible to construct a many-body state that is a gauge singlet, the local Hilbert space does not admit a state that is a gauge singlet.

by a site-centered reflection together with $\pi$ rotations along $x$ and $y$ axes in internal spin-1/2 space maps under KW duality to the non-Abelian group $D_8$ (dihedral group of order 8) defined in Eq. (3.18b).

2. As a concrete application, we study the zero-temperature phase diagram of the quantum spin-1/2 antiferromagnetic $XYZ$ chain with nearest- and next-nearest-neighbor couplings under the KW and JW dualities in Sec. 4. For this quantum $XYZ$ chain, the presence of LSM anomalies precludes a non-degenerate gapped ground state that is simultaneously symmetric under both the crystalline (translation or reflection) symmetry group (1.1b) and the global internal symmetry group (1.1c). As such, in the parameter space of interest, three gapped phases are realized that spontaneously break either the crystalline symmetry or the global internal symmetry. The boundaries separating in parameter space these gapped spontaneously symmetry broken (SSB) phases are continuous phase transitions that realize deconfined quantum criticality (DQC) [121]. We present a triality of phase diagrams with Figs. 5 and 6 such that the dual phase diagrams contain non-degenerate symmetric and gapped ground states due to the absence of any LSM anomaly after gauging. We find that the continuous DQC phase transitions prior to gauging the quantum $XYZ$ chain are to be reinterpreted as continuous phase transitions that are either of the conventional Landau-Ginzburg type or of the topological type after gauging.

3. To solidify the correspondence between LSM anomalies prior to gauging and mixing of internal and crystalline symmetries after gauging, we study in Sec. 5 the family labeled by $n = 2, 3, \cdots$ of $\mathbb{Z}_n$ clock models with $\mathbb{Z}_n \times \mathbb{Z}_n$ global internal symmetry. We show that, when a $\mathbb{Z}_n \subset \mathbb{Z}_n \times \mathbb{Z}_n$ subgroup is gauged, the dual of the remaining $\mathbb{Z}_n$ symmetry generator mixes with translation for any $n$, while a mixing occurs with reflection only when $n$ is even. This result is consistent with the following conjecture. The mixing induced by gauging betwen dual crystallline symmetries and dual global internal symmetries occurs if and only if the crystalline and global internal symmetries are subject to an LSM anomaly prior to gauging.

4. We unravel a connection between Hamiltonians with spatially modulated internal symmetries, such as a $\mathbb{Z}_n$-dipole symmetry, and Hamiltonians with global (spatially uniform) internal symmetries, such as $\mathbb{Z}_n \times \mathbb{Z}_n$ symmetry. We show that gauging a $\mathbb{Z}_n$-charge symmetry induces a duality between Hamiltonians with $\mathbb{Z}_n$-charge, $\mathbb{Z}_n$-dipole, and translation (or link-centered reflection) symmetries, and Hamiltonians with global uniform $\mathbb{Z}_n \times \mathbb{Z}_n$ symmetry, translation or (site-centered reflection) symmetry, and an LSM anomaly.

## 1.3 Comparison with the literature

Since 2016, LSM Theorems have been reinterpeted as mixed 't Hooft anomalies involving crystalline symmetries in the following loose sense. On the one hand, it has been argued that LSM Theorems can be understood as 't Hooft anomalies of emergent internal symmetries arising from crystalline symmetries in the low-energy continuum limit [17, 21, 25]. On the other hand, Refs. [15, 18, 23, 28] have made the salient observation that LSM Theorems can be applicable to the boundaries of crystalline topological phases in one higher dimension.

Most studies of mixed 't-Hooft anomalies have treated the cases of internal symmetries in relativistic quantum field theories for which the partial gauging leaves the space-time symmetries unaffected. In this context, the dualities induced by gauging a subgroup of a finite group are worked out in Refs. [66, 67]. For instance, the duality between the $\mathbb{Z}_2 \times \mathbb{Z}_2 \times \mathbb{Z}_2$ global internal symmetry with a mixed anomaly involving all three $\mathbb{Z}_2$ subgroups and the $D_8$ internal

symmetry has been established in Ref. [66]. Our results in Sec. 3 parallels this fact for LSM anomalies which involves crystalline symmetries. From this point of view, our results confirm the crystalline equivalence principle [55,59] that suggests a one-to-one correspondence between global internal and crystalline symmetries.

The connection between LSM anomalies with translation symmetry and mixed 't Hooft anomalies has been studied for lattice models [39,41,53] since late 2022. It has been shown in Refs. [41,53] that a dual non-invertible translation symmetry appears when global internal symmetries that are participating in LSM anomalies are dynamically gauged. Here, we complement this picture by studying the dualities that are induced by gauging a subgroup that *does not participate in the LSM anomaly*.

We are unaware of prior derivations or studies of the KW and JW dual Hamiltonians (4.9) and (4.19) of the quantum spin-1/2 antiferromagnetic $XYZ$ chain with nearest- and next-nearest-neighbor antiferromagnetic couplings obeying periodic boundary conditions. This is also true of the KW dual Hamiltonian (B.9) that is dictated by the choice of open boundary conditions. To the best of our knowledge, KW dualization has been mostly applied in the literature to the (classical) Ising limiting cases of quantum spin-1/2 antiferromagnetic $XYZ$ chains or to the one-dimensional transverse-field Ising model, as is reviewed in Secs. 2 and 4. The derivation and discussion of Figs. 5 and 6 are the main original results of this paper.

The KW and JW dual Hamiltonians (4.9) and (4.19) are examples of quantum Hamiltonians with simultaneous charge, dipole, and translation symmetries. Gauging the charge symmetry simply brings back Hamiltonians (4.9) and (4.19) to the quantum spin-1/2 XYZ chain with the Hamiltonian (4.1) according to the triality encoded by Fig. 1. The study of Hamiltonians with charge and multipolar symmetries has gained popularity (see Refs. [122–126] and references therein).However, our observation that gauging the charge symmetry in the presence of the additional dipole and translation symmetries can produce a local Hamiltonian with translation and global internal symmetry characterized by an LSM anomaly is another original result of this paper.

### 1.4 Organization

The rest of the paper is organized as follows.

In Sec. 2, we review the implementation of the KW and JW dualities as bond-algebra isomorphisms due to gauging an internal $\mathbb{Z}_2$ symmetry. Therein, we establish the triality of three bond algebras.

In Sec. 3, we discuss how additional internal and crystalline symmetries are modified under gauging an internal sub-symmetry. In particular, we show that the LSM anomaly disappears after gauging at the cost of a group extension between crystalline and internal symmetries.

In Sec. 4, we study the phase diagram of the quantum spin-1/2 $XYZ$ chain and its fate under the gauging-related dualities.

Section 5 showcases a generalization to the $\mathbb{Z}_n$-clock models, where we consider an LSM anomaly between internal $\mathbb{Z}_n \times \mathbb{Z}_n$ symmetry and translations and reflections. We conjecture that the LSM anomaly with the reflection symmetry is present only when $n$ is even. We confirm this conjecture by showing that the mixing between reflection and internal symmetries only appears when $n$ is even while mixing with translation is always present. We conclude in Sec. 6.

## 2 Triality of $\mathbb{Z}_2$-symmetric bond algebras on a chain

The first incarnation of duality was discovered by Jordan and Wigner in 1928 [91], who showed by algebraic means that there exists a one-to-one correspondence between creation

and annihilation operators of hard-core bosons on the one hand and spinless fermions on the other hand, provided both can be labeled by an index belonging to an ordered set (as would be the case when this label enumerates the sites of a one-dimensional lattice for example).[2] The second incarnation of duality was discovered by Kramers and Wannier in 1941 [84–90], who showed that the low- and high-temperature expansions of the classical Ising model on the square lattice with nearest-neighbor interactions were related by a one-to-one transformation of the temperature. Common to both incarnations of duality is the following defining property. If there exists a correspondence between a set of observables $\widehat{O}_\iota$ labeled by the index $\iota$ whose (quantum) statistical properties are governed by the (quantum) partition function $Z$ and a second set of observables $\widehat{O}_\iota^\vee$ labeled by the index $\iota^\vee$ whose (quantum) statistical properties are governed by the (quantum) partition function $Z^\vee$ such that the equality

$$\left\langle \prod_\iota \widehat{O}_\iota \right\rangle_Z = \left\langle \prod_{\iota^\vee} \widehat{O}_{\iota^\vee}^\vee \right\rangle_{Z^\vee} , \tag{2.1}$$

between their correlation functions hold, then the pairs of observables $(\widehat{O}_\iota^\vee, \widehat{O}_{\iota^\vee}^\vee)$ and the pair of partition functions $(Z, Z^\vee)$ form dual pairs. The Jordan-Wigner duality was used by Lieb, Schultz, and Mattis to show that the quantum $XY$ spin-1/2 chain with nearest-neighbor anti-ferromagnetic coupling is critical [1]. Kramers and Wannier predicted the value taken by the transition temperature in the Ising model by postulating that it undergoes no more than one transition between the high- and low-temperature phases.

It was recognized by McKean in 1964 that the Kramers-Wannier duality can be derived by means of the Poisson summation formula for the Abelian group $\mathbb{Z}_2$ [92, 100, 107, 108]. In the 1970's, in connection with lattice gauge theories [101], the interplay between global and local symmetries in establishing dualities took center stage starting with Kadanoff and Ceva on the one hand and Wegner on the other hand [93–99, 102–106, 109]. The counterpart to lattice dualities in field theory is bosonization [127–129]. Subtle signatures of lattice dualities in massive field theories were investigated in Refs. [130,131]. An influential approach to dualities was proposed by Fröhlich et al. in 2004 who sought to read off the possible strong/weak-coupling dualities leaving a given critical model fixed solely from knowledge of its universality class [65, 67, 132–136]. A field-theoretical generalization of this approach has been used to study various possible strong/weak-coupling as well as boson/fermion dualities [110, 112, 113].

The goal of this section is to treat the Jordan-Wigner (JW) and Kramers-Wannier (KW) dualities on equal footing. To this end, we are going to review the construction of Kramers-Wannier and Jordan-Wigner dualities obeyed by lattice bond algebras [137, 138] following a gauging approach [51, 111]. Equipped with theses tools, we will present our main results in Sec. 3 in which we study the fate of crystalline transformations of the lattice such as translation and reflection under the dualities (triality) of Sec. 2.

Starting from $\mathbb{Z}_2$-symmetric quantum spin-1/2 $XYZ$ chains defined on the lattice

$$\Lambda := \left\{ j \ \middle| \ j = 1, \cdots, 2N \right\} , \tag{2.2a}$$

we are thus going to gauge the global $\mathbb{Z}_2$ symmetry in two ways. The first way delivers a bosonic bond algebra with global $\mathbb{Z}_2$-symmetry that is supported on the dual lattice

$$\Lambda^\star := \left\{ j^\star \equiv j + \frac{1}{2} \ \middle| \ j \in \Lambda \right\} , \tag{2.2b}$$

---

[2]Wigner and Jordan also introduced in Ref. [91] Majorana operators, i.e., Hermitian operators obeying a Clifford algebra.

i.e., the links of the lattice $\Lambda$. The second way delivers a fermionic bond algebra with global $\mathbb{Z}_2$ fermion parity symmetry that is supported on the lattice $\Lambda$.[3] We will then establish a triality between all three bond algebras, i.e., any pair of the three bond algebras form dual pairs provided appropriate consistency conditions are imposed.

## 2.1 The $\mathbb{Z}_2$ symmetric bond algebra

To each site $j \in \Lambda$, we assign the triplet $\hat{\boldsymbol{\sigma}}_j$ of operators whose components $\hat{\sigma}_j^\alpha$ with $\alpha = x, y, z$ obey the Pauli algebra

$$\hat{\sigma}_j^\alpha \hat{\sigma}_j^\beta = \delta^{\alpha\beta} \widehat{\mathbb{1}}_{\mathcal{H}_b} + \mathrm{i}\epsilon^{\alpha\beta\gamma} \hat{\sigma}_j^\gamma, \qquad \left[\hat{\sigma}_i^\alpha, \hat{\sigma}_j^\beta\right] = 0, \qquad i < j \in \Lambda, \tag{2.3a}$$

where $\alpha, \beta, \gamma = x, y, z$ and the summation convention over repeated indices is implied. We will be interested in organizing the sub-space of linear operators that are symmetric with respect to a $\mathbb{Z}_2^z$ symmetry generated by

$$\widehat{U}_{r_\pi^z} := \prod_{j=1}^{2N} \hat{\sigma}_j^z. \tag{2.3b}$$

Here, $\widehat{U}_{r_\pi^z}$ implements a global rotation by $\pi$ about the $z$ axis in internal spin-1/2 space attached to the lattice $\Lambda$. We consider general symmetry twisted boundary conditions labeled by $b = 0, 1 \in \mathbb{Z}_2^z$ as

$$\hat{\sigma}_{j+2N}^x = \left(\widehat{U}_{r_\pi^z}\right)^b \hat{\sigma}_j^x \left(\widehat{U}_{r_\pi^z}^\dagger\right)^b = (-1)^b \widehat{\sigma}_j^x,$$

$$\hat{\sigma}_{j+2N}^y = \left(\widehat{U}_{r_\pi^z}\right)^b \hat{\sigma}_j^y \left(\widehat{U}_{r_\pi^z}^\dagger\right)^b = (-1)^b \widehat{\sigma}_j^y, \tag{2.3c}$$

$$\hat{\sigma}_{j+2N}^z = \left(\widehat{U}_{r_\pi^z}\right)^b \hat{\sigma}_j^z \left(\widehat{U}_{r_\pi^z}^\dagger\right)^b = \widehat{\sigma}_j^z.$$

These operators act on the $2^{2N}$-dimensional Hilbert space

$$\mathcal{H}_b := \mathrm{span}\left\{\bigotimes_{j\in\Lambda} \left(\frac{\hat{\sigma}_j^x - \mathrm{i}\hat{\sigma}_j^y}{2}\right)^{n_j} |\uparrow\rangle_j \;\middle|\; n_j = 0, 1, \quad \hat{\sigma}_j^z |\uparrow\rangle_j = |\uparrow\rangle_j\right\} \cong \mathbb{C}^{2^{2N}}. \tag{2.3d}$$

We define the bond algebra

$$\mathfrak{B}_b \equiv \left\langle \hat{\sigma}_j^z, \quad \hat{\sigma}_j^x \hat{\sigma}_{j+1}^x \;\middle|\; j \in \Lambda \right\rangle, \tag{2.4}$$

that is spanned by all complex-valued linear combinations of products of the generators $\hat{\sigma}_i^z$ and $\hat{\sigma}_j^x \hat{\sigma}_{j+1}^x$ for any $i, j \in \Lambda$. The bond algebra $\mathfrak{B}_b$ is equivalent to the algebra of all operators acting on the Hilbert space $\mathcal{H}_b$ that are symmetric under the $\mathbb{Z}_2^z$ symmetry. Since the algebra $\mathfrak{B}_b$ is $\mathbb{Z}_2^z$-symmetric, all operators in it can be block diagonalized into eigenspaces of $\widehat{U}_{r_\pi^z}$. Correspondingly, it is also convenient to decompose the Hilbert space (2.3d) in terms of the definite eigenvalues of the operator $\widehat{U}_{r_\pi^z}$, i.e., the decomposition

$$\mathcal{H}_b = \mathcal{H}_{b;+} \oplus \mathcal{H}_{b;-}, \qquad \mathcal{H}_{b;\pm} := \frac{1}{2}\left(\widehat{\mathbb{1}}_{\mathcal{H}_b} \pm \widehat{U}_{r_\pi^z}\right) \mathcal{H}_b. \tag{2.5}$$

In what follows, we are going to construct two additional bond algebras $\mathfrak{B}_{b'}$ and $\mathfrak{B}_f$ and explain under what conditions any pair of the triplet of bond algebras

$$\mathfrak{B}_b, \qquad \mathfrak{B}_{b'}, \qquad \mathfrak{B}_f, \tag{2.6}$$

---

[3]As we shall explain in Sec. 2.3, it will be convenient to implement a unitary transformation which renders the dual fermionic bond algebra on the lattice $\Lambda$.

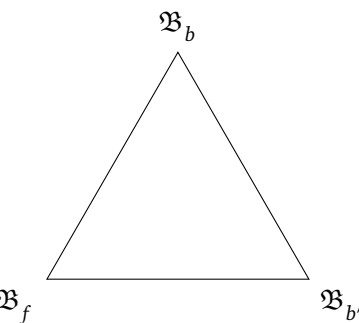

Figure 1: Triality between the triplet of bond algebras (2.6).

are dual to each other. If we place each one of the three bond algebras at the vertices of a triangle as is done in Fig. 1, we may interpret each side of this triangle as a duality relation. We call this web of dualities triality. The strategy that we shall use to establish each of the dualities consists of the following three steps:

1. We gauge the global $\mathbb{Z}_2^z$ symmetry by extending the Hilbert space to include gauge degrees of freedom. This is done in two ways which correspond to introducing bosonic or fermionic gauge degrees of freedom, respectively.

2. We perform a unitary transformation on the extended Hilbert space, which includes both the matter and gauge degrees of freedom. This unitary effectively localizes the Gauss constraints on the matter degrees of freedom.

3. We solve the Gauss constraints and project onto the gauge invariant subspace of the extended Hilbert space. Upon doing so, the matter degrees of freedom freeze out, thus delivering the dual bond algebra.

These dualities are invertible and therefore the same procedure can be carried out starting from either the bond algebra $\mathfrak{B}_{b'}$ or $\mathfrak{B}_f$, as we shall detail below.

## 2.2 Bosonic gauging and the Kramers-Wannier duality

To gauge the global $\mathbb{Z}_2^z$ symmetry generated by the unitary operator (2.3b), we introduce $\mathbb{Z}_2$-valued gauge fields on the links of the lattice $\Lambda$. In other words, to each site $j^\star \in \Lambda^\star$ of the dual lattice, we assign the triplet $\hat{\tau}_{j^\star}$ of operators whose components $\hat{\tau}_{j^\star}^\alpha$ with $\alpha = x, y, z$ obey the Pauli algebra

$$\hat{\tau}_{j^\star}^\alpha \hat{\tau}_{j^\star}^\beta = \delta^{\alpha\beta} \widehat{\mathbb{1}}_{\mathcal{H}_{b'}} + i\epsilon^{\alpha\beta\gamma} \hat{\tau}_{j^\star}^\gamma, \qquad \left[\hat{\tau}_{i^\star}^\alpha, \hat{\tau}_{j^\star}^\beta\right] = 0, \qquad i^\star < j^\star \in \Lambda^\star. \qquad (2.7a)$$

In analogy with (2.3c), we also introduce the twisted boundary conditions

$$\hat{\tau}_{j^\star+2N}^x = (-1)^{b'} \hat{\tau}_{j^\star}^x, \quad \hat{\tau}_{j^\star+2N}^y = (-1)^{b'} \hat{\tau}_{j^\star}^y, \quad \hat{\tau}_{j^\star+2N}^z = \hat{\tau}_{j^\star}^z, \qquad b' = 0, 1. \qquad (2.7b)$$

In what follows, we will see that $b'$ plays an important role in understanding how symmetry eigensectors map under gauging. These link operators act on the $2^{2N}$-dimensional Hilbert space

$$\mathcal{H}_{b'} := \text{span}\left\{ \bigotimes_{j^\star \in \Lambda^\star} \left(\frac{\hat{\tau}_{j^\star}^x - i\hat{\tau}_{j^\star}^y}{2}\right)^{n_{j^\star}} |\uparrow\rangle_{j^\star} \;\middle|\; n_{j^\star} = 0, 1, \quad \hat{\tau}_{j^\star}^z |\uparrow\rangle_{j^\star} = |\uparrow\rangle_{j^\star} \right\} \cong \mathbb{C}^{2^{2N}}. \qquad (2.7c)$$

We define the extended bond algebra

$$\mathfrak{B}_{b,b'} \equiv \left\langle \hat{\sigma}_j^z, \quad \hat{\sigma}_j^x \, \hat{\tau}_{j^\star}^z \, \hat{\sigma}_{j+1}^x \, \middle| \, j \in \Lambda \right\rangle, \tag{2.8a}$$

where $j^\star := j + 1/2$ as defined in Eq. (2.2b) (this is always assumed throughout the paper). Any operator in $\mathfrak{B}_{b,b'}$ acts on the extended $2^{4N}$-dimensional Hilbert space

$$\mathcal{H}_{b,b'} := \mathcal{H}_b \otimes \mathcal{H}_{b'}. \tag{2.8b}$$

What distinguishes $\mathfrak{B}_{b,b'}$ from the set of all operators on $\mathcal{H}_{b,b'}$ is that any element from $\mathfrak{B}_{b,b'}$ is invariant under conjugation with any one of the $2N$ local unitary operators (i.e., Gauss operators)

$$\widehat{G}_{b,b';j} := \hat{\tau}_{j^\star-1}^x \, \hat{\sigma}_j^z \, \hat{\tau}_{j^\star}^x = \widehat{G}_{b,b';j+2N}, \tag{2.9a}$$

that satisfy the conditions

$$\left(\widehat{G}_{b,b';j}\right)^2 = \widehat{\mathbb{1}}_{\mathcal{H}_{b,b'}}, \qquad \left[\widehat{G}_{b,b';i}, \widehat{G}_{b,b';j}\right] = 0, \qquad \prod_j \widehat{G}_{b,b';j} = (-1)^{b'} \widehat{U}_{r_\pi^z}. \tag{2.9b}$$

Products of operators $\widehat{G}_{b,b';j}$ over subsets of $\Lambda$ generate local $\mathbb{Z}_2$ gauge transformations. A generic local $\mathbb{Z}_2$ gauge transformation

$$\hat{\sigma}_j^x \mapsto (-1)^{\lambda_j} \hat{\sigma}_j^x, \qquad \hat{\sigma}_j^y \mapsto (-1)^{\lambda_j} \hat{\sigma}_j^y, \qquad \hat{\sigma}_j^z \mapsto \hat{\sigma}_j^z, \tag{2.10a}$$

$$\hat{\tau}_{j^\star}^x \mapsto \hat{\tau}_{j^\star}^x, \qquad \hat{\tau}_{j^\star}^y \mapsto (-1)^{\lambda_{j+1}-\lambda_j} \hat{\tau}_{j^\star}^y, \qquad \hat{\tau}_{j^\star}^z \mapsto (-1)^{\lambda_{j+1}-\lambda_j} \hat{\tau}_{j^\star}^z, \tag{2.10b}$$

with $\lambda_j = 0, 1$ is implemented by the operator

$$\widehat{G}_{b,b';\boldsymbol{\lambda}} := \prod_{j=1}^{2N} \left(\widehat{G}_{b,b';j}\right)^{\lambda_j}, \qquad \boldsymbol{\lambda} = (\lambda_1, \cdots, \lambda_{2N}), \tag{2.10c}$$

and specified by the string of $\mathbb{Z}_2$-valued scalars $\lambda_j = 0, 1$ with $j = 1, \cdots, 2N$. While the bond algebra (2.8a) is invariant under any local transformations (2.10), these are yet to be associated to *gauge symmetries* or, equivalently, redundancies in our description.[4] We elevate the local transformations (2.10) to local gauge symmetries by requiring that any two states in the Hilbert space $\mathcal{H}_{b,b'}$ are equivalent if they are related by a gauge transformation. In particular, we demand that any state $|\psi_{\mathrm{phys}}\rangle$ is a physical one if and only if

$$\widehat{G}_{b,b';j} |\psi_{\mathrm{phys}}\rangle = +|\psi_{\mathrm{phys}}\rangle, \tag{2.11a}$$

for any $j = 1, \cdots, 2N$. The choice of + sign for any $j$ corresponds to a background with no $\mathbb{Z}_2$ matter, i.e., we have a pure gauge theory. Observe that Eqs. (2.11a) and (2.9b) imply that

$$\widehat{U}_{r_\pi^z} |\psi_{\mathrm{phys}}\rangle = (-1)^{b'} |\psi_{\mathrm{phys}}\rangle. \tag{2.11b}$$

We project the extended Hilbert space (2.8b) into the gauge invariant sector where condition (2.11) holds. This is facilitated by first performing the unitary transformation [51] (see also Refs. [103, 109, 139–141])

$$\widehat{U}_{b,b'} := \prod_{j=1}^{2N} \widehat{U}_{b,b';j^\star}, \qquad \widehat{U}_{b,b';j^\star} := \widehat{P}_{b,b';j^\star}^+ + \hat{\tau}_{j^\star}^x \, \widehat{P}_{b,b';j^\star}^-, \tag{2.12a}$$

---

[4]To make the connection with gauge theories in $(1+1)$-dimensional spacetime continuum, we observe that $\hat{\tau}_{j^\star}^x$ is reminiscent of a $\mathbb{Z}_2$-valued electric field $\sim e^{iE}$, while $\hat{\tau}_{j^\star}^z$ is reminiscent of a $\mathbb{Z}_2$-valued gauge field $\sim e^{iA}$. Accordingly, under the local transformation specified by a $\mathbb{Z}_2$-valued scalar field $\boldsymbol{\lambda}$, the electric field is invariant, while the gauge field changes by $\pi\,\mathrm{d}\boldsymbol{\lambda}$.

with pairwise commuting projectors

$$\widehat{P}_{b,b';j^\star}^{\pm} := \frac{1}{2}\left(\widehat{\mathbb{1}}_{\mathcal{H}_{b,b'}} \pm \hat{\sigma}_j^x \hat{\sigma}_{j+1}^x\right), \qquad j^\star \in \Lambda^\star. \tag{2.12b}$$

For any $j = 1, \cdots, 2N$, there follows the transformation laws

$$\widehat{U}_{b,b'}\,\hat{\sigma}_j^x\left(\widehat{U}_{b,b'}\right)^\dagger = \hat{\sigma}_j^x, \qquad\qquad \widehat{U}_{b,b'}\,\hat{\sigma}_j^z\left(\widehat{U}_{b,b'}\right)^\dagger = \hat{\tau}_{j^\star-1}^x\,\hat{\sigma}_j^z\,\hat{\tau}_{j^\star}^x, \tag{2.13a}$$

$$\widehat{U}_{b,b'}\,\hat{\tau}_{j^\star}^x\left(\widehat{U}_{b,b'}\right)^\dagger = \hat{\tau}_{j^\star}^x, \qquad\qquad \widehat{U}_{b,b'}\,\hat{\tau}_{j^\star}^z\left(\widehat{U}_{b,b'}\right)^\dagger = \hat{\sigma}_j^x\,\hat{\tau}_{j^\star}^z\,\hat{\sigma}_{j+1}^x, \tag{2.13b}$$

for the generators of the Pauli algebras on the lattices $\Lambda$ and $\Lambda^\star$ together with the image

$$\widehat{U}_{b,b'}\,\widehat{G}_{b,b';j}\left(\widehat{U}_{b,b'}\right)^\dagger = \hat{\sigma}_j^z, \tag{2.13c}$$

of the local Gauss operator.

Thus, projection onto the subspace where the condition (2.11) holds amounts to setting the action of $\hat{\sigma}_j^z$ on physical states to the identity ($\hat{\sigma}_j^z \equiv 1$) after the unitary transformation (2.13). More concretely, if we define the projector

$$\widehat{P}_{b,b';\mathrm{G}} := \prod_{j\in\Lambda} \frac{1}{2}\left[\widehat{\mathbb{1}}_{\mathcal{H}_{b,b'}} + \widehat{U}_{b,b'}\,\widehat{G}_{b,b';j}\left(\widehat{U}_{b,b'}\right)^\dagger\right], \tag{2.14a}$$

onto the $2^{2N}$-dimensional gauge-invariant subspace

$$\mathcal{H}_{b'}^\vee := \widehat{P}_{b,b';\mathrm{G}}\,\mathcal{H}_{b,b'} \subset \mathcal{H}_{b,b'}, \tag{2.14b}$$

we find that the $2N$ triplets of projected operators

$$\hat{\tau}_{j^\star}^{x\vee} := \widehat{P}_{b,b';\mathrm{G}}\left[\widehat{U}_{b,b'}\,\hat{\tau}_{j^\star}^x\left(\widehat{U}_{b,b'}\right)^\dagger\right]\widehat{P}_{b,b';\mathrm{G}}$$
$$= \widehat{P}_{b,b';\mathrm{G}}\,\hat{\tau}_{j^\star}^x\,\widehat{P}_{b,b';\mathrm{G}}, \tag{2.15a}$$

$$\hat{\tau}_{j^\star}^{y\vee} := \widehat{P}_{b,b';\mathrm{G}}\left[\widehat{U}_{b,b'}\,\hat{\sigma}_j^x\,\hat{\tau}_{j^\star}^y\,\hat{\sigma}_{j+1}^x\left(\widehat{U}_{b,b'}\right)^\dagger\right]\widehat{P}_{b,b';\mathrm{G}}$$
$$= \widehat{P}_{b,b';\mathrm{G}}\,\hat{\tau}_{j^\star}^y\,\widehat{P}_{b,b';\mathrm{G}}, \tag{2.15b}$$

$$\hat{\tau}_{j^\star}^{z\vee} := \widehat{P}_{b,b';\mathrm{G}}\left[\widehat{U}_{b,b'}\,\hat{\sigma}_j^x\,\hat{\tau}_{j^\star}^z\,\hat{\sigma}_{j+1}^x\left(\widehat{U}_{b,b'}\right)^\dagger\right]\widehat{P}_{b,b';\mathrm{G}}$$
$$= \widehat{P}_{b,b';\mathrm{G}}\,\hat{\tau}_{j^\star}^z\,\widehat{P}_{b,b';\mathrm{G}}, \tag{2.15c}$$

realize a Pauli algebra on the Hilbert space $\mathcal{H}_{b'}^\vee$ that is isomorphic to the Pauli algebra (2.7a) on the Hilbert space $\mathcal{H}_{b'}$. This implies that the projection to $\mathcal{H}_{b'}^\vee$ of the bond algebra $\mathfrak{B}_{b,b'}$ delivers the dual bond algebra

$$\mathfrak{B}_{b'} := \widehat{P}_{b,b';\mathrm{G}}\left[\widehat{U}_{b,b'}\,\mathfrak{B}_{b,b'}\left(\widehat{U}_{b,b'}\right)^\dagger\right]\widehat{P}_{b,b';\mathrm{G}}$$
$$= \left\langle \hat{\tau}_{j^\star-1}^{x\vee}\,\hat{\tau}_{j^\star}^{x\vee}, \quad \hat{\tau}_{j^\star}^{z\vee} \,\middle|\, j^\star \in \Lambda^\star \right\rangle. \tag{2.16}$$

This is the bond algebra of operators that are symmetric under the dual $\mathbb{Z}_2^{z^\vee}$ symmetry with the generator

$$\widehat{U}_{r_\pi^z}^\vee := \prod_{j^\star\in\Lambda^\star} \hat{\tau}_{j^\star}^{z\vee}, \tag{2.17}$$

of the global rotation by $\pi$ about the $z$ axis in internal spin-1/2 space attached to the dual lattice $\Lambda^\star$. Note that the twisted boundary conditions in (2.7b) were nothing but symmetry twisted

boundary conditions with respect to $\mathbb{Z}_2^{z^\vee}$. As was done for the Hilbert space $\mathcal{H}_b$ in Eq. (2.5), it is convenient to decompose the Hilbert space $\mathcal{H}_{b'}^\vee$ in terms of the definite eigenvalue sectors of the operator $\widehat{U}_{r_\pi^z}^\vee$, i.e., the decomposition

$$\mathcal{H}_{b'}^\vee = \mathcal{H}_{b';+}^\vee \oplus \mathcal{H}_{b';-}^\vee, \tag{2.18a}$$

holds, where

$$\mathcal{H}_{b';\pm}^\vee := \frac{1}{2}\left(\widehat{\mathbb{1}}_{\mathcal{H}_{b'}^\vee} \pm \widehat{U}_{r_\pi^z}^\vee\right)\mathcal{H}_{b'}^\vee. \tag{2.18b}$$

The duality between the bond algebras (2.4) and (2.16) demands the following consistency conditions. Because of the twisted boundary conditions (2.3c) and (2.7b), the pair of operators

$$\left(\hat{\sigma}_j^z, \qquad \hat{\tau}_{j^\star-1}^{x\,\vee}\hat{\tau}_{j^\star}^{x\,\vee}\right), \tag{2.19a}$$

and the pair of operators

$$\left(\hat{\sigma}_j^x\hat{\sigma}_{j+1}^x, \qquad \hat{\tau}_{j^\star}^{z\,\vee}\right), \tag{2.19b}$$

each form a dual pair if and only if the pair of operators

$$\left(\prod_{j=1}^{2N}\hat{\sigma}_j^z = \widehat{U}_{r_\pi^z}, \qquad \prod_{j=1}^{2N}\left(\hat{\tau}_{j^\star-1}^{x\,\vee}\hat{\tau}_{j^\star}^{x\,\vee}\right) = (-1)^{b'}\widehat{\mathbb{1}}_{\mathcal{H}_{b'}^\vee}\right), \tag{2.20a}$$

and the pair of operators

$$\left(\prod_{j=1}^{2N}\left(\hat{\sigma}_j^x\hat{\sigma}_{j+1}^x\right) = (-1)^b\widehat{\mathbb{1}}_{\mathcal{H}_b}, \qquad \prod_{j=1}^{2N}\hat{\tau}_{j^\star}^{z\,\vee} =: \widehat{U}_{r_\pi^z}^\vee\right), \tag{2.20b}$$

each form a dual pair, respectively. This is to say that duality between the bond algebras (2.4) and (2.16) holds only on the $2^{2N-1}$-dimensional subspaces

$$\mathcal{H}_{b;(-1)^{b'}} = \frac{1}{2}\left[\widehat{\mathbb{1}}_{\mathcal{H}_b} + (-1)^{b'}\widehat{U}_{r_\pi^z}\right]\mathcal{H}_b, \tag{2.21a}$$

$$\mathcal{H}_{b';(-1)^b}^\vee = \frac{1}{2}\left[\widehat{\mathbb{1}}_{\mathcal{H}_{b'}^\vee} + (-1)^b\widehat{U}_{r_\pi^z}^\vee\right]\mathcal{H}_{b'}^\vee, \tag{2.21b}$$

of $2^{2N}$-dimensional Hilbert spaces $\mathcal{H}_b$ and $\mathcal{H}_{b'}^\vee$, respectively. This duality between the bond algebras (2.4) and (2.16) acting on Hilbert spaces (2.21a) and (2.21b), respectively, is nothing but the Kramers-Wannier (KW) duality. Under the KW duality, the boundary conditions $(b = 0, 1)$ of the bond algebra (2.4) dictates the eigenvalue of the generator $\widehat{U}_{r_\pi^z}^\vee$ of the global dual symmetry, while the eigenvalue of the generator $\widehat{U}_{r_\pi^z}$ of the global symmetry that was gauged dictates the boundary conditions $(b' = 0, 1)$ of the dual bond algebra (2.16). Table 1 summarizes this correspondence (see Ref. [110] for an alternative field-theoretical derivation of this mapping of the symmetry eigensectors under the KW duality).

## 2.3 Fermionic gauging and the Jordan-Wigner duality

We now describe a distinct fermionic gauging of the global $\mathbb{Z}_2^z$ symmetry with generator Eq. (2.3b). In contrast to the previous section, we introduce a pair of Majorana operators on every link of the lattice $\Lambda$. These represent fermionic gauge degrees of freedom. More precisely, to

Table 1: The four Kramers-Wannier dualizations that follow from the consistency conditions (2.21). The first column specifies the twisted boundary conditions. The choice of twisted boundary conditions is selected by $b = 0, 1$ prior to dualization. The choice of twisted boundary conditions is selected by $b' = 0, 1$ after dualization. The second column gives the dual subspace of the Hilbert space prior to dualization. The third column gives the dual subspace of the Hilbert space after dualization.

| $(b, b')$ | $\mathcal{H}_{b;(-1)^{b'}}$ | $\mathcal{H}^{\vee}_{b';(-1)^{b}}$ |
|---|---|---|
| $(0,0)$ | $\frac{1}{2}\left(\widehat{\mathbb{1}}_{\mathcal{H}_b} + \widehat{U}_{r^z_\pi}\right)\mathcal{H}_b$ | $\frac{1}{2}\left(\widehat{\mathbb{1}}_{\mathcal{H}^{\vee}_{b'}} + \widehat{U}^{\vee}_{r^z_\pi}\right)\mathcal{H}^{\vee}_{b'}$ |
| $(1,0)$ | $\frac{1}{2}\left(\widehat{\mathbb{1}}_{\mathcal{H}_b} + \widehat{U}_{r^z_\pi}\right)\mathcal{H}_b$ | $\frac{1}{2}\left(\widehat{\mathbb{1}}_{\mathcal{H}^{\vee}_{b'}} - \widehat{U}^{\vee}_{r^z_\pi}\right)\mathcal{H}^{\vee}_{b'}$ |
| $(0,1)$ | $\frac{1}{2}\left(\widehat{\mathbb{1}}_{\mathcal{H}_b} - \widehat{U}_{r^z_\pi}\right)\mathcal{H}_b$ | $\frac{1}{2}\left(\widehat{\mathbb{1}}_{\mathcal{H}^{\vee}_{b'}} + \widehat{U}^{\vee}_{r^z_\pi}\right)\mathcal{H}^{\vee}_{b'}$ |
| $(1,1)$ | $\frac{1}{2}\left(\widehat{\mathbb{1}}_{\mathcal{H}_b} - \widehat{U}_{r^z_\pi}\right)\mathcal{H}_b$ | $\frac{1}{2}\left(\widehat{\mathbb{1}}_{\mathcal{H}^{\vee}_{b'}} - \widehat{U}^{\vee}_{r^z_\pi}\right)\mathcal{H}^{\vee}_{b'}$ |

each site $j^\star \in \Lambda^\star$, we assign the pair $\hat{\beta}_{j^\star} = \hat{\beta}^{\dagger}_{j^\star}$ and $\hat{\alpha}_{j^\star} = \hat{\alpha}^{\dagger}_{j^\star}$ of Majorana operators obeying the Clifford algebra

$$\left\{\hat{\alpha}_{i^\star}, \hat{\alpha}_{j^\star}\right\} = \left\{\hat{\beta}_{i^\star}, \hat{\beta}_{j^\star}\right\} = 2\delta_{i^\star, j^\star}\widehat{\mathbb{1}}_{\mathcal{H}_f}, \qquad \left\{\hat{\alpha}_{i^\star}, \hat{\beta}_{j^\star}\right\} = 0, \quad i^\star, j^\star \in \Lambda^\star. \tag{2.22a}$$

We also introduce the fermion parity operator

$$\widehat{P}_{\mathrm{F}} := \prod_{j^\star \in \Lambda^\star} \left(\mathrm{i}\hat{\beta}_{j^\star}\, \hat{\alpha}_{j^\star}\right), \tag{2.22b}$$

together with the cyclic group

$$\mathbb{Z}^{\mathrm{F}}_2 = \left\{p_{\mathrm{F}}, \left(p_{\mathrm{F}}\right)^2 \equiv e\right\}, \tag{2.22c}$$

of order two with $p_{\mathrm{F}}$ represented by $\widehat{P}_{\mathrm{F}}$. The pair $\widehat{P}_{\mathrm{F}}$ and $\mathbb{Z}^{\mathrm{F}}_2$ will play a central role in what follows. We work in a Hilbert space with boundary conditions twisted with respect to the fermion parity operator, i.e.,

$$\begin{aligned} \hat{\alpha}_{j^\star+2N} &= \left(\widehat{P}_{\mathrm{F}}\right)^f \hat{\alpha}_{j^\star} \left(\widehat{P}^{\dagger}_{\mathrm{F}}\right)^f = (-1)^f\, \hat{\alpha}_{j^\star}, \\ \hat{\beta}_{j^\star+2N} &= \left(\widehat{P}_{\mathrm{F}}\right)^f \hat{\beta}_{j^\star} \left(\widehat{P}^{\dagger}_{\mathrm{F}}\right)^f = (-1)^f\, \hat{\beta}_{j^\star}, \end{aligned} \tag{2.22d}$$

where $f = 0, 1$. These $2N$ doublets of Majorana operators act on the $2^{2N}$-dimensional Hilbert space

$$\mathcal{H}_f := \mathrm{span}\left\{ \left[\prod_{j^\star \in \Lambda^\star} \left(\frac{\hat{\beta}_{j^\star} - \mathrm{i}\hat{\alpha}_{j^\star}}{2}\right)^{n_{j^\star}}\right] |0\rangle \;\middle|\; n_{j^\star} = 0, 1, \qquad \frac{\hat{\beta}_{j^\star} + \mathrm{i}\hat{\alpha}_{j^\star}}{2}|0\rangle = 0 \right\} \cong \mathbb{C}^{2^{2N}}. \tag{2.22e}$$

We define the extended bond algebra

$$\mathfrak{B}_{b,f} \equiv \left\langle \hat{\sigma}^z_j, \quad \hat{\sigma}^x_j \left(\mathrm{i}\hat{\beta}_{j^\star}\, \hat{\alpha}_{j^\star}\right)\hat{\sigma}^x_{j+1} \;\middle|\; j \in \Lambda \right\rangle, \tag{2.23a}$$

that is spanned by all complex-valued linear combinations of products of the generators $\hat{\sigma}^z_i$ and $\hat{\sigma}^x_j \left(\mathrm{i}\hat{\beta}_{j^\star}\, \hat{\alpha}_{j^\star}\right)\hat{\sigma}^x_{j+1}$ for any $i, j \in \Lambda$. Any element of $\mathfrak{B}_{b,f}$ acts on the extended $2^{4N}$-dimensional Hilbert space

$$\mathcal{H}_{b,f} := \mathcal{H}_b \otimes \mathcal{H}_f. \tag{2.23b}$$

What distinguishes $\mathfrak{B}_{b,f}$ from the set of all operators on $\mathcal{H}_{b,f}$ is that any element from $\mathfrak{B}_{b,f}$ is invariant under conjugation with any one of the $2N$ local unitary operators (i.e., Gauss operators)

$$\widehat{G}_{b,f;j} := i\hat{\beta}_{j^\star-1}\,\hat{\sigma}_j^z\,\hat{\alpha}_{j^\star} = \widehat{G}_{b,f;j+2N}\,, \tag{2.24a}$$

that satisfy the conditions

$$\left(\widehat{G}_{b,f;j}\right)^2 = \widehat{\mathbb{1}}_{\mathcal{H}_{b,f}}\,, \qquad \left[\widehat{G}_{b,f;i},\widehat{G}_{b,f;j}\right] = 0\,, \qquad \prod_{j\in\Lambda}\widehat{G}_{b,f;j} = (-1)^f\,\widehat{P}_{\mathrm{F}}\,\widehat{U}_{r_\pi^z}\,. \tag{2.24b}$$

Products of operators $\widehat{G}_{b,f;j}$ over subsets of $\Lambda$ generate $\mathbb{Z}_2$ gauge transformations. A generic $\mathbb{Z}_2$ gauge transformation

$$\hat{\sigma}_j^x \mapsto (-1)^{\lambda_j}\,\hat{\sigma}_j^x\,, \qquad \hat{\sigma}_j^y \mapsto (-1)^{\lambda_j}\,\hat{\sigma}_j^y\,, \qquad\qquad \hat{\sigma}_j^z \mapsto \hat{\sigma}_j^z\,, \tag{2.25a}$$

$$\hat{\alpha}_{j^\star} \mapsto (-1)^{\lambda_j}\,\hat{\alpha}_{j^\star}\,, \qquad \hat{\beta}_{j^\star} \mapsto (-1)^{\lambda_{j+1}}\,\hat{\beta}_{j^\star}\,, \tag{2.25b}$$

with $\lambda_j = 0,1$ is implemented by the operator

$$\widehat{G}_{b,f;\boldsymbol{\lambda}} := \prod_{j=1}^{2N}\left(\widehat{G}_{b,f;j}\right)^{\lambda_j}\,, \qquad \boldsymbol{\lambda} = (\lambda_1,\cdots,\lambda_{2N})\,, \tag{2.25c}$$

and specified by the string of $\mathbb{Z}_2$-valued scalars $\lambda_j = 0,1$ with $j = 1,\cdots,2N$. While the bond algebra (2.23a) is invariant under any local transformations (2.25), these are yet to be associated to *gauge symmetries* or, equivalently, redundancies in our description.[5] We elevate the local transformations (2.25) to local gauge symmetries by requiring that any two states in the Hilbert space $\mathcal{H}_{b,f}$ are equivalent if they are related by a gauge transformation. In particular, we demand that any state $|\psi_{\mathrm{phys}}\rangle$ is a physical one if and only if

$$\widehat{G}_{b,f;j}\,|\psi_{\mathrm{phys}}\rangle = +|\psi_{\mathrm{phys}}\rangle\,, \tag{2.26a}$$

for any $j = 1,\cdots,2N$. The choice of $+$ sign for any $j$ corresponds to a background with no $\mathbb{Z}_2$ matter, i.e., we have a pure gauge theory. Observe that Eqs. (2.26a) and (2.24b) imply that (compare with Eq. (2.11b))

$$\widehat{U}_{r_\pi^z}\,|\psi_{\mathrm{phys}}\rangle = (-1)^f\,\widehat{P}_{\mathrm{F}}\,|\psi_{\mathrm{phys}}\rangle\,. \tag{2.26b}$$

We project the extended Hilbert space (2.23b) into the gauge invariant sector where condition (2.26) holds. This is facilitated by first performing the unitary transformation

$$\widehat{U}_{b,f} := \prod_{j=1}^{2N}\widehat{U}_{b,f;j}\,, \tag{2.27a}$$

where

$$\widehat{U}_{b,f;j} := \left(\hat{\sigma}_j^x\right)^{\widehat{P}_{b,f;j}^-} = \widehat{P}_{b,f;j}^+ + \hat{\sigma}_j^x\,\widehat{P}_{b,f;j}^- = \widehat{U}_{b,f;j+2N}\,, \tag{2.27b}$$

with pairwise commuting projectors

$$\widehat{P}_{b,f;j}^{\pm} := \frac{1}{2}\left(\widehat{\mathbb{1}}_{\mathcal{H}_{b,f}} \pm i\hat{\beta}_{j^\star-1}\,\hat{\alpha}_{j^\star}\right)\,, \qquad j\in\Lambda\,. \tag{2.27c}$$

---

[5]To make the connection with gauge theories in $(1+1)$-dimensional continuum, we observe that both $i\hat{\beta}_{j^\star-1}\,\hat{\alpha}_{j^\star}$ and $i\hat{\beta}_{j^\star}\,\hat{\alpha}_{j^\star}$ are reminiscent of a $\mathbb{Z}_2$-valued electric field $\sim e^{iE}$ and a $\mathbb{Z}_2$-valued gauge field $\sim e^{iA}$ in that they obey the same transformation laws under local $\mathbb{Z}_2$ gauge transformations, respectively.

For any $j \in \Lambda$, there follows the transformation rules

$$\widehat{U}_{b,f} \, \hat{\sigma}_j^x \left(\widehat{U}_{b,f}\right)^\dagger = \hat{\sigma}_j^x \,, \qquad\qquad \widehat{U}_{b,f} \, \hat{\sigma}_j^z \left(\widehat{U}_{b,f}\right)^\dagger = \mathrm{i}\hat{\beta}_{j^\star - 1}\, \hat{\sigma}_j^z\, \hat{\alpha}_{j^\star},$$
$$\widehat{U}_{b,f} \, \hat{\beta}_{j^\star} \left(\widehat{U}_{b,f}\right)^\dagger = \hat{\beta}_{j^\star}\, \hat{\sigma}_{j+1}^x\,, \qquad \widehat{U}_{b,f} \, \hat{\alpha}_{j^\star} \left(\widehat{U}_{b,f}\right)^\dagger = \hat{\sigma}_j^x\, \hat{\alpha}_{j^\star}\,, \tag{2.28a}$$

for the spin operators on the lattice $\Lambda$ and Majorana operators on the dual lattice $\Lambda^\star$ together with the image

$$\widehat{U}_{b,f} \, \widehat{G}_{b,f;j} \left(\widehat{U}_{b,f}\right)^\dagger = \hat{\sigma}_j^z\,. \tag{2.28b}$$

Thus, projection onto the subspace where the condition (2.26) holds amounts to setting the action of $\hat{\sigma}_j^z$ on physical states to the identity ($\hat{\sigma}_j^z \equiv 1$) after the unitary transformation (2.28). More concretely, if we define the projector

$$\widehat{P}_{b,f;\mathrm{G}} := \prod_{j \in \Lambda} \frac{1}{2}\left[\widehat{\mathbb{1}}_{\mathcal{H}_{b,f}} + \widehat{U}_{b,f}\, \widehat{G}_{b,f;j}\left(\widehat{U}_{b,f}\right)^\dagger\right], \tag{2.29a}$$

onto the $2^{2N}$-dimensional gauge-invariant subspace

$$\mathcal{H}_f^\vee := \widehat{P}_{b,f;\mathrm{G}}\, \mathcal{H}_{b,f} \subset \mathcal{H}_{b,f}\,, \tag{2.29b}$$

we find that the $2N$ doublets of projected operators

$$\hat{\beta}_{j+1}^\vee := \widehat{P}_{b,f;\mathrm{G}}\left[\widehat{U}_{b,f}\left(\hat{\beta}_{j^\star}\, \hat{\sigma}_{j+1}^x\right)\left(\widehat{U}_{b,f}\right)^\dagger\right]\widehat{P}_{b,f;\mathrm{G}}$$
$$= \widehat{P}_{b,f;\mathrm{G}}\, \hat{\beta}_{j^\star}\, \widehat{P}_{b,f;\mathrm{G}}\,, \tag{2.30a}$$
$$\hat{\alpha}_j^\vee := \widehat{P}_{b,f;\mathrm{G}}\left[\widehat{U}_{b,f}\left(\hat{\sigma}_j^x\, \hat{\alpha}_{j^\star}\right)\left(\widehat{U}_{b,f}\right)^\dagger\right]\widehat{P}_{b,f;\mathrm{G}}$$
$$= \widehat{P}_{b,f;\mathrm{G}}\, \hat{\alpha}_{j^\star}\, \widehat{P}_{b,f;\mathrm{G}}\,, \tag{2.30b}$$

realize a Clifford algebra on the Hilbert space $\mathcal{H}_f^\vee$ that is isomorphic to the Clifford algebra (2.22a) on the Hilbert space $\mathcal{H}_f$. The lattice label that we choose for $\hat{\beta}_{j+1}^\vee$ and $\hat{\alpha}_j^\vee$ is a matter of convention since the relation between $j$ and $j^\star = j + \frac{1}{2}$ is one to one.[6] We also find that the projection to $\mathcal{H}_f^\vee$ of the bond algebra $\mathfrak{B}_{b,f}$ is the bond algebra

$$\mathfrak{B}_f := \left\langle \mathrm{i}\hat{\beta}_j^\vee\, \hat{\alpha}_j^\vee, \qquad \mathrm{i}\hat{\beta}_{j+1}^\vee\, \hat{\alpha}_j^\vee \,\middle|\, j \in \Lambda \right\rangle, \tag{2.31a}$$

which is the algebra of operators invariant under conjugation by the generator

$$\widehat{P}_{\mathrm{F}}^\vee := \prod_{j \in \Lambda}\left(\mathrm{i}\hat{\beta}_j^\vee\, \hat{\alpha}_j^\vee\right), \tag{2.31b}$$

of a global fermion-parity symmetry $\mathbb{Z}_2^{\mathrm{F}}$. As was done for the Hilbert space $\mathcal{H}_b$ in Eq. (2.5), it is convenient to decompose the Hilbert space $\mathcal{H}_f^\vee$ to the definite eigenvalue sectors of the operator $\widehat{P}_{\mathrm{F}}^\vee$, i.e., the decomposition

$$\mathcal{H}_f^\vee = \mathcal{H}_{f;+}^\vee \oplus \mathcal{H}_{f;-}^\vee\,, \tag{2.32a}$$

---

[6]This choice implies a relative translation of the $\hat{\beta}_j^\vee$ operators compared to the $\hat{\alpha}_j^\vee$ operators. It is done to simplify the discussion of the phase diagram in Sec. 4. As we shall see in Sec. 2.4, while such a "half"-translation is a unitary transformation on the fermionic bond algebras, it corresponds to implementing the KW duality described in Sec. 2.2 on the bosonic bond algebras obtained by gauging fermion parity.

Table 2: The four Jordan-Wigner dualizations that follow from the consistency conditions (2.34). The first column specifies the twisted boundary conditions. The choice of twisted boundary conditions is selected by $b = 0, 1$ prior to dualization. The choice of twisted boundary conditions is selected by $f = 0, 1$ after dualization. The second column gives the dual subspace of the Hilbert space prior to dualization. The third column gives the dual subspace of the Hilbert space after dualization.

| $(b, f)$ | $\mathcal{H}_{b;(-1)^{b+f+1}}$ | $\mathcal{H}^{\vee}_{f;(-1)^{b+f+1}}$ |
|---|---|---|
| $(0,0)$ | $\frac{1}{2}\left(\widehat{\mathbb{1}}_{\mathcal{H}_b} - \widehat{U}_{r^z_\pi}\right)\mathcal{H}_b$ | $\frac{1}{2}\left(\widehat{\mathbb{1}}_{\mathcal{H}^{\vee}_f} - \widehat{P}^{\vee}_{\mathrm{F}}\right)\mathcal{H}^{\vee}_f$ |
| $(1,0)$ | $\frac{1}{2}\left(\widehat{\mathbb{1}}_{\mathcal{H}_b} + \widehat{U}_{r^z_\pi}\right)\mathcal{H}_b$ | $\frac{1}{2}\left(\widehat{\mathbb{1}}_{\mathcal{H}^{\vee}_f} + \widehat{P}^{\vee}_{\mathrm{F}}\right)\mathcal{H}^{\vee}_f$ |
| $(0,1)$ | $\frac{1}{2}\left(\widehat{\mathbb{1}}_{\mathcal{H}_b} + \widehat{U}_{r^z_\pi}\right)\mathcal{H}_b$ | $\frac{1}{2}\left(\widehat{\mathbb{1}}_{\mathcal{H}^{\vee}_f} + \widehat{P}^{\vee}_{\mathrm{F}}\right)\mathcal{H}^{\vee}_f$ |
| $(1,1)$ | $\frac{1}{2}\left(\widehat{\mathbb{1}}_{\mathcal{H}_b} - \widehat{U}_{r^z_\pi}\right)\mathcal{H}_b$ | $\frac{1}{2}\left(\widehat{\mathbb{1}}_{\mathcal{H}^{\vee}_f} - \widehat{P}^{\vee}_{\mathrm{F}}\right)\mathcal{H}^{\vee}_f$ |

holds where

$$\mathcal{H}^{\vee}_{f;\pm} := \frac{1}{2}\left(\widehat{\mathbb{1}}_{\mathcal{H}^{\vee}_f} \pm \widehat{P}^{\vee}_{\mathrm{F}}\right)\mathcal{H}^{\vee}_f. \tag{2.32b}$$

The duality between the bond algebras (2.4) and (2.31) demands certain consistency conditions. In particular, the twisted boundary conditions (2.3c) and (2.22d) require that the pairs of operators

$$\left(\hat{\sigma}^z_j, \quad \mathrm{i}\hat{\beta}^{\vee}_j \hat{\alpha}^{\vee}_j\right), \tag{2.33a}$$

and

$$\left(\hat{\sigma}^x_j \hat{\sigma}^x_{j+1}, \quad \mathrm{i}\hat{\beta}^{\vee}_{j+1} \hat{\alpha}^{\vee}_j\right), \tag{2.33b}$$

each form a dual pair if and only if the pairs of operators

$$\left(\prod_{j=1}^{2N} \hat{\sigma}^z_j = \widehat{U}_{r^z_\pi}, \quad \prod_{j=1}^{2N}\left(\mathrm{i}\hat{\beta}^{\vee}_j \hat{\alpha}^{\vee}_j\right) = \widehat{P}^{\vee}_{\mathrm{F}}\right), \tag{2.33c}$$

and

$$\left(\prod_{j=1}^{2N}\left(\hat{\sigma}^x_j \hat{\sigma}^x_{j+1}\right) = (-1)^b \widehat{\mathbb{1}}_{\mathcal{H}_b}, \quad \prod_{j=1}^{2N}\left(\mathrm{i}\hat{\beta}^{\vee}_{j+1} \hat{\alpha}^{\vee}_j\right) = (-1)^{f+1} \widehat{P}^{\vee}_{\mathrm{F}}\right), \tag{2.33d}$$

each form a dual pair, respectively. This is to say that the duality between the bond algebras (2.4) and (2.31) holds only on the $2^{2N-1}$-dimensional subspaces

$$\mathcal{H}_{b;(-1)^{b+f+1}} = \frac{1}{2}\left[\widehat{\mathbb{1}}_{\mathcal{H}_b} + (-1)^{b+f+1} \widehat{U}_{r^z_\pi}\right]\mathcal{H}_b, \tag{2.34a}$$

$$\mathcal{H}^{\vee}_{f;(-1)^{b+f+1}} = \frac{1}{2}\left[\widehat{\mathbb{1}}_{\mathcal{H}^{\vee}_f} + (-1)^{b+f+1} \widehat{P}^{\vee}_{\mathrm{F}}\right]\mathcal{H}^{\vee}_f, \tag{2.34b}$$

of $2^{2N}$-dimensional Hilbert spaces $\mathcal{H}_b$ and $\mathcal{H}^{\vee}_f$, respectively. This is the Jordan-Wigner (JW) duality. Table 2 summarizes the correspondence between symmetry eigensectors on either side of this duality (see Refs. [41, 110] for an alternative field-theoretical derivation of the mapping of symmetry eigensectors under the JW duality).

## 2.4 A triality of bond algebras

In Sec. 2.2, we gauged the internal global symmetry group $\mathbb{Z}_2^z$ of the bond algebra $\mathfrak{B}_b$ defined in Eq. (2.4) by minimal coupling to the local generator $\hat{\tau}_{j^\star}^z$ of rotation by $\pi$ about the $z$ axis in internal spin-1/2 space of the site $j^\star \in \Lambda^\star$ with the help of the local Gauss operator defined in Eq. (2.9). In Sec. 2.3, we gauged instead the internal global symmetry group $\mathbb{Z}_2^z$ by minimal coupling to the local generator $\mathrm{i}\hat{\beta}_{j^\star}\,\hat{\alpha}_{j^\star}$ of fermion-parity on the site $j^\star \in \Lambda^\star$, with the help of the local Gauss operator defined in Eq. (2.24).

To complete the triality,[7] we construct the two dualizations $\mathfrak{B}_{b'}$ and $\mathfrak{B}_b$ of the bond algebra

$$\mathfrak{B}_f \equiv \left\langle \mathrm{i}\hat{\beta}_j\,\hat{\alpha}_j\,,\ \mathrm{i}\hat{\beta}_{j+1}\,\hat{\alpha}_j\ \middle|\ j \in \Lambda \right\rangle, \tag{2.35a}$$

where the Majorana operators $\hat{\alpha}_j = \hat{\alpha}_j^\dagger$ and $\hat{\beta}_j = \hat{\beta}_j^\dagger$ with $i, j \in \Lambda$ satisfy the Clifford algebra

$$\left\{\hat{\alpha}_j, \hat{\alpha}_{j'}\right\} = \left\{\hat{\beta}_j, \hat{\beta}_{j'}\right\} = 2\,\delta_{j,j'}\,, \qquad \left\{\hat{\alpha}_j, \hat{\beta}_{j'}\right\} = 0\,, \tag{2.35b}$$

and obey the fermion-parity twisted boundary conditions

$$\hat{\alpha}_{j+2N} = (-1)^f\,\hat{\alpha}_j\,, \qquad \hat{\beta}_{j+2N} = (-1)^f\,\hat{\beta}_j\,, \qquad f = 0, 1\,, \tag{2.35c}$$

for any $j, j' \in \Lambda$. The domain of definition of these $2N$ doublets of Majorana operators is the Hilbert space

$$\mathcal{H}_f := \mathrm{span}\left\{ \left[\prod_{j=1}^{2N}\left(\frac{\hat{\beta}_j - \mathrm{i}\hat{\alpha}_j}{2}\right)^{n_j}\right]|0\rangle\ \middle|\ n_j = 0, 1\,,\ \frac{\hat{\beta}_j + \mathrm{i}\hat{\alpha}_j}{2}|0\rangle = 0 \right\} \cong \mathbb{C}^{2^{2N}}\,. \tag{2.35d}$$

The bond algebra (2.35a) is symmetric under conjugation by the global fermion parity.[8]

$$\widehat{P}_{\mathrm{F}} := \prod_{j=1}^{2N} \mathrm{i}\hat{\beta}_j\,\hat{\alpha}_j \tag{2.36a}$$

$$= (-1)^{f+1}\prod_{j=1}^{2N} \mathrm{i}\hat{\beta}_j\,\hat{\alpha}_{j+1}\,. \tag{2.36b}$$

We are going to show that gauging the global fermion parity symmetry generated by the representation (2.36a) of $\widehat{P}_{\mathrm{F}}$ delivers the bond algebra $\mathfrak{B}_{b'}$ on the dual lattice, while gauging the global fermion parity symmetry generated by the representation (2.36b) of $\widehat{P}_{\mathrm{F}}$ delivers the bond algebra $\mathfrak{B}_b$ on the dual lattice.

### 2.4.1 Unit-cell preserving gauging of fermion parity

We trade the bond algebra $\mathfrak{B}_f$ defined in Eq. (2.35) by the minimally coupled bond algebra

$$\mathfrak{B}_{f,b'} := \left\langle \mathrm{i}\hat{\beta}_j\,\hat{\alpha}_j\,,\ \mathrm{i}\hat{\beta}_{j+1}\,\hat{\tau}_{j^\star}^z\,\hat{\alpha}_j\ \middle|\ j \in \Lambda \right\rangle, \tag{2.37a}$$

with the tensor product

$$\mathcal{H}_{f,b'} := \mathcal{H}_f \otimes \mathcal{H}_{b'}\,, \tag{2.37b}$$

---

[7]We leave it to the reader to construct the two dualizations $\mathfrak{B}_b$ and $\mathfrak{B}_f$ by gauging the bond algebra $\mathfrak{B}_{b'}$.

[8]The rational for choosing the multiplicative real-valued phase factor on the right-hand side of Eq. (2.31b) will be given in Sec. 3.3.

as domain of definition [here, $\mathcal{H}_{b'}$ was defined in Eq. (2.7)]. This extended bond algebra is symmetric with respect to any one of the $2N$ pairwise-commuting Gauss operators

$$\widehat{G}_{f,b';j} := \hat{\tau}^x_{j^\star-1}\left(\mathrm{i}\hat{\beta}_j\,\hat{\alpha}_j\right)\hat{\tau}^x_{j^\star} = \widehat{G}_{f,b';j+2N}, \qquad j\in\Lambda\,. \tag{2.38}$$

These are Gauss operators since they will soon be used to define kinematic constraints on the Hilbert space as is standard in the gauging procedure. We call these Gauss operators (2.38) unit-cell preserving, for the local transformation it implements on the Majorana operators only acts non-trivially on a single site of the direct lattice $\Lambda$. One verifies that

$$\prod_{j\in\Lambda}\widehat{G}_{f,b';j} = \prod_{j\in\Lambda}\hat{\tau}^x_{j^\star-1}\left(\mathrm{i}\hat{\beta}_j\,\hat{\alpha}_j\right)\hat{\tau}^x_{j^\star} = (-1)^{b'}\prod_{j\in\Lambda}\left(\mathrm{i}\hat{\beta}_j\,\hat{\alpha}_j\right) \equiv (-1)^{b'}\,\widehat{P}_{\mathrm{F}}\,. \tag{2.39}$$

In words, the product of all local Gauss operators equals the global fermion parity up to a sign fixed by the twisted boundary conditions $b' = 0, 1$.

We define the unitary transformation of the Hilbert space (2.37b) through

$$\widehat{U}_{f,b'} := \prod_{j\in\Lambda}\widehat{U}_{f,b';j}, \qquad \widehat{U}_{f,b';j} := \left(\hat{\tau}^x_{j^\star}\right)^{\widehat{P}^-_{f,b';j}} = \widehat{P}^+_{f,b';j} + \hat{\tau}^x_{j^\star}\,\widehat{P}^-_{f,b';j}\,, \tag{2.40a}$$

with the $2N$ pairwise-commuting projectors

$$\widehat{P}^\pm_j := \frac{1\pm\mathrm{i}\hat{\beta}_{j+1}\,\hat{\alpha}_j}{2} = \widehat{P}^\pm_{j+2N}\,. \tag{2.40b}$$

For any $j = 1, \ldots, 2N$, the following transformation laws hold

$$\widehat{U}_{f,b'}\,\hat{\tau}^x_{j^\star}\left(\widehat{U}_{f,b'}\right)^\dagger = \hat{\tau}^x_{j^\star}, \qquad \widehat{U}_{f,b'}\,\hat{\tau}^z_{j^\star}\left(\widehat{U}_{f,b'}\right)^\dagger = \mathrm{i}\hat{\beta}_{j+1}\,\hat{\tau}^z_{j^\star}\,\hat{\alpha}_j\,, \tag{2.41a}$$

$$\widehat{U}_{f,b'}\,\hat{\beta}_j\left(\widehat{U}_{f,b'}\right)^\dagger = \hat{\tau}^x_{j^\star-1}\,\hat{\beta}_j\,, \qquad \widehat{U}_{f,b'}\,\hat{\alpha}_j\left(\widehat{U}_{f,b'}\right)^\dagger = \hat{\alpha}_j\,\hat{\tau}^x_{j^\star}\,, \tag{2.41b}$$

for the operators on the lattices $\Lambda$ and $\Lambda^\star$ together with the image

$$\widehat{U}_{f,b'}\,\widehat{G}_{f,b';j}\left(\widehat{U}_{f,b'}\right)^\dagger = \mathrm{i}\hat{\beta}_j\,\hat{\alpha}_j\,, \tag{2.41c}$$

of the local Gauss operator. Thus, if we define the projector

$$\widehat{P}_{f,b';\mathrm{G}} := \prod_{j\in\Lambda}\frac{1}{2}\left[\widehat{\mathbb{1}}_{\mathcal{H}_{f,b'}} + \widehat{U}_{f,b'}\,\widehat{G}_{f,b';j}\left(\widehat{U}_{f,b'}\right)^\dagger\right]\,, \tag{2.42a}$$

onto the $2^{2N}$-dimensional gauge-invariant subspace

$$\mathcal{H}^\vee_{b'} := \widehat{P}_{f,b';\mathrm{G}}\,\mathcal{H}_{f,b'} \subset \mathcal{H}_{f,b'}\,, \tag{2.42b}$$

we find that the $2N$ triplets of projected operators

$$\hat{\tau}^{x\,\vee}_{j^\star} := \widehat{P}_{f,b';\mathrm{G}}\left[\widehat{U}_{f,b'}\,\hat{\tau}^x_{j^\star}\left(\widehat{U}_{f,b'}\right)^\dagger\right]\widehat{P}_{f,b';\mathrm{G}}$$
$$= \widehat{P}_{f,b';\mathrm{G}}\,\hat{\tau}^x_{j^\star}\,\widehat{P}_{f,b';\mathrm{G}}\,, \tag{2.43a}$$

$$\hat{\tau}^{y\,\vee}_{j^\star} := \widehat{P}_{f,b';\mathrm{G}}\left[\widehat{U}_{f,b'}\,\mathrm{i}\hat{\beta}_{j+1}\,\hat{\tau}^y_{j^\star}\,\hat{\alpha}_j\left(\widehat{U}_{f,b'}\right)^\dagger\right]\widehat{P}_{f,b';\mathrm{G}}$$
$$= \widehat{P}_{f,b';\mathrm{G}}\,\hat{\tau}^y_{j^\star}\,\widehat{P}_{f,b';\mathrm{G}}\,, \tag{2.43b}$$

$$\hat{\tau}^{z\,\vee}_{j^\star} := \widehat{P}_{f,b';\mathrm{G}}\left[\widehat{U}_{f,b'}\,\mathrm{i}\hat{\beta}_{j+1}\,\hat{\tau}^z_{j^\star}\,\hat{\alpha}_j\left(\widehat{U}_{f,b'}\right)^\dagger\right]\widehat{P}_{f,b';\mathrm{G}}$$
$$= \widehat{P}_{f,b';\mathrm{G}}\,\hat{\tau}^z_{j^\star}\,\widehat{P}_{f,b';\mathrm{G}}\,, \tag{2.43c}$$

Table 3: Operator dualities for the triality between the bond algebras $\mathfrak{B}_f$ defined in Eq. (2.37), $\mathfrak{B}_{b'}$ defined in Eq. (2.44), and $\mathfrak{B}_b$ defined in Eq. (2.52). Any two operators from the same column form a dual pair.

| Symbol for bond algebra | Generator | Generator | Symmetry group generated by |
|:---:|:---:|:---:|:---:|
| $\mathfrak{B}_f$ | $\mathrm{i}\hat{\beta}_j\,\hat{\alpha}_j$ | $\mathrm{i}\hat{\beta}_{j+1}\,\hat{\alpha}_j$ | $\widehat{P}_{\mathrm{F}} = \prod\limits_{j\in\Lambda}\left(\mathrm{i}\hat{\beta}_j\,\hat{\alpha}_j\right)$ |
| $\mathfrak{B}_{b'}$ | $\hat{\tau}^{x\,\vee}_{j^\star-1}\,\hat{\tau}^{x\,\vee}_{j^\star}$ | $\hat{\tau}^{z\,\vee}_{j^\star}$ | $\widehat{U}^{\vee}_{r^z_\pi} = \prod\limits_{j^\star\in\Lambda^\star}\hat{\tau}^{z\,\vee}_{j^\star}$ |
| $\mathfrak{B}_b$ | $\hat{\sigma}^{z\,\vee}_{j^\star}$ | $\hat{\sigma}^{x\,\vee}_{j^\star}\,\hat{\sigma}^{x\,\vee}_{j^\star+1}$ | $\widehat{U}^{\vee}_{r^z_\pi} = \prod\limits_{j^\star\in\Lambda^\star}\hat{\sigma}^{z\,\vee}_{j^\star}$ |

realize the same Pauli algebra and obey the same twisted boundary conditions as the $2N$ triplets $\hat{\tau}_{j^\star}$ on the dual lattice. We also find that the projection to $\mathcal{H}^{\vee}_{b'}$ of the bond algebra $\mathfrak{B}_{f,b'}$ is the bond algebra

$$
\begin{aligned}
\mathfrak{B}_{b'} &:= \widehat{P}_{f,b';\mathrm{G}}\left[\widehat{U}_{f,b'}\,\mathfrak{B}_{f,b'}\left(\widehat{U}_{f,b'}\right)^\dagger\right]\widehat{P}_{f,b';\mathrm{G}} \\
&= \left\langle \hat{\tau}^{x\,\vee}_{j^\star-1}\,\hat{\tau}^{x\,\vee}_{j^\star},\ \hat{\tau}^{z\,\vee}_{j^\star}\ \middle|\ j^\star\in\Lambda^\star \right\rangle,
\end{aligned}
\tag{2.44a}
$$

which is symmetric with respect to a $\mathbb{Z}_2$ symmetry generated by

$$
\widehat{U}^{\vee}_{r^z_\pi} := \prod_{j^\star\in\Lambda^\star}\hat{\tau}^{z\,\vee}_{j^\star},
\tag{2.44b}
$$

of the global rotation by $\pi$ about the $z$ axis in internal spin-1/2 space attached to the dual lattice $\Lambda^\star$. We have recovered Eq. (2.16) starting from the bond algebra $\mathfrak{B}_f$ instead of the bond algebra $\mathfrak{B}_b$. The duality from $\mathfrak{B}_f$ to $\mathfrak{B}_{b'}$ is summarized in Tables 3 and 4.

### 2.4.2 Unit-cell non-preserving gauging of fermion parity

Next, we trade the bond algebra $\mathfrak{B}_f$ defined in Eq. (2.35) by the minimally coupled bond algebra

$$
\mathfrak{B}_{f,b} := \left\langle \mathrm{i}\hat{\beta}_j\,\hat{\sigma}^z_{j^\star}\,\hat{\alpha}_j,\qquad \mathrm{i}\hat{\beta}_{j+1}\,\hat{\alpha}_j\ \middle|\ j\in\Lambda \right\rangle,
\tag{2.45a}
$$

with the tensor product

$$
\mathcal{H}_{f,b} := \mathcal{H}_f\otimes\mathcal{H}_b,
\tag{2.45b}
$$

as domain of definition [here, $\mathcal{H}_b$ is defined as in Eq. (2.3d) except for the substitution of $\Lambda$ by $\Lambda^\star$]. This extended bond algebra is invariant under conjugation by any one of the $2N$ pairwise-commuting Gauss operators

$$
\widehat{G}_{f,b;j} := \hat{\sigma}^x_{j^\star+1}\left(\mathrm{i}\hat{\beta}_{j+1}\,\hat{\alpha}_j\right)\hat{\sigma}^x_{j^\star} = \widehat{G}_{f,b;j+2N},\qquad j\in\Lambda.
\tag{2.46}
$$

We call the Gauss operator (2.46) unit-cell non-preserving, for the local transformation it implements on the Majorana operators only acts non-trivially on two consecutive sites of the direct lattice $\Lambda$. We observe that the Gauss operator (2.46) can be obtained from the Gauss operator (2.38) by translating only the $\hat{\beta}_j$ operators by one unit-cell. As we shall see, such

a "half"-translation of the bond algebra (2.35) will deliver the KW dual of the bosonic bond algebra (2.44). One verifies that

$$\prod_{j\in\Lambda}\widehat{G}_{f,b;j} = \prod_{j\in\Lambda}\hat{\sigma}^x_{j^\star+1}\left(i\hat{\beta}_{j+1}\,\hat{\alpha}_j\right)\hat{\sigma}^x_{j^\star} = (-1)^b\prod_{j\in\Lambda}\left(i\hat{\beta}_j\,\hat{\alpha}_{j+1}\right) \equiv (-1)^{b+f+1}\,\widehat{P}_F. \qquad (2.47)$$

In words, the product of all local Gauss operators equals the global fermion parity up to a sign fixed by the twisted boundary conditions $f,b = 0,1$.

We define the unitary transformation of the Hilbert space (2.45b) through

$$\widehat{U}_{f,b} := \prod_{j\in\Lambda}\widehat{U}_{f,b;j}, \qquad \widehat{U}_{f,b;j} := \left(\hat{\sigma}^x_{j^\star}\right)^{\widehat{P}^-_{f,b;j}} = \widehat{P}^+_{f,b;j} + \hat{\sigma}^x_{j^\star}\,\widehat{P}^-_{f,b;j}, \qquad (2.48a)$$

with the $2N$ pairwise-commuting projectors

$$\widehat{P}^\pm_j := \frac{1\pm i\hat{\beta}_j\,\hat{\alpha}_j}{2} = \widehat{P}^\pm_{j+2N}. \qquad (2.48b)$$

For any $j = 1,\cdots,2N$, there follows the transformation laws

$$\widehat{U}_{f,b}\,\hat{\sigma}^x_{j^\star}\left(\widehat{U}_{f,b}\right)^\dagger = \hat{\sigma}^x_{j^\star}, \qquad \widehat{U}_{f,b}\,\hat{\sigma}^z_{j^\star}\left(\widehat{U}_{f,b}\right)^\dagger = i\hat{\beta}_j\,\hat{\sigma}^z_{j^\star}\,\hat{\alpha}_j, \qquad (2.49a)$$

$$\widehat{U}_{f,b}\,\hat{\beta}_j\left(\widehat{U}_{f,b}\right)^\dagger = \hat{\beta}_j\,\hat{\sigma}^x_{j^\star}, \qquad \widehat{U}_{f,b}\,\hat{\alpha}_j\left(\widehat{U}_{f,b}\right)^\dagger = \hat{\alpha}_j\,\hat{\sigma}^x_{j^\star}, \qquad (2.49b)$$

for the operators on the lattices $\Lambda$ and $\Lambda^\star$ together with the image

$$\widehat{U}_{f,b}\,\widehat{G}_{f,b;j}\left(\widehat{U}_{f,b}\right)^\dagger = i\hat{\beta}_{j+1}\,\hat{\alpha}_j, \qquad (2.49c)$$

of the local Gauss operator. Thus, if we define the projector

$$\widehat{P}_{f,b;G} := \prod_{j\in\Lambda}\frac{1}{2}\left[\widehat{\mathbb{1}}_{\mathcal{H}_{f,b}} + \widehat{U}_{f,b}\,\widehat{G}_{f,b;j}\left(\widehat{U}_{f,b}\right)^\dagger\right], \qquad (2.50a)$$

onto the $2^{2N}$-dimensional gauge-invariant subspace

$$\mathcal{H}^\vee_b := \widehat{P}_{f,b;G}\,\mathcal{H}_{f,b} \subset \mathcal{H}_{f,b}, \qquad (2.50b)$$

we find that the $2N$ triplets of projected operators

$$\begin{aligned}\hat{\sigma}^{x\vee}_{j^\star} &:= \widehat{P}_{f,b;G}\left[\widehat{U}_{f,b}\,\hat{\sigma}^x_{j^\star}\left(\widehat{U}_{f,b}\right)^\dagger\right]\widehat{P}_{f,b;G} \\ &= \widehat{P}_{f,b;G}\,\hat{\sigma}^x_{j^\star}\,\widehat{P}_{f,b;G}, \end{aligned} \qquad (2.51a)$$

$$\begin{aligned}\hat{\sigma}^{y\vee}_{j^\star} &:= \widehat{P}_{f,b;G}\left[\widehat{U}_{f,b}\,i\hat{\beta}_j\,\hat{\sigma}^y_{j^\star}\,\hat{\alpha}_j\left(\widehat{U}_{f,b}\right)^\dagger\right]\widehat{P}_{f,b;G} \\ &= \widehat{P}_{f,b;G}\,\hat{\sigma}^y_{j^\star}\,\widehat{P}_{f,b;G}, \end{aligned} \qquad (2.51b)$$

$$\begin{aligned}\hat{\sigma}^{z\vee}_{j^\star} &:= \widehat{P}_{f,b;G}\left[\widehat{U}_{f,b}\,i\hat{\beta}_j\,\hat{\sigma}^z_{j^\star}\,\hat{\alpha}_j\left(\widehat{U}_{f,b}\right)^\dagger\right]\widehat{P}_{f,b;G} \\ &= \widehat{P}_{f,b;G}\,\hat{\sigma}^z_{j^\star}\,\widehat{P}_{f,b;G}, \end{aligned} \qquad (2.51c)$$

realize the same Pauli algebra and obey the same twisted boundary conditions as the $2N$ triplets $\hat{\sigma}_{j^\star}$ on the dual lattice. We also find that the projection to $\mathcal{H}^\vee_b$ of the bond algebra $\mathfrak{B}_{f,b}$ is the bond algebra

$$\begin{aligned}\mathfrak{B}_b &:= \widehat{P}_{f,b;G}\left[\widehat{U}_{f,b}\,\mathfrak{B}_{f,b}\left(\widehat{U}_{f,b}\right)^\dagger\right]\widehat{P}_{f,b;G} \\ &= \left\langle\hat{\sigma}^{z\vee}_{j^\star}, \qquad \hat{\sigma}^{x\vee}_{j^\star}\hat{\sigma}^{x\vee}_{j^\star+1}\,\Big|\,j^\star\in\Lambda^\star\right\rangle, \end{aligned} \qquad (2.52a)$$

Table 4: Compatibility conditions for the triality between the bond algebras $\mathfrak{B}_f$ defined in Eq. (2.37), $\mathfrak{B}_{b'}$ defined in Eq. (2.44), and $\mathfrak{B}_b$ defined in Eq. (2.52). The Hilbert space on which the bond algebra $\mathfrak{B}_f$ is defined is $\mathcal{H}_f \cong \mathbb{C}^{2^{2N}}$ with $f = 0, 1$ selecting the twisted boundary conditions. The Hilbert space on which the bond algebra $\mathfrak{B}_{b'}$ is defined is $\mathcal{H}_{b'}^\vee \cong \mathbb{C}^{2^{2N}}$ with $b' = 0, 1$ selecting the twisted boundary conditions. The Hilbert space on which the bond algebra $\mathfrak{B}_b$ is defined is $\mathcal{H}_b^\vee \cong \mathbb{C}^{2^{2N}}$ with $b = 0, 1$ selecting the twisted boundary conditions. Choosing two out of the triplet $(f, b', b)$ determines the third according to the rule $f = b + b' + 1 \bmod 2$. Triality is defined by the fact that duality holds between any two bond algebras $\mathfrak{B}_f$, $\mathfrak{B}_{b'}$, and $\mathfrak{B}_b$ provided their domain of definitions are restricted to $\mathcal{H}_{f;(-1)^{b+f+1}} \cong \mathbb{C}^{2^{2N-1}}$, $\mathcal{H}_{b';(-1)^b}^\vee \cong \mathbb{C}^{2^{2N-1}}$, and $\mathcal{H}_{b;(-1)^{b'}}^\vee \cong \mathbb{C}^{2^{2N-1}}$, respectively.

| $(f, b', b)$ | $\mathcal{H}_{f;(-1)^{b+f+1}}$ | $\mathcal{H}_{b';(-1)^b}^\vee$ | $\mathcal{H}_{b;(-1)^{b'}}^\vee$ |
|---|---|---|---|
| $(0,1,0)$ | $\frac{1}{2}\left(\widehat{\mathbb{1}}_{\mathcal{H}_f} - \widehat{P}_{\mathrm{F}}\right)\mathcal{H}_f$ | $\frac{1}{2}\left(\widehat{\mathbb{1}}_{\mathcal{H}_{b'}^\vee} + \widehat{U}_{r_\pi^z}^\vee\right)\mathcal{H}_{b'}^\vee$ | $\frac{1}{2}\left(\widehat{\mathbb{1}}_{\mathcal{H}_b^\vee} - \widehat{U}_{r_\pi^z}^\vee\right)\mathcal{H}_b^\vee$ |
| $(0,0,1)$ | $\frac{1}{2}\left(\widehat{\mathbb{1}}_{\mathcal{H}_f} + \widehat{P}_{\mathrm{F}}\right)\mathcal{H}_f$ | $\frac{1}{2}\left(\widehat{\mathbb{1}}_{\mathcal{H}_{b'}^\vee} - \widehat{U}_{r_\pi^z}^\vee\right)\mathcal{H}_{b'}^\vee$ | $\frac{1}{2}\left(\widehat{\mathbb{1}}_{\mathcal{H}_b^\vee} + \widehat{U}_{r_\pi^z}^\vee\right)\mathcal{H}_b^\vee$ |
| $(1,0,0)$ | $\frac{1}{2}\left(\widehat{\mathbb{1}}_{\mathcal{H}_f} + \widehat{P}_{\mathrm{F}}\right)\mathcal{H}_f$ | $\frac{1}{2}\left(\widehat{\mathbb{1}}_{\mathcal{H}_{b'}^\vee} + \widehat{U}_{r_\pi^z}^\vee\right)\mathcal{H}_{b'}^\vee$ | $\frac{1}{2}\left(\widehat{\mathbb{1}}_{\mathcal{H}_b^\vee} + \widehat{U}_{r_\pi^z}^\vee\right)\mathcal{H}_b^\vee$ |
| $(1,1,1)$ | $\frac{1}{2}\left(\widehat{\mathbb{1}}_{\mathcal{H}_f} - \widehat{P}_{\mathrm{F}}\right)\mathcal{H}_f$ | $\frac{1}{2}\left(\widehat{\mathbb{1}}_{\mathcal{H}_{b'}^\vee} - \widehat{U}_{r_\pi^z}^\vee\right)\mathcal{H}_{b'}^\vee$ | $\frac{1}{2}\left(\widehat{\mathbb{1}}_{\mathcal{H}_b^\vee} - \widehat{U}_{r_\pi^z}^\vee\right)\mathcal{H}_b^\vee$ |

with the generator

$$\widehat{U}_{r_\pi^z}^\vee := \prod_{j^\star \in \Lambda^\star} \hat{\sigma}_{j^\star}^{z\,\vee}, \tag{2.52b}$$

of the global rotation by $\pi$ about the $z$ axis in internal spin-1/2 space attached to the dual lattice $\Lambda^\star$. We have recovered Eq. (2.4) starting from the bond algebra $\mathfrak{B}_f$ (up to the substitution $\Lambda \to \Lambda^\star$). The duality from $\mathfrak{B}_f$ to $\mathfrak{B}_b$ is summarized in Tables 3 and 4.

Demanding the triality of the three bond algebras $\mathfrak{B}_f$, $\mathfrak{B}_{b'}$, and $\mathfrak{B}_b$ that are defined in Eqs. (2.35a), (2.44a), and (2.52a), respectively, puts a constraint on the possible boundary conditions specified by the triplet $(b, b', f)$ of twisted boundary conditions. Indeed, this triality implies that one may start from any one of these bond algebras located at the vertices in Fig. 1 and execute two successive dualities in such a way that the two remaining vertices from Fig. 1 are visited. The duality between the bond algebras $\mathfrak{B}_f$ and $\mathfrak{B}_{b'}$ holds on the restricted subspaces

$$\mathcal{H}_{f;(-1)^{b'}} \longleftrightarrow \mathcal{H}_{b';(-1)^{f+b'+1}}^\vee, \tag{2.53a}$$

while the duality between the bond algebras $\mathfrak{B}_f$ and $\mathfrak{B}_b$ holds on the restricted subspaces

$$\mathcal{H}_{f;(-1)^{b+f+1}} \longleftrightarrow \mathcal{H}_{b;(-1)^{b+f+1}}^\vee. \tag{2.53b}$$

Let us choose the corresponding subspaces of $\mathcal{H}_f$ in Eqs. (2.53a) and (2.53b) to be identical. We then find that the triality of all three bond algebras holds when

$$\mathcal{H}_{b';(-1)^{f+b'+1}}^\vee \longleftrightarrow \mathcal{H}_{f;(-1)^{b'}} \equiv \mathcal{H}_{f;(-1)^{b+f+1}} \longleftrightarrow \mathcal{H}_{b;(-1)^{f+b+1}}^\vee, \tag{2.54a}$$

which implies the relation

$$f = b + b' + 1 \bmod 2. \tag{2.54b}$$

The duality between $\mathcal{H}^{\vee}_{b';(-1)^{f+b'+1}}$ and $\mathcal{H}^{\vee}_{b;(-1)^{f+b+1}}$ follows since one can first dualize the former to obtain $\mathcal{H}_{f;(-1)^{b'}}$ and then dualize $\mathcal{H}_{f;(-1)^{b+f+1}}$ to obtain $\mathcal{H}^{\vee}_{b;(-1)^{f+b+1}}$ if the condition (2.54b) holds.

## 3 LSM anomalies and triality

In Sec. 2, we have established dualities between any two of the three bond algebras $\mathfrak{B}_b$, $\mathfrak{B}_{b'}$, and $\mathfrak{B}_f$. What is common to all three bond algebras is the presence of a cyclic symmetry group of order two, namely $\mathbb{Z}_2^z$, $\mathbb{Z}_2^{z^{\vee}}$, and $\mathbb{Z}_2^F$, respectively. Any Hamiltonian $\widehat{H}_b$ that is an element of the bond algebra $\mathfrak{B}_b$ has a $\mathbb{Z}_2^z$ symmetry generated by $\widehat{U}_{r_{\pi}^z}$. It follows that its duals $\widehat{H}_{b'}^{\vee}$ and $\widehat{H}_f^{\vee}$ obtained by KW and JW dualities, respectively, are symmetric under the dual symmetries $\mathbb{Z}_2^{z^{\vee}}$ and $\mathbb{Z}_2^F$, respectively. The question that we address in this section is the fate of additional crystalline and internal symmetries of such a Hamiltonian $\widehat{H}_b$ under the dualities described in Sec. 2. In particular, we show how the presence of an LSM anomaly manifests itself in the dual bond algebras $\mathfrak{B}_{b'}$ and $\mathfrak{B}_f$.

### 3.1 Symmetry structure with an LSM anomaly

We consider the bond algebra $\mathfrak{B}_b$ with $b = 0$.[9] and impose two independent crystalline symmetries of the lattice $\Lambda$, namely translation and (site-centered) reflection

$$
\begin{aligned}
\widehat{U}_t \, \hat{\boldsymbol{\sigma}}_j \, \widehat{U}_t^{\dagger} &= \hat{\boldsymbol{\sigma}}_{t(j)}, & t(j) &:= j + 1 \bmod 2N, \\
\widehat{U}_r \, \hat{\boldsymbol{\sigma}}_j \, \widehat{U}_r^{\dagger} &= \hat{\boldsymbol{\sigma}}_{r(j)}, & r(j) &:= 2N - j \bmod 2N,
\end{aligned}
\tag{3.1a}
$$

implemented by the unitary operators

$$
\widehat{U}_t := \prod_{j=1}^{2N-1} \frac{1}{2}\left(\widehat{\mathbb{1}}_{\mathcal{H}_{b=0}} + \hat{\boldsymbol{\sigma}}_j \cdot \hat{\boldsymbol{\sigma}}_{t(j)}\right), \qquad \widehat{U}_r := \prod_{j=1}^{N-1} \frac{1}{2}\left(\widehat{\mathbb{1}}_{\mathcal{H}_{b=0}} + \hat{\boldsymbol{\sigma}}_j \cdot \hat{\boldsymbol{\sigma}}_{r(j)}\right),
\tag{3.1b}
$$

respectively. The product on the second term in Eq. (3.1b) has the upper bound $N-1$ since the reflection has two fixed points $N$ and $2N$ in $\Lambda$. The pair of operators $\widehat{U}_t$ and $\widehat{U}_r$ generates a $2^{2N}$-dimensional representation of the space group

$$G_{\text{spa}} := \mathbb{Z}_{2N}^t \rtimes \mathbb{Z}_2^r, \tag{3.2a}$$

with

$$\mathbb{Z}_{2N}^t \equiv \left\{t, t^2, \cdots, t^{2N-1}, t^{2N} \equiv e\right\}, \qquad \mathbb{Z}_2^r \equiv \left\{r, r^2 \equiv e\right\}, \qquad r\,t = t^{2N-1}\,r. \tag{3.2b}$$

Next, we impose the global internal symmetries implemented by the unitary operators

$$\widehat{U}_{r_{\pi}^x} := \prod_{j=1}^{2N} \hat{\sigma}_j^x, \qquad \widehat{U}_{r_{\pi}^y} := \prod_{j=1}^{2N} \hat{\sigma}_j^y, \qquad \widehat{U}_{r_{\pi}^z} := (-1)^N \, \widehat{U}_{r_{\pi}^x} \widehat{U}_{r_{\pi}^y} = \prod_{j=1}^{2N} \hat{\sigma}_j^z. \tag{3.3}$$

---

[9]This choice ensures that the total symmetry group is a direct product of crystalline and internal symmetries.

The pair of operators $\widehat{U}_{r_\pi^x}$ and $\widehat{U}_{r_\pi^y}$ generates a $2^{2N}$-dimensional representation of the global internal symmetry group

$$G_{\text{int}} \equiv \mathbb{Z}_2^x \times \mathbb{Z}_2^y, \tag{3.4a}$$

with

$$\mathbb{Z}_2^x \equiv \left\{ r_\pi^x, (r_\pi^x)^2 \equiv e \right\}, \qquad \mathbb{Z}_2^y \equiv \left\{ r_\pi^y, (r_\pi^y)^2 \equiv e \right\}. \tag{3.4b}$$

Note that the $\mathbb{Z}_2^z$ symmetry of the bond algebra (2.4) corresponds to the diagonal element in the group $\mathbb{Z}_2^x \times \mathbb{Z}_2^y$, i.e., $r_\pi^z = r_\pi^x r_\pi^y$. Importantly, the total symmetry group has the direct product structure

$$G_{\text{tot}} \equiv G_{\text{spa}} \times G_{\text{int}}. \tag{3.5}$$

While the global representation of the $\mathbb{Z}_2^x \times \mathbb{Z}_2^y$ group in Eq. (3.3) is a group homomorphism, it is locally projective due to the Pauli algebra

$$\hat{\sigma}_j^x \hat{\sigma}_j^y = -\hat{\sigma}_j^y \hat{\sigma}_j^x, \qquad \hat{\sigma}_j^y \hat{\sigma}_j^z = -\hat{\sigma}_j^z \hat{\sigma}_j^y, \qquad \hat{\sigma}_j^z \hat{\sigma}_j^x = -\hat{\sigma}_j^x \hat{\sigma}_j^z, \tag{3.6}$$

for any $j \in \Lambda$. Therefore, it follows that the presence of the $G_{\text{tot}}$ symmetry constrains the phase diagram of any symmetric Hamiltonian owing to the (generalized) Lieb-Schultz-Mattis (LSM) Theorems. We are going to invoke two LSM Theorems [1,26,31] that apply to one-dimensional spin chains with translation and reflection symmetries, respectively. Importantly, the proofs of these two theorems make use of the fact that the total symmetry group (3.5) is a direct product of the crystalline space group with the internal group. This is another motivation for choosing $b = 0$. In what follows, $|\Lambda|$ denotes the cardinality ($2N$) of the set $\Lambda$.

**Theorem 1** (Translation LSM). Consider a one-dimensional lattice Hamiltonian with the symmetry group $G_{\text{tot}} \equiv Z_{|\Lambda|}^t \times \mathbb{Z}_2^x \times \mathbb{Z}_2^y$, where the subgroup $Z_{|\Lambda|}^t$ generates lattice translations and the subgroup $\mathbb{Z}_2^x \times \mathbb{Z}_2^y$ generates internal discrete spin-rotation symmetry. If the unit cell with respect to the translation symmetry $Z_{|\Lambda|}^t$ hosts a half-integer spin representation of $\mathbb{Z}_2^x \times \mathbb{Z}_2^y$, then the ground states cannot be simultaneously gapped, non-degenerate, and $G_{\text{tot}}$-symmetric.

**Definition 1** (Translation LSM anomaly). When Theorem 1 holds, we say that there is a translation LSM anomaly.

**Theorem 2** (Reflection LSM). Consider a one-dimensional lattice Hamiltonian with the symmetry group $G_{\text{tot}} \equiv Z_2^r \times \mathbb{Z}_2^x \times \mathbb{Z}_2^y$, where the subgroup $Z_2^r$ generates site-centered reflection and the subgroup $\mathbb{Z}_2^x \times \mathbb{Z}_2^y$ generates internal discrete spin-rotation symmetry. If each reflection center hosts a half-integer spin representation of $\mathbb{Z}_2^x \times \mathbb{Z}_2^y$, then the ground states cannot be simultaneously gapped, non-degenerate, and $G_{\text{tot}}$-symmetric.

**Definition 2** (Reflection LSM anomaly). When the reflection LSM Theorem 2 holds, we say that there is a reflection LSM anomaly.

*Remark* (LSM anomaly versus mixed 't Hooft anomaly). The translation LSM and reflection LSM Theorems (anomalies) have been interpreted as the presence of a mixed 't Hooft anomaly between crystalline symmetry groups, either $\mathbb{Z}_{2N}^t$ or $\mathbb{Z}_2^r$, and internal symmetry group $\mathbb{Z}_2^x \times \mathbb{Z}_2^y$ [15,17,23,28,39]. Accordingly, one cannot gauge the full internal symmetry group $G_{\text{int}}$, while maintaining the space group $G_{\text{spa}}$. However, a non-anomalous subgroup $H_{\text{int}} \subset G_{\text{int}}$ can still be consistently gauged.

In what follows, we will show that under the KW and JW dualities introduced in Secs. 2.2 and 2.3, respectively, the direct product structure of $G_{\text{tot}}$ is altered through a mixing of crystalline and internal symmetries. As both dualities correspond to gauging the non-anomalous diagonal subgroup $\mathbb{Z}_2^z \subset \mathbb{Z}_2^x \times \mathbb{Z}_2^y$, our main result can be interpreted as the incompatibility

between gauge-invariant representations of elements in the subgroup $(\mathbb{Z}_2^x \times \mathbb{Z}_2^y)/\mathbb{Z}_2^z$ and crystalline symmetries $\mathbb{Z}_{2N}^t$ and $\mathbb{Z}_2^r$ under the KW or JW dualities. We conjecture that an analogue of this result holds for general space groups $G_{spa}$ and internal symmetry groups $G_{int}$ if an LSM anomaly is present. In Sec. 5, we confirm that this conjecture is true for the generalization to $G_{int} = \mathbb{Z}_n \times \mathbb{Z}_n$ and $H_{int} = \mathbb{Z}_n$.

## 3.2 Kramers-Wannier dual of the LSM anomaly

We are going to construct the dual total symmetry group $G_{tot}^{\vee}$ under the KW duality introduced in Sec. 2.2. To this end, we define the action of the crystalline and internal symmetries on the extended Hilbert space $\mathcal{H}_{b,b'}$ defined in Eq. (2.8b) and then project these symmetries onto the dual Hilbert space $\mathcal{H}_{b'}^{\vee}$. For simplicity, we set $b' = 0$.

The extension of the crystalline symmetries (3.1) on the Hilbert space $\mathcal{H}_{b,b'}$ are obtained by demanding the covariance of the Gauss operators (2.9a) under translation and reflection. We thus define the unitary operators

$$\widehat{U}_t^{\text{ext}} := \left[ \prod_{j=1}^{2N-1} \frac{1}{2} \left( \widehat{\mathbb{1}}_{\mathcal{H}_{b=0,b'=0}} + \hat{\boldsymbol{\sigma}}_j \cdot \hat{\boldsymbol{\sigma}}_{t(j)} \right) \right] \left[ \prod_{j=1}^{2N-1} \frac{1}{2} \left( \widehat{\mathbb{1}}_{\mathcal{H}_{b=0,b'=0}} + \hat{\boldsymbol{\tau}}_{j^{\star}} \cdot \hat{\boldsymbol{\tau}}_{t(j^{\star})} \right) \right], \tag{3.7a}$$

$$\widehat{U}_r^{\text{ext}} := \left[ \prod_{j=1}^{N-1} \frac{1}{2} \left( \widehat{\mathbb{1}}_{\mathcal{H}_{b=0,b'=0}} + \hat{\boldsymbol{\sigma}}_j \cdot \hat{\boldsymbol{\sigma}}_{r(j)} \right) \right] \left[ \prod_{j=1}^{N} \frac{1}{2} \left( \widehat{\mathbb{1}}_{\mathcal{H}_{b=0,b'=0}} + \hat{\boldsymbol{\tau}}_{j^{\star}} \cdot \hat{\boldsymbol{\tau}}_{r(j^{\star})} \right) \right], \tag{3.7b}$$

that implement the transformation rules (3.1) for the $\hat{\boldsymbol{\sigma}}$ operators on lattice $\Lambda$, and the transformation rules

$$\widehat{U}_t^{\text{ext}} \, \hat{\boldsymbol{\tau}}_{j^{\star}} \left( \widehat{U}_t^{\text{ext}} \right)^{\dagger} = \hat{\boldsymbol{\tau}}_{t(j^{\star})}, \qquad t(j^{\star}) := j^{\star} + 1 \bmod 2N, \tag{3.7c}$$

$$\widehat{U}_r^{\text{ext}} \, \hat{\boldsymbol{\tau}}_{j^{\star}} \left( \widehat{U}_r^{\text{ext}} \right)^{\dagger} = \hat{\boldsymbol{\tau}}_{r(j^{\star})}, \qquad r(j^{\star}) := 2N - j^{\star} \bmod 2N, \tag{3.7d}$$

for the $\hat{\boldsymbol{\tau}}$ operators on the dual lattice $\Lambda^{\star}$. Transformation rules (3.7c) and (3.7d) correspond to two independent crystalline symmetries of the dual lattice $\Lambda^{\star}$, namely translation and (link-centered) reflection symmetries. As promised, the operators $\widehat{U}_t^{\text{ext}}$ and $\widehat{U}_r^{\text{ext}}$ are not gauge invariant but transform the local Gauss operators (2.9) according to the covariant rules

$$\widehat{U}_t^{\text{ext}} \, \widehat{G}_j \left( \widehat{U}_t^{\text{ext}} \right)^{\dagger} = \widehat{G}_{t(j)}, \tag{3.8a}$$

$$\widehat{U}_r^{\text{ext}} \, \widehat{G}_j \left( \widehat{U}_r^{\text{ext}} \right)^{\dagger} = \widehat{G}_{r(j)}, \tag{3.8b}$$

respectively, for any $j \in \Lambda$. After the projection to the dual Hilbert space $\mathcal{H}_{b'=0}^{\vee}$, the counterparts to the translation (3.7a) and reflection (3.7b) are implemented by the unitary operators

$$\widehat{U}_t^{\vee} := \prod_{j=1}^{2N-1} \frac{1}{2} \left( \widehat{\mathbb{1}}_{\mathcal{H}_{b'=0}^{\vee}} + \hat{\boldsymbol{\tau}}_{j^{\star}}^{\vee} \cdot \hat{\boldsymbol{\tau}}_{t(j^{\star})}^{\vee} \right), \tag{3.9a}$$

$$\widehat{U}_r^{\vee} := \prod_{j=1}^{N} \frac{1}{2} \left( \widehat{\mathbb{1}}_{\mathcal{H}_{b'=0}^{\vee}} + \hat{\boldsymbol{\tau}}_{j^{\star}}^{\vee} \cdot \hat{\boldsymbol{\tau}}_{r(j^{\star})}^{\vee} \right), \tag{3.9b}$$

respectively. The product on the right-hand side of Eq. (3.9b) has the upper bound $N$ since the reflection has no fixed points in $\Lambda^{\star}$. The pair of operators $\widehat{U}_t^{\vee}$ and $\widehat{U}_r^{\vee}$ generates a $2^{2N}$-dimensional representation of the space group $G_{spa}$ through the semi-direct product

$$G_{spa}^{\vee} := \mathbb{Z}_{2N}^t \rtimes \mathbb{Z}_2^r, \tag{3.10a}$$

with

$$\mathbb{Z}_{2N}^{t} \equiv \left\{ t, \ t^2, \ \cdots, \ t^{2N-1}, \ t^{2N} \equiv e \right\}, \qquad \mathbb{Z}_2^{r} \equiv \left\{ r, \ r^2 \equiv e \right\}. \tag{3.10b}$$

We note that the dual space group $G_{\text{spa}}^{\vee}$ is isomorphic to the space group $G_{\text{spa}}$ defined in Eq. (3.2). However, the action of the dual reflection symmetry (3.9b) differs from that of reflection symmetry (3.1b) in the sense that it acts as a link-centered reflection on the dual lattice $\Lambda^{\star}$ and does not admit any fixed points on $\Lambda^{\star}$.

The duals of the internal symmetries (3.3) are constructed by using the isomorphism between the bond algebras (2.4) and (2.16).[10] However, the corresponding local representations $\hat{\sigma}_j^x$ and $\hat{\sigma}_j^y$ do not belong to the bond algebra (2.4). In other words, they are not invariant under the global symmetry $\widehat{U}_{r_\pi^z}$. Therefore, when extending to the Hilbert space $\mathcal{H}_{b=0, b'=0}$, the operators $\widehat{U}_{r_\pi^x}$ and $\widehat{U}_{r_\pi^y}$ must be minimally coupled by the appropriate insertions of $\hat{\tau}_{j^{\star}}^z$ operators. We focus on the operator $\widehat{U}_{r_\pi^x}$ as the case of $\widehat{U}_{r_\pi^y}$ is treated analogously. We can extend the action of $\widehat{U}_{r_\pi^x}$ to the Hilbert space $\mathcal{H}_{b=0, b'=0}$ either according to the definition

$$\widehat{U}_{r_\pi^x}^{\text{ext}} := \prod_{j=1}^{N} \hat{\sigma}_{2j-1}^x \, \hat{\tau}_{(2j-1)^{\star}}^z \, \hat{\sigma}_{2j}^x, \tag{3.11a}$$

where $\hat{\tau}_{(2j)^{\star}-1}^z$ are inserted only on odd sites of the dual lattice $\Lambda^{\star}$, or according to the definition

$$\widehat{U}_{r_\pi^x}^{\text{ext}} := \prod_{j=1}^{N} \hat{\sigma}_{2j}^x \, \hat{\tau}_{(2j)^{\star}}^z \, \hat{\sigma}_{2j+1}^x, \tag{3.11b}$$

where $\hat{\tau}_{(2j)^{\star}-1}^z$ are inserted only on even sites of the dual lattice $\Lambda^{\star}$. Crucially, neither definition (3.11a) nor definition (3.11b) are invariant under translation (3.7a) or reflection (3.7b), i.e., the extended operator $\widehat{U}_{r_\pi^x}^{\text{ext}}$ is not invariant under the action of translation by one unit cell or by reflection. This incompatibility is rooted in the non-trivial local projective representation (3.6). Equivalently, this is a result of the two LSM Theorems 1 and 2 with translation and reflection symmetries, respectively.

By projecting onto the Hilbert space $\mathcal{H}_{b'=0}$ operators (3.11a) and (3.11b),[11] we identify the following dual internal symmetries

$$\widehat{U}_{\text{o}}^{\vee} = \prod_{j=1}^{N} \hat{\tau}_{2j-1+\frac{1}{2}}^{z\,\vee}, \qquad \widehat{U}_{\text{e}}^{\vee} = \prod_{j=1}^{N} \hat{\tau}_{2j+\frac{1}{2}}^{z\,\vee}, \qquad \widehat{U}_{r_\pi^z}^{\vee} = \widehat{U}_{\text{o}}^{\vee} \, \widehat{U}_{\text{e}}^{\vee} = \prod_{j=1}^{2N} \hat{\tau}_{j+\frac{1}{2}}^{z\,\vee}. \tag{3.12}$$

Note that the product of $\widehat{U}_{\text{o}}^{\vee}$ and $\widehat{U}_{\text{e}}^{\vee}$ delivers the dual symmetry of the bond algebra $\mathfrak{B}_{b'=0}$ defined in Eq. (2.17). The pair of operators $\widehat{U}_{\text{o}}^{\vee}$ and $\widehat{U}_{\text{e}}^{\vee}$ generates a $2^{2N}$-dimensional representation of the symmetry group $G_{\text{int}}$ through the direct product

$$G_{\text{int}}^{\vee} \equiv \mathbb{Z}_2^{\text{o}} \times \mathbb{Z}_2^{\text{e}}, \tag{3.13a}$$

with

$$\mathbb{Z}_2^{\text{o}} \equiv \left\{ r_{\text{o}}, \ (r_{\text{o}})^2 \equiv e \right\}, \qquad \mathbb{Z}_2^{\text{e}} \equiv \left\{ r_{\text{e}}, \ (r_{\text{e}})^2 \equiv e \right\}. \tag{3.13b}$$

---

[10]The fact that internal symmetries are tensor products over all sites of some local symmetry is crucial to validate the use of the dual bond algebra. For example, applying the isomorphism between the bond algebras (2.4) and (2.16) on the generators of the crystalline symmetries produces operators that are gauge invariant, i.e., they commute with the local Gauss operators. This is quite different from Eq. (3.8), according to which the local Gauss operators transform non-trivially but in a covariant manner under conjugation by $\widehat{U}_t^{\text{ext}}$ and $\widehat{U}_r^{\text{ext}}$.

[11]Operators $\widehat{U}_{\text{o}}^{\vee}$ and $\widehat{U}_{\text{e}}^{\vee}$ also follow from similarly dualizing $\widehat{U}_{r_\pi^y}$. Only $\hat{\tau}_j^{z\,\vee}$ enters in the products making up $\widehat{U}_{\text{e}}^{\vee}$ and $\widehat{U}_{\text{o}}^{\vee}$ in Eq. (3.12). Hence, these dual generators of the internal symmetries are not realized projectively locally.

Unlike the case in Sec. 3.1 with the space group $G_{\mathrm{spa}}$, the action of $G_{\mathrm{spa}}^{\vee}$ on $G_{\mathrm{int}}^{\vee}$ is now non-trivial as it is given by the composition rules

$$\widehat{U}_t^{\vee} \, \widehat{U}_{\mathrm{o}}^{\vee} \left(\widehat{U}_t^{\vee}\right)^{\dagger} = \widehat{U}_{\mathrm{e}}^{\vee}, \qquad \widehat{U}_t^{\vee} \, \widehat{U}_{\mathrm{e}}^{\vee} \left(\widehat{U}_t^{\vee}\right)^{\dagger} = \widehat{U}_{\mathrm{o}}^{\vee}, \tag{3.14a}$$

and

$$\widehat{U}_r^{\vee} \, \widehat{U}_{\mathrm{o}}^{\vee} \left(\widehat{U}_r^{\vee}\right)^{\dagger} = \widehat{U}_{\mathrm{e}}^{\vee}, \qquad \widehat{U}_r^{\vee} \, \widehat{U}_{\mathrm{e}}^{\vee} \left(\widehat{U}_r^{\vee}\right)^{\dagger} = \widehat{U}_{\mathrm{o}}^{\vee}. \tag{3.14b}$$

In other words, the dual symmetry group

$$G_{\mathrm{tot}}^{\vee} \equiv G_{\mathrm{spa}}^{\vee} \ltimes G_{\mathrm{int}}^{\vee}, \tag{3.15}$$

of the Hamiltonian $\widehat{H}_{b'=0}^{\vee}$ in the bond algebra (2.16) with $b' = 0$ that is dual to the Hamiltonian $\widehat{H}_{b=0}$ in the bond algebra (2.4) with $b = 0$ is a semi-direct product of crystalline symmetries $G_{\mathrm{spa}}^{\vee}$ and internal symmetries $G_{\mathrm{int}}^{\vee}$.

One observes that the two LSM Theorems 1 and 2 do not apply to the dual symmetry group $G_{\mathrm{tot}}^{\vee}$. This is because the local representation of $G_{\mathrm{int}}^{\vee}$ is not projective (see footnote 11) unlike that of $G_{\mathrm{int}}$. We further note that while being isomorphic to $G_{\mathrm{spa}}$ the dual crystalline symmetry group $G_{\mathrm{spa}}^{\vee}$ is such that

1. the "natural" unit cell on which the internal symmetry group $G_{\mathrm{int}}^{\vee}$ acts onsite is associated with the generator $\left(\widehat{U}_t^{\vee}\right)^2$ of translations, i.e., it is twice that of the unit cell associated with the generator $\widehat{U}_t^{\vee}$ of translations,[12]

2. the operator $\widehat{U}_r^{\vee}$ acts as a link-centered reflection on lattice $\Lambda^{\star}$ such that there are no invariant unit cells.

Both properties can be interpreted as a trivialization of mixed anomalies between internal and spatial symmetries under the gauging of a subgroup of the internal symmetries.

There is another useful reinterpretation of the dual internal symmetries whose generators are defined in Eq. (3.12). First, we have the identity

$$\widehat{U}_{r_{\pi}^z}^{\vee} =: e^{\mathrm{i}\pi \widehat{Q}^{\vee}}, \qquad \widehat{Q}^{\vee} := \sum_{j^{\star} \in \Lambda^{\star}} \widehat{Q}_{j^{\star}}^{\vee}, \qquad \left[\widehat{Q}_{i^{\star}}^{\vee}, \widehat{Q}_{j^{\star}}^{\vee}\right] = 0, \qquad \forall i^{\star}, j^{\star} \in \Lambda^{\star}, \tag{3.16}$$

where the local Hermitean operator $\widehat{Q}_{j^{\star}}^{\vee}$ has the $\mathbb{Z}_2$-valued local charge eigenvalue $q_{j^{\star}} = 0, 1$. Second, we have the identity

$$\widehat{U}_{\mathrm{o}}^{\vee} = (-\mathrm{i})^{2N} \, e^{\mathrm{i}\pi \widehat{D}^{\vee}} = \widehat{U}_t^{\vee} \, \widehat{U}_{\mathrm{e}}^{\vee} \left(\widehat{U}_t^{\vee}\right)^{\dagger}, \qquad \widehat{D}^{\vee} := \sum_{j^{\star} \in \Lambda^{\star}} \widehat{Q}_{j^{\star}}^{\vee} \, j^{\star}. \tag{3.17}$$

The symmetry generator $\widehat{U}_{\mathrm{o}}^{\vee}$ can thus be thought of as the exponential of the conserved global $\mathbb{Z}_2$-dipole operator $\widehat{D}^{\vee}$ associated to the conserved global $\mathbb{Z}_2$-charge operator $\widehat{Q}^{\vee}$. The punchline is now the following. Gauging the $\mathbb{Z}_2$ charge symmetry generated by $\widehat{U}_{r_{\pi}^z}^{\vee}$ induces a duality between Hamiltonians invariant under both $\mathbb{Z}_2$-dipole and translation (or link-centered reflection) symmetries.[13] that are free from LSM anomalies and Hamiltonians invariant under $\mathbb{Z}_2^x \times \mathbb{Z}_2^y$ internal and translation (or site-centered reflection) symmetries with LSM anomalies.

---

[12]The factor of two here is directly related to the fact that the local non-trivial projective representation (3.6) becomes a trivial representation on doubled unit cells.

[13]The presence of both translation (or link-centered reflection) and dipole symmetries imply the presence of a charge symmetry.

In other words, spatially modulated symmetries, such as a dipole symmetry, can be mapped to a global uniform symmetry at the cost of introducing an LSM anomaly.

In anticipation of the discussion of the phase diagram of the quantum spin-$1/2$ $XYZ$ chain in Sec. 4, we close this discussion by focusing on the reflection symmetry subgroup $\mathbb{Z}_2^r$ of $G_{\text{spa}}^\vee$. As a consequence of the underlying LSM anomaly, the Abelian group

$$\mathbb{Z}_2^r \times \mathbb{Z}_2^x \times \mathbb{Z}_2^y, \tag{3.18a}$$

formed by the subgroup of reflection symmetry $\mathbb{Z}_2^r$ together with the group of internal symmetries $G_{\text{int}} \equiv \mathbb{Z}_2^x \times \mathbb{Z}_2^y$ is mapped to the non-Abelian dihedral group of order eight

$$
\begin{aligned}
D_8 &:= \left\{ r,\, r^2 \equiv e \right\} \ltimes \left\{ e,\, r_{\text{o}},\, r_{\text{e}},\, r_{\text{o}} r_{\text{e}} \,\middle|\, \left(r_{\text{o}}\right)^2 \equiv \left(r_{\text{e}}\right)^2 \equiv e, \qquad r_{\text{o}} r_{\text{e}} = r_{\text{e}} r_{\text{o}} \right\} \\
&= \left\{ e,\, a,\, a^2,\, a^3,\, r,\, r\,a,\, r\,a^2,\, r\,a^3 \,\middle|\, a \equiv r\,r_{\text{o}}, \quad a^4 \equiv r^2 \equiv e, \quad r\,a\,r = a^3 \right\},
\end{aligned}
\tag{3.18b}
$$

after gauging the diagonal subgroup $\mathbb{Z}_2^z \subset \mathbb{Z}_2^x \times \mathbb{Z}_2^y$ by KW duality.

### 3.3 Jordan-Wigner dual of the LSM anomaly

We are going to construct the dual total symmetry group $G_{\text{tot}}^{\vee,\text{F}}$ under the JW duality introduced in Sec. 2.3. To this end, we define the action of the crystalline and internal symmetries on the extended Hilbert space $\mathcal{H}_{b,f}$ defined in Eq. (2.23b) and then project these symmetries onto the dual Hilbert space $\mathcal{H}_f^\vee$. We keep the boundary condition $f$ unspecified for the time being.

The extension of the crystalline symmetries (3.1) on the Hilbert space $\mathcal{H}_{b,f}$ are obtained by demanding the covariance of the Gauss operators (2.24a) under translation and reflection. We thus define the unitary operators

$$\widehat{U}_{t,f}^\vee := \left(\mathrm{i}\hat{\beta}_1\,\hat{\alpha}_1\right)^f \prod_{j=1}^{2N-1} \frac{\mathrm{i}}{2}\left[\left(\hat{\beta}_j^\vee - \hat{\beta}_{t(j)}^\vee\right)\left(\hat{\alpha}_j^\vee - \hat{\alpha}_{t(j)}^\vee\right)\right], \tag{3.19a}$$

$$\widehat{U}_{r,f}^\vee := \left(\mathrm{i}\hat{\beta}_{2N}\,\hat{\alpha}_{2N}\right)^f \prod_{j=1}^{2N} \frac{1}{\sqrt{2}}\left(\widehat{\mathbb{1}}_{\mathcal{H}_f^\vee} + \hat{\beta}_{r(j)}^\vee\,\hat{\alpha}_j^\vee\right), \tag{3.19b}$$

where the global fermion parity $\widehat{P}_{\text{F}}^\vee$ takes the form (2.31b). For any $j \in \Lambda$, conjugation of $\hat{\alpha}_j^\vee$ and $\hat{\beta}_j^\vee$ by $\widehat{U}_{t,f}^\vee$ and $\widehat{U}_{r,f}^\vee$ implement the maps

$$\hat{\alpha}_j^\vee \mapsto (-1)^{f\,\delta_{j,2N}}\,\hat{\alpha}_{t(j)}^\vee, \qquad \hat{\beta}_j^\vee \mapsto (-1)^{f\,\delta_{j,2N}}\,\hat{\beta}_{t(j)}^\vee, \tag{3.20a}$$

$$\hat{\alpha}_j^\vee \mapsto +(-1)^{f\,\delta_{j,2N}}\,\hat{\beta}_{r(j)}^\vee, \qquad \hat{\beta}_j^\vee \mapsto -(-1)^{f\,\delta_{j,2N}}\,\hat{\alpha}_{r(j)}^\vee, \tag{3.20b}$$

respectively.

We note that, unlike the dual spin operators $\hat{\tau}_{j^\star}$ defined on the dual lattice $\Lambda^\star$ in Sec. 3.2, the Majorana operators are defined on the direct lattice $\Lambda$. This is due to the fact that we applied an isomorphism implementing an additional half lattice translation in the process of JW duality [see Eq. (2.30)]. Due to this nuance, the reflection symmetry acts differently on the Majorana degrees of freedom than it did on the spins from Sec. 3.2, since none of the sites of $\Lambda^\star$ are invariant under reflection, while the sites $j = N,\, 2N \in \Lambda$ are left fixed under reflection. Furthermore, in the fermionic case, reflection is not an order two operation. Instead, one verifies that

$$\left(\widehat{U}_{r,f}^\vee\right)^2 = -\widehat{P}_{\text{F}}^\vee. \tag{3.21a}$$

Similarly, translation is not an order $2N$ operator if $f = 1$, instead

$$\left(\widehat{U}_{t,f}^{\vee}\right)^{2N} = \left(\widehat{P}_{\mathrm{F}}^{\vee}\right)^f . \qquad (3.21\mathrm{b})$$

This leads to a mixing of crystalline symmetries with the fermion parity. We denote the crystalline group obtained after JW duality as $\mathrm{G}_{\mathrm{spa}}^{\vee,\mathrm{F}}$. This group is obtained by the central extension of $\mathrm{G}_{\mathrm{spa}}^{\vee}$ defined in Eq. (3.10) by fermion parity $\mathbb{Z}_2^{\mathrm{F}}$ specified by the short exact sequence

$$0 \to \mathbb{Z}_2^{\mathrm{F}} \to \mathrm{G}_{\mathrm{spa}}^{\vee,\mathrm{F}} \to \mathrm{G}_{\mathrm{spa}}^{\vee} \to 0 , \qquad (3.22)$$

with the extension class $[\gamma_f] \in H^2(\mathrm{G}_{\mathrm{spa}}^{\vee}, \mathbb{Z}_2^{\mathrm{F}})$ and the extension map

$$\gamma_f(r,r) := p_{\mathrm{F}} , \qquad \gamma_f(t^a, t^b) = \left(p_{\mathrm{F}}\right)^{f \lfloor (a+b)/2N \rfloor} , \qquad \gamma_f(r,t) = \left(p_{\mathrm{F}}\right)^f , \qquad (3.23)$$

where $p_{\mathrm{F}}$ was defined in Eq. (2.22c) and $\lfloor \cdot \rfloor$ is the lower floor function. All other maps can be derived using these relations and the cocycle condition for $\gamma_f$. Having defined the crystalline symmetries, we now turn to the internal symmetries.

After the JW duality, the internal symmetry operators are obtained by dualizing $\widehat{U}_{r_\pi^x}$ and $\widehat{U}_{r_\pi^y}$ in Eq. (3.3). More precisely, under the JW duality.[14]

$$\widehat{U}_{r_\pi^x} = \prod_{j=1}^{N} \hat{\sigma}_{2j-1}^x \hat{\sigma}_{2j}^x \longmapsto \widehat{U}_{\mathrm{o}}^{\vee} := \prod_{j=1}^{N} \left( \mathrm{i} \hat{\alpha}_{2j-1}^{\vee} \hat{\beta}_{2j}^{\vee} \right) , \qquad (3.24\mathrm{a})$$

$$\widehat{U}_{r_\pi^y} = \prod_{j=1}^{N} \hat{\sigma}_{2j-1}^y \hat{\sigma}_{2j}^y \longmapsto \widehat{U}_{\mathrm{e}}^{\vee} := \prod_{j=1}^{N} \left( \mathrm{i} \hat{\beta}_{2j-1}^{\vee} \hat{\alpha}_{2j}^{\vee} \right) . \qquad (3.24\mathrm{b})$$

The pair $\widehat{U}_{\mathrm{o}}^{\vee}$ and $\widehat{U}_{\mathrm{e}}^{\vee}$ of dual internal symmetry operators compose to the fermion parity operator,

$$\widehat{U}_{\mathrm{o}}^{\vee} \widehat{U}_{\mathrm{e}}^{\vee} = \widehat{P}_{\mathrm{F}}^{\vee} . \qquad (3.24\mathrm{c})$$

The pair of operators $\widehat{U}_{\mathrm{o}}^{\vee}$ and $\widehat{U}_{\mathrm{e}}^{\vee}$ generates a $2^{2N}$-dimensional representation of the internal symmetry group

$$\mathrm{G}_{\mathrm{int}}^{\vee,\mathrm{F}} \equiv \mathbb{Z}_2^{\mathrm{o}} \times \mathbb{Z}_2^{\mathrm{e}} , \qquad (3.25\mathrm{a})$$

with

$$\mathbb{Z}_2^{\mathrm{o}} \equiv \left\{ r_{\mathrm{o}}, \, (r_{\mathrm{o}})^2 \equiv e \right\} , \qquad \mathbb{Z}_2^{\mathrm{e}} \equiv \left\{ r_{\mathrm{e}}, \, (r_{\mathrm{e}})^2 \equiv e \right\} . \qquad (3.25\mathrm{b})$$

The generators (3.19) of the dual crystalline symmetries act on the operators $\widehat{U}_{\mathrm{o}}^{\vee}$ and $\widehat{U}_{\mathrm{e}}^{\vee}$ according to the composition rules

$$
\begin{aligned}
\widehat{U}_t^{\vee} \widehat{U}_{\mathrm{o}}^{\vee} \left(\widehat{U}_t^{\vee}\right)^{\dagger} &= (-1)^{f+1} \widehat{U}_{\mathrm{e}}^{\vee} , & \widehat{U}_r^{\vee} \widehat{U}_{\mathrm{o}}^{\vee} \left(\widehat{U}_r^{\vee}\right)^{\dagger} &= (-1)^{f+1} \widehat{U}_{\mathrm{e}}^{\vee} , \\
\widehat{U}_t^{\vee} \widehat{U}_{\mathrm{e}}^{\vee} \left(\widehat{U}_t^{\vee}\right)^{\dagger} &= (-1)^{f+1} \widehat{U}_{\mathrm{o}}^{\vee} , & \widehat{U}_r^{\vee} \widehat{U}_{\mathrm{e}}^{\vee} \left(\widehat{U}_r^{\vee}\right)^{\dagger} &= (-1)^{f+1} \widehat{U}_{\mathrm{o}}^{\vee} , \\
\widehat{U}_t^{\vee} \widehat{P}_{\mathrm{F}}^{\vee} \left(\widehat{U}_t^{\vee}\right)^{\dagger} &= \widehat{P}_{\mathrm{F}}^{\vee} , & \widehat{U}_r^{\vee} \widehat{P}_{\mathrm{F}}^{\vee} \left(\widehat{U}_r^{\vee}\right)^{\dagger} &= \widehat{P}_{\mathrm{F}}^{\vee} .
\end{aligned} \qquad (3.26)
$$

The total symmetry group $\mathrm{G}_{\mathrm{tot}}^{\vee,\mathrm{F}}$ is obtained by taking the semi-direct product of $\mathrm{G}_{\mathrm{spa}}^{\vee,\mathrm{F}}$ and $\mathrm{G}_{\mathrm{int}}^{\vee,\mathrm{F}}$ together with coseting by the fermion parity group $\mathbb{Z}_2^{\mathrm{F}}$ defined in Eq. (2.22c), i.e.,

$$\mathrm{G}_{\mathrm{tot}}^{\vee,\mathrm{F}} = \left( \mathrm{G}_{\mathrm{spa}}^{\vee,\mathrm{F}} \ltimes \mathrm{G}_{\mathrm{int}}^{\vee,\mathrm{F}} \right) \Big/ \mathbb{Z}_2^{\mathrm{F}} . \qquad (3.27\mathrm{a})$$

---

[14]We obtain the operator $\widehat{U}_{\mathrm{o}}^{\vee}$ from dualizing $\widehat{U}_{r_\pi^x}$ and multiplying with $(-1)^N$. This multiplicative factor simplifies the algebra.

Here, the semi-direct product $G_{\mathrm{spa}}^{\vee,\mathrm{F}} \ltimes G_{\mathrm{int}}^{\vee,\mathrm{F}}$ is specified by the action

$$
\begin{aligned}
t\, r_{\mathrm{o}}\, t^{-1} &= r_{\mathrm{e}}, & r\, r_{\mathrm{o}}\, r^{-1} &= r_{\mathrm{e}}, \\
t\, r_{\mathrm{e}}\, t^{-1} &= r_{\mathrm{o}}, & r\, r_{\mathrm{e}}\, r^{-1} &= r_{0}, \\
t\, p_{\mathrm{F}}\, t^{-1} &= p_{\mathrm{F}}, & r\, p_{\mathrm{F}}\, r^{-1} &= p_{\mathrm{F}},
\end{aligned} \tag{3.27b}
$$

of dual crystalline symmetry group $G_{\mathrm{spa}}^{\vee,\mathrm{F}}$ on the dual internal symmetry group $G_{\mathrm{int}}^{\vee,\mathrm{F}}$. We empha-size that the structure of $G_{\mathrm{tot}}^{\vee,\mathrm{F}}$ is different from $G_{\mathrm{tot}}^{\vee}$ in Eq. (3.15) obtained via the KW duality. More precisely, under the JW duality the resulting dual total symmetry group $G_{\mathrm{tot}}^{\vee,\mathrm{F}}$ is assem-bled from the crystalline $G_{\mathrm{spa}}^{\vee,\mathrm{F}}$ and internal $G_{\mathrm{int}}^{\vee,\mathrm{F}}$ symmetry groups using a nontrivial central extension in addition to the semi-direct product structure. In contrast, the dual of $G_{\mathrm{tot}}$ under the KW duality described in Sec. 3.2 is a semi-direct product of the crystalline and internal symmetry groups. Having set $b = b' = 0$, in Secs. 3.1 and 3.2, triality of the bond algebras enforces $f = 1$ as prescribed in Eq. (2.54). Finally, we observe that the two LSM Theorems 1 and 2 do not apply to the dual symmetry group $G_{\mathrm{tot}}^{\vee,\mathrm{F}}$. This is because the local representation of $G_{\mathrm{int}}^{\vee,\mathrm{F}}$ is not projective, unlike that of $G_{\mathrm{int}}$. As was the case with the KW dual $G_{\mathrm{int}}^{\vee}$ generated by the operators in Eq. (3.12), the trivialization of mixed anomalies between internal and spatial symmetries under the JW gauging of a subgroup of the internal symmetries can be attributed to a doubling of the natural unit cell for the JW dual internal symmetries.

It is possible to reinterpret the dual symmetries that are defined in Eq. (3.24) as $\mathbb{Z}_2$-charge and $\mathbb{Z}_2$-dipole symmetries, respectively. To see this, we employ a "half"-translation

$$
\hat{\alpha}_j^{\vee} \mapsto \hat{\beta}_{j+1}^{\vee}, \qquad \hat{\beta}_j^{\vee} \mapsto \hat{\alpha}_j^{\vee}, \tag{3.28}
$$

after which the operators $\widehat{U}_{\mathrm{o}}^{\vee}$ and $\widehat{U}_{\mathrm{e}}^{\vee}$ act on only even and only odd sites, respectively. If so, the dual internal symmetries satisfy

$$
\widehat{P}_{\mathrm{F}}^{\vee} =: e^{\mathrm{i}\pi \widehat{Q}^{\vee}}, \qquad \widehat{Q}^{\vee} := \sum_{j^\star \in \Lambda^\star} \widehat{Q}_{j^\star}^{\vee}, \qquad \left[\widehat{Q}_{i^\star}^{\vee}, \widehat{Q}_{j^\star}^{\vee}\right] = 0, \tag{3.29a}
$$

$$
\widehat{U}_{\mathrm{o}}^{\vee} = e^{\mathrm{i}\pi \widehat{D}^{\vee}} = (-1)^{f+1}\, \widehat{U}_t^{\vee}\, \widehat{U}_{\mathrm{e}}^{\vee} \left(\widehat{U}_t^{\vee}\right)^{\dagger}, \qquad \widehat{D}^{\vee} := \sum_{j^\star \in \Lambda^\star} \widehat{Q}_{j^\star}^{\vee}\, j^\star, \tag{3.29b}
$$

where the local Hermitean operator $\widehat{Q}_{j^\star}^{\vee}$ has the $\mathbb{Z}_2$-valued local charge eigenvalue $q_{j^\star} = 0, 1$, for any $i^\star, j^\star \in \Lambda^\star$. As was in Sec. 3.2, the symmetry generator $\widehat{U}_{\mathrm{o}}^{\vee}$ can thus be thought of as a fermion dipole parity operator. As we have discussed in Sec. 2.4, there are two ways of gauging fermion parity symmetry. These result in either (i) Hamiltonians with $\mathbb{Z}_2^x \times \mathbb{Z}_2^y$ internal and translation (or site-centered reflection) symmetries with LSM anomalies, or (ii) $\mathbb{Z}_2$-charge, $\mathbb{Z}_2$-dipole, and translation (or link-centered reflection) symmetries. Notice that both dual Hamiltonians are bosonic, however, the former is invariant under global uniform symmetries with LSM anomalies while the latter is invariant under spatially modulated symmetries.

# 4 Triality and the phase diagram of the quantum spin-1/2 *XYZ* chain

We apply the triality derived in Secs. 2 and 3 to the study of the zero-temperature phase diagram of the quantum spin-1/2 *XYZ* chain. This model is dualized to a spin-1/2 cluster model and a model of interacting Majorana degrees of freedom under the KW and JW dualities, respectively. The triality allows to give three equivalent interpretations of the zero-temperature phase diagram with an emphasis on the symmetry structure of the bond algebras $\mathfrak{B}_b$, $\mathfrak{B}_{b'}$, and $\mathfrak{B}_f$, respectively.

## 4.1 Quantum spin-1/2 *XYZ* chain

Consider the quantum spin-1/2 antiferromagnetic chain with nearest- and next-nearest-neighbor antiferromagnetic couplings with periodic boundary conditions described by the Hamiltonian

$$
\begin{aligned}
\widehat{H}_{b=0} := & J_1 \sum_{j\in\Lambda}\left(\Delta_x\,\hat\sigma_j^x\,\hat\sigma_{j+1}^x + \Delta_y\,\hat\sigma_j^y\,\hat\sigma_{j+1}^y + \Delta_z\,\hat\sigma_j^z\,\hat\sigma_{j+1}^z\right) \\
& + J_2 \sum_{j\in\Lambda}\left(\Delta_x\,\hat\sigma_j^x\,\hat\sigma_{j+2}^x + \Delta_y\,\hat\sigma_j^y\,\hat\sigma_{j+2}^y + \Delta_z\,\hat\sigma_j^z\,\hat\sigma_{j+2}^z\right),
\end{aligned}
\tag{4.1}
$$

with the domain of definition $\mathcal{H}_{b=0}$ defined in Eq. (2.3). Both the dimensionful couplings $J_1$ and $J_2$ together with the dimensionless couplings $\Delta_x$, $\Delta_y$, and $\Delta_z$ are taken to be real-valued and non-negative. For convenience, we set the cardinality of the lattice to be

$$
|\Lambda| \equiv 2N = 0 \bmod 4,
\tag{4.2}
$$

i.e., $N$ is an even integer. The symmetries of $\widehat{H}_{b=0}$ that we shall keep track of are given in Sec. 3.1. We choose to work with periodic boundary conditions for the same reasons as in Sec. 3.1, i.e., for $b=0$, the total relevant symmetry group $G_{tot}$ is a direct product of a crystalline symmetry group $G_{spa}$ and of an internal symmetry group $G_{int}$. Accordingly, the spectrum of $\widehat{H}_{b=0}$ is constrained by LSM Theorems 1 and 2. Either one of Theorems 1 and 2 requires that any gapped phase in the parameter space of the model either breaks spontaneously the global internal symmetry $G_{int}$ or the crystalline symmetry $G_{spa}$, or it is infinitely degenerate in the thermodynamic limit [87–89, 142, 143]. The question that will be answered in Sec. 4 is that of the fate of Theorems 1 and 2 under the triality of Secs. 2 and 3. To this end, we shall reinterpret the zero-temperature phase diagram of $\widehat{H}_{b=0}$ after it has undergone a KW and JW dualization to the Hamiltonians $\widehat{H}_{b'}^{\vee}$ and $\widehat{H}_f^{\vee}$, respectively.

We define the ratios

$$
J := \frac{J_2}{J_1}, \qquad \Delta := \frac{\Delta_y}{\Delta_x},
\tag{4.3a}
$$

and consider, for simplicity, the reduced parameter space.[15]

$$
0 \le \Delta \le \infty, \qquad \Delta_z = 0, \qquad 0 \le J \le \infty.
\tag{4.3b}
$$

The energy eigenvalues and eigenvectors of Hamiltonian (4.1) are known in closed forms at the four corners

$$
(\Delta,J)=(0,0), \qquad (\Delta,J)=(\infty,0), \qquad (\Delta,J)=(0,\infty), \qquad (\Delta,J)=(\infty,\infty),
\tag{4.4a}
$$

and the pair of points [1]

$$
(\Delta,J)=(1,0), \qquad (\Delta,J)=(1,\infty).
\tag{4.4b}
$$

The gapped ground states are also known in closed form along the so-called Majumdar-Ghosh (MG) line [144–147]

$$
0 \le \Delta \le \infty, \qquad J = \frac{1}{2}.
\tag{4.4c}
$$

---

[15]The full coupling space, if expressed in terms of dimensionless couplings only, is $(\Delta_x, \Delta_y, \Delta_z, J) \in [0,\infty[\times[0,\infty[\times[0,\infty[\times[0,\infty[$. A detailed study of the corresponding zero-temperature phase diagram for $0 \le J \le 1/2$ can be found in Ref. [121]. Knowledge of the phase diagram for the cut $(\Delta_x, \Delta_y, 0, J) \in [0,\infty[\times[0,\infty[\times[0,\infty[$ is equivalent to knowledge of the phase diagram for the cut $(\Delta_x, 0, \Delta_z, J) \in [0,\infty[\times[0,\infty[\times[0,\infty[$ through the application of a unitary SU(2)-rotation about the $x$ axis in spin-1/2 space.

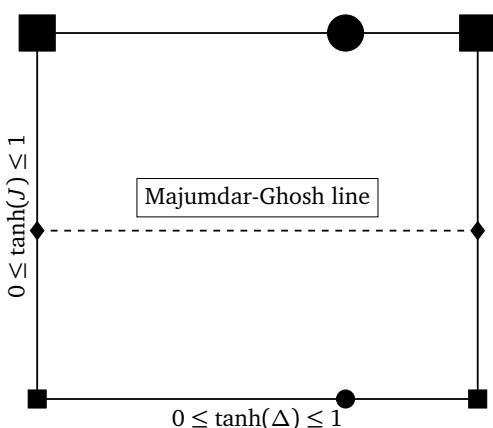

Figure 2: Exactly soluble points in the phase diagram of Hamiltonian (4.1) in the reduced coupling space (4.3). The small squares at the lower left and right corners of the phase diagram each realize the antiferromagnetic nearest-neighbor Ising chain. The large squares at the upper left and right corners of the phase diagram each realize two identical and decoupled antiferromagnetic nearest-neighbor Ising chains. The ground states are long-range ordered, gapped, and two-fold (four-fold) degenerate for the lower (upper) corners. The line $(\Delta, J = 0)$ realizes non-interacting fermions without fermion-number conservation, except when $\Delta = 1$. The small circle at $(\Delta = 1, J = 0)$ realizes non-interacting spinless fermions at half filling with a nearest-neighbor uniform hopping amplitude. The ground state is gapless with a quantum criticality encoded by a $c = 1$ conformal field theory in $(1 + 1)$-dimensional spacetime in the thermodynamic limit. The large circle at $(\Delta = 1, J = \infty)$ realizes two decoupled chains of non-interacting spinless fermions at half filling with a nearest-neighbor uniform hopping amplitude. The ground state is gapless with a quantum criticality behavior encoded by a $c = 2$ conformal field theory in $(1 + 1)$-dimensional spacetime in the thermodynamic limit. The diamonds at $(\Delta, J) = (0, 1/2), (\infty, 1/2)$ are first-order boundaries between the phases governed by the Ising fixed points (small and large squares) at $\Delta = 0$ and $\Delta = \infty$, respectively. The open Majumdar-Ghosh (dashed) line at $(\Delta, 1/2)$ with $0 < \Delta < \infty$ realizes the dimer phase. The dimer ground states are gapped and two-fold degenerate along the Majumdar-Ghosh (dashed) line.

The nature of the ground states of the Hamiltonian (4.1) at all these points in the reduced coupling space (4.3) is summarized in Fig. 2. The ground states at the four corners (4.4a) and along the open MG line $0 < \Delta < \infty$, $J = 1/2$ are gapped and degenerate. Even though the exact degeneracies for any finite cardinality $2N = |\Lambda|$ are lifted by small perturbations away from the four corners or away from the open MG line, these degeneracies are restored in the thermodynamic limit $2N \to \infty$.

Below the MG line (4.4c), there are three gapped phases [121, 148], each of which spontaneously breaks $G_{tot}$ in the thermodynamic limit through the spontaneous selection of a ground state from two-fold degenerate ground states. The Neel$_x$ phase is adiabatically connected to the fixed-point limit at the lower left corner $(\Delta, J) = (0, 0)$ and spontaneously breaks translation symmetry by one lattice spacing and the rotation symmetry about the $y$-axis in spin-1/2 space. Similarly, the Neel$_y$ phase is adiabatically connected to the fixed-point limit at the lower right corner $(\Delta, J) = (\infty, 0)$ and spontaneously breaks translation symmetry and rotation symmetry about the $x$-axis in spin-1/2 space. The dimer phase is adiabatically connected

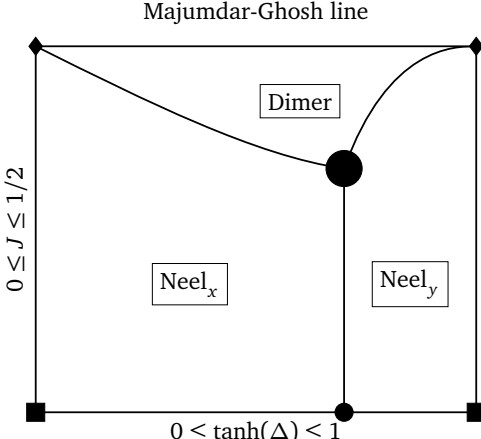

Figure 3: Phase diagram of Hamiltonian (4.1) in the reduced coupling space (4.3) with $0 \leq J \leq 1/2$. There are three phases: the Neel$_x$, the Neel$_y$, and the dimer phase. Each one of these three phases corresponds to gapped and two-fold degenerate ground states in the thermodynamic limit. In each phase, a non-degenerate ground state is selected by spontaneous symmetry breaking of the symmetry group G$_{tot}$ defined in Eq. (3.5). The dimer phase is found on both sides of the open MG line defined by $0 < \Delta < \infty$ and $J = 1/2$. All the phase boundaries with $0 < \Delta < \infty$ and $J < 1/2$ are continuous quantum phase transitions that realize deconfined quantum criticality [121]. The tricritical point (the large black circle) where the three phases meet realizes the SU(2)$_1$ conformal field theory in $(1+1)$-dimensional spacetime.

to the MG line and spontaneously breaks the symmetries under translation by one lattice spacing and the site-centered reflection, while preserving the internal symmetries. This pattern of spontaneous symmetry breaking precludes a continuous phase transition governed by the reduction of a symmetry in one direction across the transition between any two of these three phases that would follow the Landau-Ginzburg paradigm of phase transitions. Nevertheless, the boundaries between any two of these three gapped phases when $0 < \Delta < \infty$ realize continuous quantum phase transitions. In fact, they are examples of deconfined quantum critical transitions [121, 149]. A deconfined quantum critical transition is driven by the deconfinement of point defects in one phase that nucleate locally the local order of the phase on the other side of the transition [114–117]. The two end points

$$(\Delta, J) = (0, 1/2), \qquad (\Delta, J) = (\infty, 1/2), \tag{4.4d}$$

of the MG line are gapped with a degeneracy proportional to the cardinality $2N = |\Lambda|$. Each becomes the phase boundary in the antiferromagnetic Ising chain with competing nearest- and next-nearest-neighbor interactions at which a first-order phase transition takes place in the thermodynamic limit [87–89, 142, 143]. The phase diagram of the Hamiltonian (4.1) has been studied by numerical means both in the reduced coupling space (4.3) [150] as well as without the restriction $\Delta_z = 0$ [121, 151–154]. Below the MG line (4.4c), the phase diagram deduced from numerical and analytical arguments is given in Fig. 3.

Since all three phases below the MG line break translation by one lattice spacing, they can be distinguished by order parameters that break the symmetries in the subgroup

$$\mathbb{Z}_2^r \times \mathbb{Z}_2^x \times \mathbb{Z}_2^y \subset G_{tot}, \tag{4.5}$$

defined in Eq. (3.5). In what follows, we will limit the discussion to this subgroup for simplicity. We will discuss the duals of the ground states of each gapped phase and the duals of those

operators defined in Eq. (4.8), whose expectations values detect the long-range orders that distinguish the gapped phases. At the two corners $(\Delta, J) = (0,0)$ and $(\Delta, J) = (\infty, 0)$ and along the MG line $(\Delta, J) = (\Delta, 1/2)$, the two-fold degenerate ground states are as follows.

1. At the lower left corner $(\Delta, J) = (0,0)$, the two degenerate ground states are

$$|\mathrm{Neel}_\mathrm{o}^x\rangle := |\rightarrow, \leftarrow, \rightarrow, \leftarrow, \cdots\rangle, \qquad |\mathrm{Neel}_\mathrm{e}^x\rangle := |\leftarrow, \rightarrow, \leftarrow, \rightarrow, \cdots\rangle, \tag{4.6a}$$

   where the kets $|\rightarrow\rangle_j$ and $|\leftarrow\rangle_j$ denote the eigenstates of $\hat{\sigma}_j^x$ with eigenvalues $+1$ and $-1$, respectively.

2. At the lower right corner $(\Delta, J) = (\infty, 0)$, the two degenerate ground states are

$$|\mathrm{Neel}_\mathrm{o}^y\rangle := |\nearrow, \swarrow, \nearrow, \swarrow, \cdots\rangle, \qquad |\mathrm{Neel}_\mathrm{e}^y\rangle := |\swarrow, \nearrow, \swarrow, \nearrow, \cdots\rangle, \tag{4.6b}$$

   where the kets $|\nearrow\rangle_j$ and $|\swarrow\rangle_j$ denote the eigenstates of $\hat{\sigma}_j^y$ with eigenvalues $+1$ and $-1$, respectively.

3. Along the MG line $(\Delta, J) = (\Delta, 1/2)$, the two degenerate ground states are

$$|\mathrm{Dimer}_\mathrm{o}\rangle := \bigotimes_{j=1}^{N} |[2j-1, 2j]\rangle, \qquad |\mathrm{Dimer}_\mathrm{e}\rangle := \bigotimes_{j=1}^{N} |[2j, 2j+1]\rangle, \tag{4.6c}$$

   where $|[j, j+1]\rangle$ denotes the singlet state for two spins localized on consecutive sites $j$ and $j+1$.

These ground states are distinguished by the non-vanishing expectations values of the order parameters

$$\widehat{O}_{\mathrm{Neel}^x}^\mathrm{o} := \frac{1}{2N} \sum_{j=1}^{2N} (-1)^{j+1} \hat{\sigma}_j^x,$$

$$\widehat{O}_{\mathrm{Neel}^y}^\mathrm{o} := \frac{1}{2N} \sum_{j=1}^{2N} (-1)^{j+1} \hat{\sigma}_j^y, \tag{4.7}$$

$$\widehat{O}_{\mathrm{dimer}} := \frac{1}{N} \sum_{j=1}^{2N} (-1)^j \frac{1}{3} \hat{\boldsymbol{\sigma}}_j \cdot \hat{\boldsymbol{\sigma}}_{j+1},$$

respectively. The order parameters for the $\mathrm{Neel}_x$ and $\mathrm{Neel}_y$ phases are odd under $\widehat{U}_{r_\pi^z}$ symmetry, while the dimer order parameter is even. In other words, the order parameter for the two Neel phases do not belong to the bond algebra (2.4) and do not have an image in the dual bond algebras (2.16) and (2.31). For this reason, it is more convenient to define the operators

$$\widehat{C}_{j,j+n}^x := \hat{\sigma}_j^x \hat{\sigma}_{j+n}^x, \tag{4.8a}$$

$$\widehat{C}_{j,j+n}^y := \hat{\sigma}_j^y \hat{\sigma}_{j+n}^y, \tag{4.8b}$$

$$\widehat{D}_j := \frac{1}{3} \hat{\boldsymbol{\sigma}}_j \cdot \hat{\boldsymbol{\sigma}}_{j+1}, \tag{4.8c}$$

for any $j \in \Lambda$ and any $n = 1, \cdots, |\Lambda| - 1$, all of which are even under $\widehat{U}_{r_\pi^z}$ symmetry. The first two are bilocal operators, whose expectation values are the two-point correlation functions detecting the magnetic ordering in $x$- and $y$-directions. The last one is the local operator, whose staggered summation over the lattice is the order parameter of the dimer phase. The expectation values of the order parameters (4.7) and operators (4.8) in the ground states (4.6) are given in Table 5.

Table 5: The expectation values of the order parameters (4.7) and operators (4.8) in the ground states (4.6) of Hamiltonian (4.1). The states $|\text{Neel}^x\rangle^+$ and $|\text{Neel}^y\rangle^+$ are defined in Eqs. (4.16) and (4.17), respectively.

| | $\widehat{O}^o_{\text{Neel}^x}$ | $\widehat{O}^o_{\text{Neel}^y}$ | $\widehat{O}_{\text{dimer}}$ | $\widehat{C}^x_{j,j+n}$ | $\widehat{C}^y_{j,j+n}$ | $\widehat{D}_j$ |
|---|---|---|---|---|---|---|
| $|\text{Neel}^x_o\rangle$ | $+1$ | $0$ | $0$ | $(-1)^n$ | $0$ | $-\frac{1}{3}$ |
| $|\text{Neel}^x_e\rangle$ | $-1$ | $0$ | $0$ | $(-1)^n$ | $0$ | $-\frac{1}{3}$ |
| $|\text{Neel}^x\rangle^+$ | $0$ | $0$ | $0$ | $(-1)^n$ | $0$ | $-\frac{1}{3}$ |
| $|\text{Neel}^y_o\rangle$ | $0$ | $+1$ | $0$ | $0$ | $(-1)^n$ | $-\frac{1}{3}$ |
| $|\text{Neel}^y_e\rangle$ | $0$ | $-1$ | $0$ | $0$ | $(-1)^n$ | $-\frac{1}{3}$ |
| $|\text{Neel}^y\rangle^+$ | $0$ | $0$ | $0$ | $0$ | $(-1)^n$ | $-\frac{1}{3}$ |
| $|\text{Dimer}_o\rangle$ | $0$ | $0$ | $+1$ | $-\delta_{(-1)^j,-1}\,\delta_{n,1}$ | $-\delta_{(-1)^j,-1}\,\delta_{n,1}$ | $-\delta_{(-1)^j,-1}$ |
| $|\text{Dimer}_e\rangle$ | $0$ | $0$ | $-1$ | $-\delta_{(-1)^j,+1}\,\delta_{n,1}$ | $-\delta_{(-1)^j,+1}\,\delta_{n,1}$ | $-\delta_{(-1)^j,+1}$ |

## 4.2 Kramers-Wannier dual $D_8$-symmetric spin-1/2 cluster chain

We now study the Hamiltonian dual to the Hamiltonian (4.1) under the KW duality. As in Sec. 3.2, we select periodic boundary conditions ($b' = 0$) after the KW duality. Naive use of the dual bond algebra (2.16) delivers the Hamiltonian

$$
\begin{aligned}
\widehat{H}^\vee_{b'=0} :=& J_1 \sum_{j^\star \in \Lambda^\star} \left[ \Delta_x\, \widehat{\tau}^{z\,\vee}_{j^\star} - \Delta_y \left( \widehat{\tau}^{x\,\vee}_{j^\star-1} \widehat{\tau}^{z\,\vee}_{j^\star} \widehat{\tau}^{x\,\vee}_{j^\star+1} \right) + \Delta_z \left( \widehat{\tau}^{x\,\vee}_{j^\star-1} \widehat{\tau}^{x\,\vee}_{j^\star+1} \right) \right] \\
&+ J_2 \sum_{j^\star \in \Lambda^\star} \left[ \Delta_x\, \widehat{\tau}^{z\,\vee}_{j^\star} \widehat{\tau}^{z\,\vee}_{j^\star+1} + \Delta_y \left( \widehat{\tau}^{x\,\vee}_{j^\star-1} \widehat{\tau}^{z\,\vee}_{j^\star} \widehat{\tau}^{x\,\vee}_{j^\star+1} \right) \left( \widehat{\tau}^{x\,\vee}_{j^\star} \widehat{\tau}^{z\,\vee}_{j^\star+1} \widehat{\tau}^{x\,\vee}_{j^\star+2} \right) \right. \\
&\left. + \Delta_z \left( \widehat{\tau}^{x\,\vee}_{j^\star-1} \widehat{\tau}^{x\,\vee}_{j^\star+1} \right) \left( \widehat{\tau}^{x\,\vee}_{j^\star} \widehat{\tau}^{x\,\vee}_{j^\star+2} \right) \right],
\end{aligned} \tag{4.9}
$$

with the domain of definition $\mathcal{H}^\vee_{b'=0}$ defined in Eq. (2.14b). However, Hamiltonians (4.1) and (4.9) only form a dual pair if their domains of definition are restricted to the subspaces $\mathcal{H}_{b=0;+}$ and $\mathcal{H}^\vee_{b'=0;+}$, respectively [recall Eq. (2.21)]. With this in mind, we will first study the phase diagram of Hamiltonian (4.9) in the full Hilbert space $\mathcal{H}^\vee_{b'=0}$. We will then discuss the duality of phases in the restricted Hilbert spaces $\mathcal{H}_{b=0;+}$ and $\mathcal{H}^\vee_{b'=0;+}$. Without loss of generality, we consider only the reduced coupling space (4.3) with $J \leq 1/2$.

The symmetries of $\widehat{H}^\vee_{b'=0}$ that we shall keep track of are given in Sec. 3.2. Because the global internal symmetry subgroup $G^\vee_{\text{int}} \subset G^\vee_{\text{tot}}$ is represented by a trivial projective representation locally, the LSM Theorems 1 and 2 are inoperative. Hence, $\widehat{H}^\vee_{b'=0}$ could exhibit a non-degenerate gapped ground state in its phase diagram, a possibility that is indeed realized. We restrict ourselves to the dual of subgroup (4.5), which is the dihedral group $D_8$ defined in Eq. (3.18b).

By inspection, the energy eigenvalues and eigenvectors of Hamiltonian (4.9) are known in closed form at the four corners (4.4a). Along the left boundary $\Delta = 0$ of the reduced coupling space (4.3), $\widehat{H}^\vee_{b'=0}$ simplifies to the classical Ising model in a uniform longitudinal magnetic field. The same is true of the right boundary $\Delta = \infty$, as the right boundary is unitarily equivalent to the left boundary [140].[16] When $J = 0$, the Hamiltonian (4.9) is a linear combination of two of the spin-1/2 cluster Hamiltonians that were introduced by Suzuki in 1971 [155], each of which is soluble in the sense that it is a sum of pairwise commuting local

---

[16] It is precisely this duality that was used in Refs. [87–89] to solve the antiferromagnetic Ising open chain with nearest- and next-nearest-neighbor couplings.

Hermitian operators that all square to the identity.[17] At the lower left corner $(\Delta, J) = (0,0)$, the ground state is the trivial paramagnet

$$|\text{PM}\rangle := |\downarrow, \cdots, \downarrow\rangle, \qquad \hat{\tau}_{j^\star}^{z\vee} |\downarrow, \cdots, \downarrow\rangle = -|\downarrow, \cdots, \downarrow\rangle, \qquad j^\star \in \Lambda^\star, \qquad (4.11)$$

which is a singlet under the $D_8$ symmetry. The lower right corner $(\Delta, J) = (\infty, 0)$ also corresponds to a non-degenerate, gapped, and $D_8$-symmetric ground state $|\text{SPT}\rangle$ that is defined implicitly by the eigenvalue equation

$$\hat{\tau}_{j^\star-1}^{x\vee} \hat{\tau}_{j^\star}^{z\vee} \hat{\tau}_{j^\star+1}^{x\vee} |\text{SPT}\rangle = +|\text{SPT}\rangle, \qquad j^\star \in \Lambda^\star. \qquad (4.12)$$

The ground state $|\text{SPT}\rangle$ defines a symmetry-protected topological (SPT) phase on a closed space manifold (owing to the periodic boundary conditions). This SPT phase is protected by the global internal symmetry $\mathbb{Z}_2^o \times \mathbb{Z}_2^e$ in the sense that it cannot be adiabatically deformed to the trivial paramagnetic state $|\text{PM}\rangle$ without a gap-closing phase transition or the breaking (spontaneous or explicit) of the $\mathbb{Z}_2^o \times \mathbb{Z}_2^e$ symmetry.

We emphasize that the correct KW dualization of the Hamiltonian (4.1) under open boundary conditions is not the Hamiltonian (4.9) under open boundary conditions. With open boundary conditions, one must modify the definition of the local Gauss operators at the two ends of the chain when gauging the theory. This change is responsible for the presence of additional terms that break the protecting $\mathbb{Z}_2^o \times \mathbb{Z}_2^e$ symmetry at the boundaries. These additional terms lift the two-fold degeneracy of the SPT ground state of the counterpart to Hamiltonian (4.9) corresponding to open boundary conditions. The KW dualization with open boundaries is explained in Appendix B.

The dual ground states can also be obtained in closed analytical form along the $J = 1/2$ line in the parameter space of the Hamiltonian (4.9). This is done by dualizing the projectors onto the MG ground states of the Hamiltonian (4.1) along the MG line (4.4c), as is detailed in Appendix A. One finds that the ground states of Hamiltonian $\widehat{H}_{b'=0}^{\vee}$ are gapped and four-fold degenerate along the open MG line. This is confirmed by performing an exact diagonalization study of the eigenvalue spectrum of $\widehat{H}_{b'=0}^{\vee}$. These four ground states are (see Appendix A)

$$|1\rangle = |\downarrow, \rightarrow, \downarrow, \leftarrow, \downarrow, \rightarrow, \downarrow, \leftarrow, \cdots\rangle, \qquad (4.13a)$$
$$|2\rangle = |\downarrow, \leftarrow, \downarrow, \rightarrow, \downarrow, \leftarrow, \downarrow, \rightarrow, \cdots\rangle, \qquad (4.13b)$$
$$|3\rangle = |\rightarrow, \downarrow, \leftarrow, \downarrow, \rightarrow, \downarrow, \leftarrow, \downarrow, \cdots\rangle, \qquad (4.13c)$$
$$|4\rangle = |\leftarrow, \downarrow, \rightarrow, \downarrow, \leftarrow, \downarrow, \rightarrow, \downarrow, \cdots\rangle, \qquad (4.13d)$$

where we chose the basis for which $|\rightarrow\rangle_j$ ($|\uparrow\rangle_j$) is the eigenstate with eigenvalue $+1$ of $\hat{\tau}_{j^\star}^{x\vee}$ ($\hat{\tau}_{j^\star}^{z\vee}$). These four-fold degenerate ground states spontaneously break the dihedral group $D_8$ down to a $\mathbb{Z}_2$ subgroup since

$$\widehat{U}_e^{\vee} \widehat{U}_{r_\pi^z}^{\vee} |1\rangle = |2\rangle, \qquad \widehat{U}_r^{\vee} |2\rangle = |3\rangle, \qquad \widehat{U}_{r_\pi^z}^{\vee} |3\rangle = \widehat{U}_r^{\vee} |1\rangle = |4\rangle, \qquad (4.14a)$$
$$\widehat{U}_o^{\vee} |1\rangle = |1\rangle, \qquad \widehat{U}_o^{\vee} |2\rangle = |2\rangle, \qquad \widehat{U}_e^{\vee} |3\rangle = |3\rangle, \qquad \widehat{U}_e^{\vee} |4\rangle = |4\rangle, \qquad (4.14b)$$

i.e., the states $|1\rangle$ and $|2\rangle$ are invariant only under the $\mathbb{Z}_2$ subgroup generated by $\widehat{U}_o^{\vee}$ while the states $|3\rangle$ and $|4\rangle$ are invariant only under the $\mathbb{Z}_2$ subgroup generated by $\widehat{U}_e^{\vee}$. The phase diagram of Hamiltonian (4.9) on the full Hilbert space $\mathcal{H}_{b'=0}^{\vee}$ is shown in Fig. 4. The phase boundaries in Fig. 3 carry over to Fig. 4 owing to the duality. As opposed to the deconfined

---

[17]Similarly, the upper corners

$$(\Delta, J) = (0, \infty), \qquad (\Delta, J) = (\infty, \infty), \qquad (4.10)$$

in the reduced coupling space (4.3) are exactly solvable and are gapped with two-fold degenerate ground states.

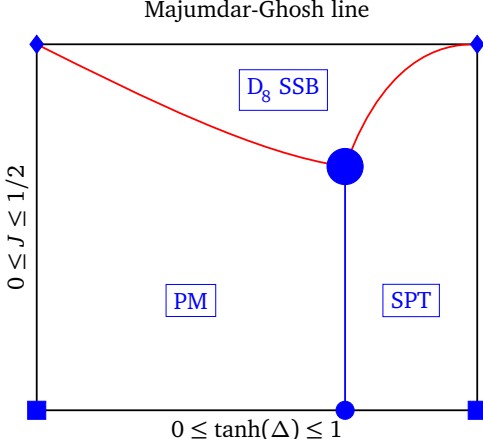

Figure 4: Phase diagram of Hamiltonian (4.9) with the Hilbert space $\mathcal{H}^{\vee}_{b'=0}$ as domain of definition. The red boundaries realize a continuous quantum phase transition that separate two phases, one of which descends from the other through spontaneous symmetry breaking by which a symmetry-breaking local order parameter acquires a non-vanishing expectation value in the symmetry-broken phase, i.e., the Landau-Ginzburg paradigm of phase transitions. The blue boundary realizes a continuous topological quantum phase transition between two phases that are distinguished by a non-local order parameter. These phases are adiabatically connected to the ground states (4.11) and (4.12) for $\Delta < 1$ and $\Delta > 1$, respectively.

quantum critical lines in Fig. 3, the phase diagram in Fig. 4 features (i) a topological transition (blue line) between the two $D_8$-singlet states that are adiabatically connected to states (4.11) and (4.12), respectively, and (ii) two conventional symmetry breaking transitions (red lines) between a doublet of states that break completely the symmetry group $D_8$ and the $D_8$-singlet states that are adiabatically connected to states (4.11) and (4.12), respectively.

The KW duality implies that the expectation value of any operator from the bond algebra (2.4) restricted to the Hilbert space $\mathcal{H}_{b=0;+}$ has the same expectation value as its dual in the bond algebra (2.16) restricted to the Hilbert space $\mathcal{H}^{\vee}_{b'=0;+}$. Under the isomorphism between the bond algebras (2.4) and (2.16), operators (4.8) dualize to

$$\widehat{C}^{x\,\vee}_{j^\star,j^\star+n-1} := \prod_{\ell=j^\star}^{j^\star+n-1} \hat{\tau}^{z\,\vee}_{\ell^\star}, \tag{4.15a}$$

$$\widehat{C}^{y\,\vee}_{j^\star,j^\star+n-1} := \hat{\tau}^{x\,\vee}_{j^\star-1}\,\hat{\tau}^{x\,\vee}_{j^\star}\left(\prod_{\ell=j^\star}^{j^\star+n-1} \hat{\tau}^{z\,\vee}_{\ell^\star}\right)\hat{\tau}^{x\,\vee}_{j^\star+n-1}\,\hat{\tau}^{x\,\vee}_{j^\star+n}, \tag{4.15b}$$

$$\widehat{D}^{\vee}_{j^\star} := \frac{1}{3}\left(\hat{\tau}^{z\,\vee}_{j^\star} - \hat{\tau}^{x\,\vee}_{j^\star-1}\,\hat{\tau}^{z\,\vee}_{j^\star}\,\hat{\tau}^{x\,\vee}_{j^\star+1} + \hat{\tau}^{x\,\vee}_{j^\star-1}\,\hat{\tau}^{x\,\vee}_{j^\star+1}\right), \tag{4.15c}$$

for any $j^\star \in \Lambda^\star$ and any $n = 1, \cdots, |\Lambda|-1$. We observe that operators $\widehat{C}^{x}_{j,j+n}$ and $\widehat{C}^{y}_{j,j+n}$ defined in Eqs. (4.8a) and (4.8b), respectively, dualize to non-local string operators, while the local operator $\widehat{D}_j$ defined in Eq. (4.8c) remains local after dualization.

At the lower left corner $(\Delta, J) = (0,0)$ of the phase diagram, only the bonding linear combination of the two Neel states

$$|\mathrm{Neel}^x\rangle^+ := \frac{1}{\sqrt{2}}\left(|\mathrm{Neel}^x_\mathrm{o}\rangle + |\mathrm{Neel}^x_\mathrm{e}\rangle\right), \qquad \widehat{U}_{r^z_\pi}|\mathrm{Neel}^x\rangle^+ = +|\mathrm{Neel}^x\rangle^+, \tag{4.16}$$

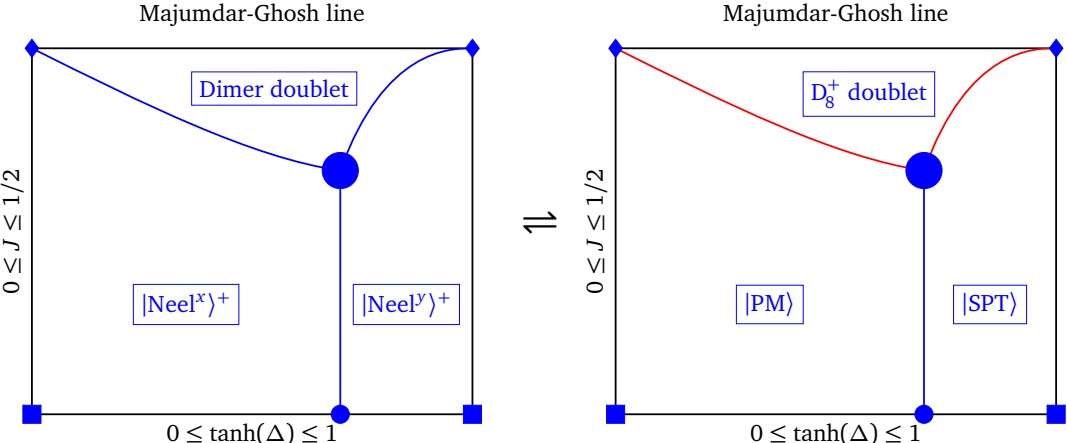

Figure 5: The phase diagram of Hamiltonian (4.1) restricted to the subspace $\mathcal{H}_{b=0;+}$ of the Hilbert space $\mathcal{H}_{b=0}$ is dual to the phase diagram of Hamiltonian (4.9) restricted to the subspace $\mathcal{H}^{\vee}_{b'=0;+}$ of the Hilbert space $\mathcal{H}^{\vee}_{b'=0}$. The pair of dual subspaces are to be found in the first line of Table 1. The two states $|\text{Neel}^x\rangle^+$ and $|\text{Neel}^y\rangle^+$ are defined in Eqs. (4.16) and (4.17), while dimer doublet refers to the ground states (4.6c). On the dual side, the paramagnetic states $|\text{PM}\rangle$, and $|\text{SPT}\rangle$ state are defined in Eqs. (4.11) and (4.12), respectively. By $D_8^+$ doublet, we refer to the states $|\text{Dimer}^{\vee}_{\text{o}}\rangle$ and $|\text{Dimer}^{\vee}_{\text{e}}\rangle$, both of which are in the subspace $\mathcal{H}^{\vee}_{b'=0;+}$ and defined in Eq. (4.18). The symbols $\rightharpoonup$ and $\leftharpoonup$ denote gauging the diagonal subgroups generated by $\widehat{U}_{r^z_{\pi}}$ and by its dual $\widehat{U}^{\vee}_{r^z_{\pi}}$, respectively.

belongs to the subspace $\mathcal{H}_{b=0;+}$. Under the KW duality, this state is mapped to the paramagnetic ground state $|\text{PM}\rangle \in \mathcal{H}^{\vee}_{b'=0;+}$ defined in Eq. (4.11). The expectation values of the dual operators (4.15) in the ground state $|\text{PM}\rangle$ are given in Table 6. The non-vanishing expectation value of the bilocal operator $\widehat{C}^x_{j,j+n}$ translates to the non-vanishing expectation value of the string operator $\widehat{C}^{x\,\vee}_{j^\star,j^\star+n-1}$ for any $j^\star$ and $n$. This is the so-called disorder operator, whose non-vanishing expectation value detects the disordered paramagnetic phase [93].

At the lower right corner $(\Delta, J) = (\infty, 0)$, only the bonding linear combination of the two Neel states

$$|\text{Neel}^y\rangle^+ := \frac{1}{\sqrt{2}}\left(|\text{Neel}^y_{\text{o}}\rangle + |\text{Neel}^y_{\text{e}}\rangle\right), \qquad \widehat{U}_{r^z_{\pi}}|\text{Neel}^y\rangle^+ = +|\text{Neel}^y\rangle^+, \qquad (4.17)$$

belongs to the subspace $\mathcal{H}_{b=0;+}$. Under the KW duality, this state is mapped to the SPT ground state with periodic boundary conditions $|\text{SPT}\rangle \in \mathcal{H}^{\vee}_{b'=0;+}$ defined in Eq. (4.12). The expectation values of the dual operators (4.15) in the ground state $|\text{SPT}\rangle$ are given in Table 6. The non-vanishing expectation value of the bilocal operator $\widehat{C}^y_{j,j+n}$ translates to the non-vanishing expectation value of the string operator $\widehat{C}^{y\,\vee}_{j^\star,j^\star+n-1}$ operator for any $j^\star$ and $n$. The string operator $\widehat{C}^{y\,\vee}_{j^\star,j^\star+n-1}$ is invariant under the dual internal symmetries $\widehat{U}^{\vee}_{\text{o}}$ and $\widehat{U}^{\vee}_{\text{e}}$ defined in Eq. (3.12), owing to the presence of $\widehat{\tau}^{x\,\vee}_{j^\star-1}\widehat{\tau}^{x\,\vee}_{j^\star}$ and $\widehat{\tau}^{x\,\vee}_{j^\star+n-1}\widehat{\tau}^{x\,\vee}_{j^\star+n}$ to the left and to right of the string of (4.15a), respectively, on the right-hand side of Eq. (4.15b). The operator $\widehat{C}^{y\,\vee}_{j^\star,j^\star+n-1}$ is the so-called string order parameter that detects the SPT ground state [118, 156, 157], while having vanishing expectation value in the trivial ground state (4.11).

Finally, along the MG line $(\Delta, J = 1/2)$, both dimer ground states (4.6c) belong to the subspace $\mathcal{H}_{b=0;+}$. However, out of the four ground states (4.13) of Hamiltonian (4.9), only

Table 6: The expectation values the operators (4.15) in the dual ground states (4.11), (4.12), and (4.18).

| | $\widehat{C}^{x\,\vee}_{j^\star,j^\star+n-1}$ | $\widehat{C}^{y\,\vee}_{j^\star,j^\star+n-1}$ | $\widehat{D}^\vee_{j^\star}$ |
|---|---|---|---|
| $\lvert\text{PM}\rangle$ | $(-1)^n$ | $0$ | $-\frac{1}{3}$ |
| $\lvert\text{SPT}\rangle$ | $0$ | $(-1)^n$ | $-\frac{1}{3}$ |
| $\lvert\text{Dimer}^\vee_{\text{o}}\rangle$ | $-\delta_{(-1)^j,-1}\,\delta_{n,1}$ | $-\delta_{(-1)^j,-1}\,\delta_{n,1}$ | $-\delta_{(-1)^j,-1}$ |
| $\lvert\text{Dimer}^\vee_{\text{e}}\rangle$ | $-\delta_{(-1)^j,+1}\,\delta_{n,1}$ | $-\delta_{(-1)^j,+1}\,\delta_{n,1}$ | $-\delta_{(-1)^j,+1}$ |

the two linear combinations

$$\lvert\text{Dimer}^\vee_{\text{o}}\rangle := \frac{1}{\sqrt{2}}\left(\lvert 1\rangle + \lvert 2\rangle\right), \qquad \widehat{U}^\vee_{r^z_\pi}\lvert\text{Dimer}^\vee_0\rangle = +\lvert\text{Dimer}^\vee_{\text{o}}\rangle, \tag{4.18a}$$

$$\lvert\text{Dimer}^\vee_{\text{e}}\rangle := \frac{1}{\sqrt{2}}\left(\lvert 3\rangle + \lvert 4\rangle\right), \qquad \widehat{U}^\vee_{r^z_\pi}\lvert\text{Dimer}^\vee_{\text{e}}\rangle = +\lvert\text{Dimer}^\vee_{\text{e}}\rangle, \tag{4.18b}$$

belong to the subspace $\mathcal{H}^\vee_{b'=0;+}$. These two states are dual to the dimer states (4.6c), respectively. We refer to this twofold degenerate ground state manifold as the $\text{D}^+_8$ doublet. The ground states (4.18) break the reflection symmetry spontaneously, while they are both singlets under the internal symmetry group $\mathbb{Z}^{\text{o}}_2 \times \mathbb{Z}^{\text{e}}_2$. The expectation values of the dual operators (4.15) in these ground states are given in Table 6. The phase diagrams of Hamiltonians (4.1) and (4.9) in the restricted subspaces $\mathcal{H}_{b=0;+}$ and $\mathcal{H}^\vee_{b'=0;+}$ are compared in Fig. 5.

## 4.3 Jordan-Wigner dual interacting Majorana chain

We now study the Hamiltonian dual to the Hamiltonian (4.1) under the JW duality. As in Sec. 3.3, we select anti-periodic boundary conditions ($f = 1$) after the JW duality. Naive use of the dual bond algebra (2.52a) delivers the Hamiltonian

$$\widehat{H}^\vee_{f=1} := J_1 \sum_{j\in\Lambda}\left(\Delta_x\,\mathrm{i}\hat{\beta}^\vee_{j+1}\,\hat{\alpha}^\vee_j + \Delta_y\,\mathrm{i}\hat{\beta}^\vee_j\,\hat{\alpha}^\vee_{j+1} + \Delta_z\,\hat{\beta}^\vee_j\,\hat{\beta}^\vee_{j+1}\,\hat{\alpha}^\vee_j\,\hat{\alpha}^\vee_{j+1}\right)$$
$$+ J_2 \sum_{j=1}^{2N}\left(\Delta_x\,\hat{\beta}^\vee_{j+1}\,\hat{\beta}^\vee_{j+2}\,\hat{\alpha}^\vee_j\,\hat{\alpha}^\vee_{j+1} + \Delta_y\,\hat{\alpha}^\vee_{j+1}\,\hat{\alpha}^\vee_{j+2}\,\hat{\beta}^\vee_j\,\hat{\beta}^\vee_{j+1} + \Delta_z\,\hat{\beta}^\vee_j\,\hat{\beta}^\vee_{j+2}\,\hat{\alpha}^\vee_j\,\hat{\alpha}^\vee_{j+2}\right),$$
$$\tag{4.19}$$

with the domain of definition $\mathcal{H}^\vee_{f=1}$ defined in Eq. (2.22). However, Hamiltonians (4.1) and (4.19) only form a dual pair if their domains of definition are restricted to the subspaces $\mathcal{H}_{b=0;+}$ and $\mathcal{H}^\vee_{f=1;+}$, respectively [recall Eq. (2.34)]. With this in mind, we will first study the phase diagram of Hamiltonian (4.19) in the full Hilbert space $\mathcal{H}^\vee_{f=1}$. We will then discuss the duality of phases in the restricted Hilbert spaces $\mathcal{H}_{b=0;+}$ and $\mathcal{H}^\vee_{f=1;+}$. Without loss of generality, we consider only the reduced coupling space (4.3) with $J \leq 1/2$.

The symmetries of $\widehat{H}^\vee_{f=1}$ that we shall keep track of are given in Sec. 3.3. Because the global internal symmetry subgroup $\text{G}^{\vee,\text{F}}_{\text{int}} \subset \text{G}^{\vee,\text{F}}_{\text{tot}}$ is represented by a trivial projective representation locally, the LSM Theorems 1 and 2 are inoperative. Hence, $\widehat{H}^\vee_{f=1}$ could exhibit a non-degenerate gapped ground state in its phase diagram, a possibility that is indeed realized. We restrict ourselves to the dual of the subgroup (4.5), which is the subgroup

$$\mathbb{Z}^{\text{Fr}}_4 \ltimes \mathbb{Z}^{\text{o}}_2 \times \mathbb{Z}^{\text{e}}_2/\mathbb{Z}^{\text{F}}_2 \subset \text{G}^{\vee,\text{F}}_{\text{tot}}, \tag{4.20a}$$

where

$$Z_4^{\mathrm{Fr}} := \left\{ r, \; r^2 = p_{\mathrm{F}}, \; r^3, \; r^4 = e \right\} . \tag{4.20b}$$

By inspection, Hamiltonian (4.19) is quadratic along the $(\Delta, J) = (\Delta, 0)$ line. At either one of the two lower corners $(\Delta, J) = (0, 0)$ or $(\Delta, J) = (\infty, 0)$, $\widehat{H}_{f=1}^{\vee}$ simplifies to a Kitaev chain [158]. We denote the ground states at the points $(\Delta, J) = (0, 0)$ and $(\Delta, J) = (\infty, 0)$ by $|\mathrm{Kitaev}\rangle$ and $|\overline{\mathrm{Kitaev}}\rangle$, respectively, such that.[18]

$$\mathrm{i}\hat{\beta}_1^{\vee} \, \hat{\alpha}_{2N}^{\vee} \, |\mathrm{Kitaev}\rangle = +|\mathrm{Kitaev}\rangle, \quad \mathrm{i}\hat{\beta}_{j+1}^{\vee} \, \hat{\alpha}_j^{\vee} \, |\mathrm{Kitaev}\rangle = -|\mathrm{Kitaev}\rangle, \quad j = 1, \cdots, 2N-1, \tag{4.21a}$$

$$\mathrm{i}\hat{\beta}_{2N}^{\vee} \, \hat{\alpha}_1^{\vee} \, |\overline{\mathrm{Kitaev}}\rangle = +|\overline{\mathrm{Kitaev}}\rangle, \quad \mathrm{i}\hat{\beta}_j^{\vee} \, \hat{\alpha}_{j+1}^{\vee} \, |\overline{\mathrm{Kitaev}}\rangle = -|\overline{\mathrm{Kitaev}}\rangle, \quad j = 1, \cdots, 2N-1. \tag{4.21b}$$

These two ground states are both symmetric under the subgroup (4.20). They are the ground states of two distinct and non-trivial invertible fermionic phases of matter.[19] When appropriate open boundary conditions are imposed, the counterpart to Hamiltonian $\widehat{H}_{f=1}$ has two-fold degenerate ground states. This degeneracy arises owing to the existence of two Majorana zero modes, one of which is localized a the left end while the other is localized at the right end, of the open chain. At the point $(\Delta, J) = (1, 0)$, $\widehat{H}_{f=1}^{\vee}$ simplifies to free spinless fermions hopping with a uniform nearest-neighbor hopping amplitude along the chain.[20]

By dualization of the projectors onto the MG ground states of Hamiltonian (4.1) along the MG line (4.4c), it is shown in Appendix A that the ground states of Hamiltonian $\widehat{H}_{f=1}^{\vee}$ are two-fold degenerate along the open MG line. We can always choose an orthonormal basis of ground states such that the basis elements are the dual to the dimer states (4.6c). This dual basis is given by

$$|\mathrm{Bonding}_{\mathrm{o}}^{\vee}\rangle := \left[ \prod_{j=1}^{N} \frac{1}{\sqrt{2}} \left( \hat{c}_{2j-1}^{\vee \dagger} + \hat{c}_{2j}^{\vee \dagger} \right) \right] |0\rangle,$$

$$|\mathrm{Bonding}_{\mathrm{e}}^{\vee}\rangle := \left[ \prod_{j=1}^{N} \frac{1}{\sqrt{2}} \left( \hat{c}_{2j}^{\vee \dagger} + \hat{c}_{2j+1}^{\vee \dagger} \right) \right] |0\rangle, \tag{4.22a}$$

where the complex fermion operators are defined as

$$\hat{c}_j^{\vee \dagger} := \frac{1}{2} (\hat{\alpha}_j^{\vee} - \mathrm{i}\hat{\beta}_j^{\vee}), \qquad \hat{c}_j^{\vee} := \frac{1}{2} (\hat{\alpha}_j^{\vee} + \mathrm{i}\hat{\beta}_j^{\vee}). \tag{4.22b}$$

These states spontaneously break the reflection symmetry while preserving the internal $\mathbb{Z}_2^{\mathrm{o}} \times \mathbb{Z}_2^{\mathrm{e}}$ symmetry.

The JW duality implies that the expectation value of any operator from the bond algebra (2.4) restricted to the Hilbert space $\mathcal{H}_{b=0;+}$ has the same expectation value as its dual in the bond algebra (2.31) restricted to the Hilbert space $\mathcal{H}_{f=1;+}^{\vee}$. Under the isomorphism between

---

[18] Terms $\mathrm{i}\hat{\beta}_{2N+1}^{\vee} \, \hat{\alpha}_{2N}^{\vee} = -\mathrm{i}\hat{\beta}_1^{\vee} \, \hat{\alpha}_{2N}^{\vee}$ and $\mathrm{i}\hat{\beta}_{2N}^{\vee} \, \hat{\alpha}_{2N+1}^{\vee} = -\mathrm{i}\hat{\beta}_{2N}^{\vee} \, \hat{\alpha}_1^{\vee}$ come with an additional minus sign because of the anti-periodic boundary conditions ($f = 1$).

[19] In fact, they are inverse of each other under the fermionic stacking operation [159–161].

[20] The upper corners $(\Delta, J) = (0, \infty)$ and $(\Delta, J) = (\infty, \infty)$ are gapped and two-fold degenerate nder antiperiodic boundary conditions as a consequence of the dualization from Sec. 2.4.1. The image under $J \to 1/J$ of the point $(\Delta, J) = (1, 0)$, is gapless, as a consequence of the dualization from Sec. 2.4.1.

Table 7: The expectation values the operators (4.15) in the dual ground states (4.11), (4.12), and (4.18).

| | $\widehat{C}_{j,j+n}^{x\,\vee}$ | $\widehat{C}_{j,j+n}^{y\,\vee}$ | $\widehat{D}_j^{\vee}$ |
|---|---|---|---|
| $\lvert \text{Kitaev}\rangle$ | $(-1)^n$ | $0$ | $-\frac{1}{3}$ |
| $\lvert \overline{\text{Kitaev}}\rangle$ | $0$ | $(-1)^n$ | $-\frac{1}{3}$ |
| $\lvert \text{Bonding}_\text{o}^{\vee}\rangle$ | $-\delta_{(-1)^j,-1}\,\delta_{n,1}$ | $-\delta_{(-1)^j,-1}\,\delta_{n,1}$ | $-\delta_{(-1)^j,-1}$ |
| $\lvert \text{Bonding}_\text{e}^{\vee}\rangle$ | $-\delta_{(-1)^j,+1}\,\delta_{n,1}$ | $-\delta_{(-1)^j,+1}\,\delta_{n,1}$ | $-\delta_{(-1)^j,+1}$ |

the bond algebras (2.4) and (2.31), operators (4.8) dualize to

$$\widehat{C}_{j,j+n}^{x\,\vee} := \prod_{\ell=j}^{j+n-1} \mathrm{i}\hat{\beta}_{\ell+1}^{\vee}\,\hat{\alpha}_\ell^{\vee}\,, \tag{4.23a}$$

$$\widehat{C}_{j,j+n}^{y\,\vee} := \prod_{\ell=j}^{j+n-1} \mathrm{i}\hat{\beta}_{\ell}^{\vee}\,\hat{\alpha}_{\ell+1}^{\vee}\,, \tag{4.23b}$$

$$\widehat{D}_j^{\vee} := \frac{1}{3}\left(\mathrm{i}\hat{\beta}_{j+1}^{\vee}\,\hat{\alpha}_j^{\vee} + \mathrm{i}\hat{\beta}_j^{\vee}\,\hat{\alpha}_{j+1}^{\vee} + \hat{\beta}_j^{\vee}\,\hat{\beta}_{j+1}^{\vee}\,\hat{\alpha}_j^{\vee}\,\hat{\alpha}_{j+1}^{\vee}\right)\,. \tag{4.23c}$$

As was the case in Sec. 4.2, we observe that operators $\widehat{C}_{j,j+n}^x$ and $\widehat{C}_{j,j+n}^y$ defined in Eqs. (4.8a) and (4.8b), respectively, dualize to non-local string operators, while the local operator $\widehat{D}_j$ defined in Eq. (4.23c) remains local after dualization.

Under the JW duality, the bonding linear combinations $\lvert \text{Neel}^x\rangle^+$ defined in Eq. (4.16) and $\lvert \text{Neel}^y\rangle^+$ defined in Eq. (4.17) of $\text{Neel}_x$ and $\text{Neel}_y$ states dualize to the two topologically nontrivial ground states $\lvert \text{Kitaev}\rangle$ and $\lvert \overline{\text{Kitaev}}\rangle$ defined in Eq. (4.21), respectively. These states can be distinguished by the expectation values of the string order parameters $\widehat{C}_{j,j+n}^{x\,\vee}$ and $\widehat{C}_{j,j+n}^{y\,\vee}$.

As opposed to the KW duality, the Hamiltonian (4.19) that obeys open boundary conditions is equivalent, up to a unitary transformation, to the dual of the Hamiltonian (4.1) that obeys open boundary conditions. It is shown in Appendix B that selecting open boundary conditions removes the consistency conditions on the bond algebra that require the projections of the Hilbert spaces $\mathcal{H}_{b=0}$ and $\mathcal{H}_{f=1}^{\vee}$ onto their subspaces $\mathcal{H}_{b=0,+}$ and $\mathcal{H}_{f=1,+}^{\vee}$ for duality to hold. The two-fold degeneracy of the $\text{Neel}_x$ and $\text{Neel}_y$ ground states dualizes to the two-fold degeneracy of the non-trivial invertible topological phases with open boundary conditions. Finally, along the MG line, the two-fold degenerate dimer ground states (4.6c) of Hamiltonian (4.1) dualize to the two-fold degenerate bond-density order ground states (4.22) of Hamiltonian (4.19). The expectations values of the operators (4.23) in the ground states of Hamiltonian (4.19) are given in Table 7.

The phase diagrams below the MG line of Hamiltonians (4.1) and (4.19) defined on their domain of definitions $\mathcal{H}_{b=0;+}$ and $\mathcal{H}_{f=1;+}^{\vee}$, respectively, are compared in Fig. 6. Whereas the vertical phase boundary at $\Delta = 1$ remains a line of quantum critical points (blue line) outside of the Landau-Ginzburg paradigm, the boundaries separating the two topologically nontrivial singlet phases from the bond-density ordered phase are ordinary Landau-Ginzburg phase transitions (red lines).

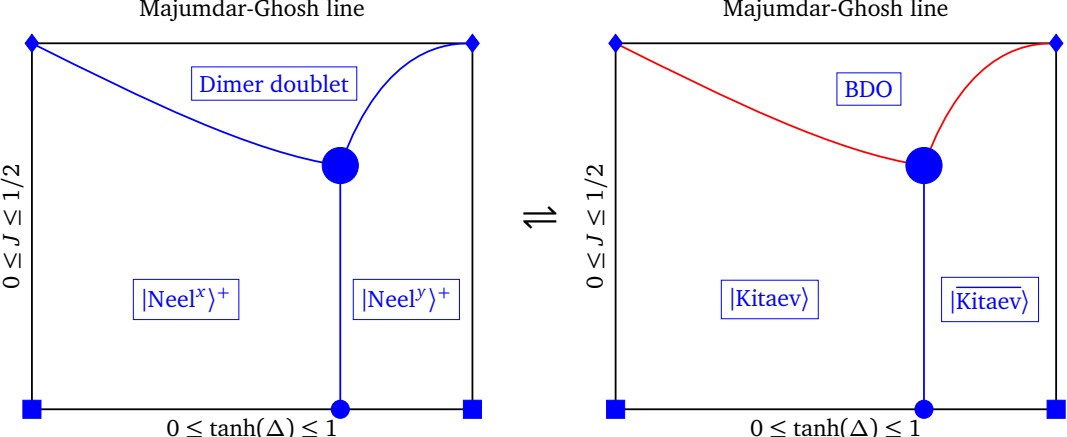

**Figure 6:** The phase diagram of Hamiltonian (4.1) restricted to the subspace $\mathcal{H}_{b=0;+}$ of the Hilbert space $\mathcal{H}_{b=0}$ is dual to the phase diagram of Hamiltonian (4.19) restricted to the subspace $\mathcal{H}^{\vee}_{f=1;+}$ of $\mathcal{H}^{\vee}_{f=1}$. The pair of dual subspaces are to be found in the third line of Table 2. The two states $|\text{Neel}^x\rangle^+$ and $|\text{Neel}^y\rangle^+$ are defined in Eqs. (4.16) and (4.17), respectively, while the box "Dimer doublet" refers to the ground states (4.6c). On the dual side, the two non-trivial and distinct invertible topological states $|\text{Kitaev}\rangle$ and $|\overline{\text{Kitaev}}\rangle$ are defined in Eq. (4.21). The box "BDO" stands for the bond-density ordered phase described by the two-fold degenerate ground states (4.22) along the MG line. The symbols $\rightharpoonup$ and $\leftharpoonup$ denote gauging the diagonal subgroups generated by $\widehat{U}_{r^z_\pi}$ and by its dual $\widehat{P}^{\vee}_{\text{F}}$ defined in Eq. (3.21a), respectively.

## 5 Quantum $\mathbb{Z}_n$ clock models with $n$ mod 2 LSM anomalies

We are going to generalize the spin-1/2 chains with global $\mathbb{Z}_2$ symmetry that we have studied in Secs. 2–4 to clock models with global $\mathbb{Z}_n$ symmetry whereby $n = 2, 3, \cdots$. Our aim is to establish how, as a consequence of an LSM anomaly, the crystalline and internal symmetries become intertwined under dualities obtained by gauging the global $\mathbb{Z}_n$ symmetry. We are going to show that the non-trivial mixing of the crystalline and internal symmetries is sensitive to the parity of $n = 2, 3, \cdots$.

### 5.1 A generalized LSM anomaly

Consider a one-dimensional lattice $\Lambda$ of cardinality $2Nn$ with the integers $N = 1, 2, \cdots$ and $n = 2, 3, \cdots$. To each site $j$ of the lattice, we assign an $n$-dimensional complex Hilbert space $\mathbb{C}^n$ on which we may represent the clock operator $\widehat{Z}_j$ and the shift operator $\widehat{X}_j$ obeying the algebra

$$\widehat{X}_i \widehat{Z}_j = \left(\omega_n\right)^{\delta_{i,j}} \widehat{Z}_j \widehat{X}_i, \qquad \left(\widehat{X}_j\right)^n = \left(\widehat{Z}_j\right)^n = \widehat{\mathbb{1}}_{\mathcal{H}_b}, \qquad \omega_n := e^{i\frac{2\pi}{n}}, \qquad i, j \in \Lambda, \qquad (5.1a)$$

by $n$-dimensional complex-valued unitary matrices.[21] As we will impose the global internal symmetry $\mathbb{Z}^z_n$ that is generated by the unitary operator

$$\widehat{U}_z := \prod_{j \in \Lambda} \widehat{Z}_j, \qquad (5.1b)$$

---

[21]The shift operator $\widehat{X}_j$ and clock operator $\widehat{Z}_j$ are unitary for any $j \in \Lambda$, i.e., $\left(\widehat{X}_j\right)^{-1} = \left(\widehat{X}_j\right)^{\dagger}$ and $\left(\widehat{Z}_j\right)^{-1} = \left(\widehat{Z}_j\right)^{\dagger}$.

we impose the twisted boundary conditions

$$\widehat{X}_{j+2Nn} = \left(\widehat{U}_z\right)^b \widehat{X}_j \left(\widehat{U}_z\right)^{-b} = \left(\omega_n\right)^b \widehat{X}_j, \qquad \widehat{Z}_{j+2Nn} = \left(\widehat{U}_z\right)^b \widehat{Z}_j \left(\widehat{U}_z\right)^{-b} = \widehat{Z}_j, \qquad (5.1c)$$

for any $b \in \mathbb{Z}_n$ on the Hilbert space

$$\mathcal{H}_b := \bigotimes_{j \in \Lambda} \mathbb{C}^n. \qquad (5.1d)$$

We define the bond algebra

$$\mathfrak{B}_b := \left\langle \widehat{Z}_j, \quad \widehat{X}_j \left(\widehat{X}_{j+1}\right)^{-1} \;\middle|\; j \in \Lambda \right\rangle, \qquad (5.2a)$$

of operators that are symmetric under the $\mathbb{Z}_n^z$ symmetry generated by Eq. (5.1b). We decompose the Hilbert space into definite eigenvalue sectors of $\widehat{U}_z$

$$\mathcal{H}_b = \bigoplus_{\alpha=0}^{n-1} \mathcal{H}_{b;\alpha}, \qquad \mathcal{H}_{b;\alpha} := \widehat{P}_{b;\alpha} \mathcal{H}_b, \qquad \alpha = 0, \cdots, n-1, \qquad (5.3)$$

where $\widehat{P}_{b;\alpha}$ is the projector to the subspace with definite eigenvalue $\left(\omega_n\right)^\alpha$ of $\widehat{U}_z$.

In addition to the global internal $\mathbb{Z}_n^z$ symmetry of the bond algebra (5.2), we presume additional crystalline and internal symmetries. To accommodate translation symmetry in a simple way, we select periodic boundary conditions by choosing $b = 0$. First, we shall impose two crystalline symmetries, namely, translations and site-centered reflection of lattice $\Lambda$ which are implemented by the unitary operators

$$\widehat{U}_t \widehat{X}_j \widehat{U}_t^\dagger = \widehat{X}_{t(j)}, \qquad \widehat{U}_t \widehat{Z}_j \widehat{U}_t^\dagger = \widehat{Z}_{t(j)}, \qquad t(j) = j+1 \bmod 2Nn, \qquad (5.4a)$$

$$\widehat{U}_r \widehat{X}_j \widehat{U}_r^\dagger = \widehat{X}_{r(j)}, \qquad \widehat{U}_r \widehat{Z}_j \widehat{U}_r^\dagger = \widehat{Z}_{r(j)}, \qquad r(j) = 2Nn-j \bmod 2Nn, \qquad (5.4b)$$

respectively. Choosing the lattice $\Lambda$ to be made of an even number of sites ensures that site-centered reflection exists for any $n$ and has the two fixed points $j = Nn$ and $j = 2Nn$. The operators $\widehat{U}_t$ and $\widehat{U}_r$ generate the representation of the space group

$$G_{\text{spa}} \equiv \mathbb{Z}_{2Nn}^t \rtimes \mathbb{Z}_2^r. \qquad (5.4c)$$

Next, we impose an additional global internal symmetry $\mathbb{Z}_n^x$ that is implemented by the unitary operator

$$\widehat{U}_x := \prod_{j \in \Lambda} \widehat{X}_j, \qquad (5.5a)$$

i.e., the product of all local shift operators. Together, $\widehat{U}_x$ and $\widehat{U}_z$ generate a global representation of the Abelian group $\mathbb{Z}_n^x \times \mathbb{Z}_n^z$. Thus, the total symmetry group is the direct product

$$G_{\text{tot}} \equiv G_{\text{spa}} \times G_{\text{int}}, \qquad G_{\text{int}} \equiv \mathbb{Z}_n^x \times \mathbb{Z}_n^z. \qquad (5.5b)$$

While the global representation of $G_{\text{int}}$ is a group homomorphism, it is locally projective due to the algebra

$$\widehat{X}_j \widehat{Z}_j = \omega_n \widehat{Z}_j \widehat{X}_j, \qquad j \in \Lambda. \qquad (5.6a)$$

More precisely, distinct projective representations of the group $\mathbb{Z}_n \times \mathbb{Z}_n$ are labeled by the equivalence classes $[\omega] = 0, 1, \cdots, n-1$ taking values in the second cohomology group

$$[\omega] \in H^2\big(\mathbb{Z}_n \times \mathbb{Z}_n, U(1)\big) = \mathbb{Z}_n. \qquad (5.6b)$$

The algebra (5.6a) is a representative of the generator $[\omega] = 1$ of the cohomology group (5.6b). Because of the projective algebra (5.6a), the following LSM Theorem with translation symmetry applies.

**Theorem 3** (Generalized translation LSM). Consider a one-dimensional lattice Hamiltonian with the symmetry group $G_{tot} \equiv Z_{|\Lambda|}^t \times \mathbb{Z}_n^x \times \mathbb{Z}_n^y$, where the subgroup $Z_{|\Lambda|}^t$ generates lattice translations and the subgroup $\mathbb{Z}_n^x \times \mathbb{Z}_n^y$ with $n = 2, 3, \cdots$ generates global internal discrete clock-rotation symmetry. Let $[\omega] \in H^2\big(\mathbb{Z}_n \times \mathbb{Z}_n, U(1)\big) = \mathbb{Z}_n$ denote the second cohomology class associated with the local representation of $\mathbb{Z}_n^x \times \mathbb{Z}_n^z$ at any site of $\Lambda$. If $[\omega] \neq 0 \bmod n$, then the ground states cannot be simultaneously gapped, non-degenerate, and $G_{tot}$-symmetric.

**Definition 3** (Generalized translation LSM anomaly). When LSM Theorem 3 applies, we say that there is a translation LSM anomaly.

*Remark.* Since LSM Theorem 3 holds for any integer $n = 2, 3, \cdots$, the dual of $\widehat{U}_x$ is not translationally invariant for any value of $n = 2, 3, \cdots$.

Unlike with Theorems 1–3, we are not aware of a rigorous proof of the Conjecture 4 that follows (see Refs. [18, 162]). Conjecture 4 is expected to hold based on the lattice homotopy arguments introduced in Ref. [18] and crystalline equivalence principle introduced in Refs. [28, 55].

**Conjecture 4** (Generalized reflection LSM). Consider a one-dimensional lattice Hamiltonian with the symmetry group $G_{tot} \equiv Z_2^r \times \mathbb{Z}_n^x \times \mathbb{Z}_n^z$, where the subgroup $Z_2^r$ is generated by a site-centered reflection, while the subgroup $\mathbb{Z}_n^x \times \mathbb{Z}_n^y$ with $n = 2, 3, \cdots$ is generated by two global internal discrete clock-rotation symmetries. Let $[\omega] \in H^2\big(\mathbb{Z}_n \times \mathbb{Z}_n, U(1)\big) = \mathbb{Z}_n$ denote the second cohomology class associated with the local representation of $\mathbb{Z}_n^x \times \mathbb{Z}_n^z$ at any one of the fixed points of the reflection. If $[\omega] \neq 2k \bmod n$ for some integer $k$, then the ground states cannot be simultaneously gapped, non-degenerate, and $G_{tot}$-symmetric.

**Definition 4** (Reflection LSM anomaly). When **Conjecture 4** applies, we say that there is a reflection LSM anomaly.

*Remark.* Conjecture 4 reduces to Theorem 2 for $n = 2$ and $[\omega] = 1$, which is the only non-trivial projective representation realized by half-integer spins. Furthermore, when $n$ is odd, the condition $[\omega] = 2k \bmod n$ is always satisfied for some integer $k$. Hence, there is no generalized reflection LSM anomaly when $n$ is odd. For the algebra (5.6a), we have $[\omega] = 1$ which implies that a non-degenerate, gapped, and $G_{tot}$-symmetric ground state is possible only when $n = 3, 5, \cdots$, while it is ruled out by Conjecture 4 when $n = 2, 4, \cdots$. In what follows, we are going to confirm this claim by showing that the operator $\widehat{U}_x$ cannot be dualized and remain invariant under reflection when $n$ is even, while it can be when $n$ is odd.

## 5.2 Kramers-Wannier dual of the generalized LSM anomaly

Starting from the bond algebra $\mathfrak{B}_b$ in Eq. (5.2a), we are going to perform a gauging of $\widehat{U}_z$. This gauging furnishes a dual bond algebra, where the duality is nothing but a $\mathbb{Z}_n$ generalization of KW duality described in Sec. 2.2. We are then going to invoke an additional $\mathbb{Z}_n^x$ symmetry that is generated by $\widehat{U}_x$ defined in Eq. (5.5a) and construct its dual $\widehat{U}_x^\vee$ under the $\mathbb{Z}_n$ generalization of KW duality. Our main result will be that the action of reflection on $\widehat{U}_x^\vee$ turns out to be non-trivial (trivial) if $n = 0 \bmod 2$ ($n = 1 \bmod 2$). This result is aligned with the LSM anomaly conjecture 4.

In order to gauge $\widehat{U}_z$ defined in Eq. (5.1c), we introduce $\mathbb{Z}_n$-valued gauge degrees of freedom on the dual lattice $\Lambda^\star$. To each site $j^\star$ of the dual lattice $\Lambda^\star$, we therefore associate an $n$-dimensional Hilbert space. The operator algebra attached to $j^\star \in \Lambda^\star$ is generated by the clock operator $\widehat{Z}_{j^\star}^\star$ and the shift operator $\widehat{X}_{j^\star}^\star$ that satisfy the same algebra (5.1a) except for substituting $\Lambda$ by $\Lambda^\star$. To gauge the global symmetry (5.1b), we define the unitary local Gauss operator

$$\widehat{G}_j := \big(\widehat{X}_{j^\star-1}^\star\big)^{-1} \widehat{Z}_j \widehat{X}_{j^\star}^\star, \tag{5.7}$$

for any site $j \in \Lambda$. These local Gauss operators commute pairwise on distinct sites. By analogy to Eq. (5.1c), the twisted boundary conditions

$$\widehat{X}^{\star}_{j^{\star}+2Nn} = \left( \omega_n \right)^{b'} \widehat{X}^{\star}_{j^{\star}} \,, \qquad \widehat{Z}^{\star}_{j^{\star}+2Nn} = \widehat{Z}^{\star}_{j^{\star}} \,, \tag{5.8}$$

are imposed for the $\mathbb{Z}_n$-valued gauge degrees of freedom. The extended Hilbert space including the original (matter) and $\mathbb{Z}_n$-valued gauge degrees of freedom defined on $\Lambda$ and $\Lambda^{\star}$, respectively, admits the tensor decomposition

$$\mathcal{H}_{b,b'} := \mathcal{H}_b \otimes \mathcal{H}_{b'} \,, \qquad \mathcal{H}_b := \bigotimes_{j \in \Lambda} \mathbb{C}^n \,, \qquad \mathcal{H}_{b'} := \bigotimes_{j^{\star} \in \Lambda^{\star}} \mathbb{C}^n \,. \tag{5.9}$$

A generic gauge transformation

$$\begin{aligned} \widehat{Z}_j &\longmapsto \widehat{Z}_j \,, & \widehat{X}_j &\longmapsto \omega_n^{-\lambda_j} \widehat{X}_j \,, \\ \widehat{X}^{\star}_{j^{\star}} &\longmapsto \widehat{X}^{\star}_{j^{\star}} \,, & \widehat{Z}^{\star}_{j^{\star}} &\longmapsto \omega_n^{(\lambda_j - \lambda_{j+1})} \widehat{Z}^{\star}_{j^{\star}} \,, \end{aligned} \tag{5.10}$$

is specified by the variables $\lambda_j \in \mathbb{Z}_n$ with $j \in \Lambda$ and implemented by conjugation with the unitary operator

$$\widehat{G}_{\boldsymbol{\lambda}} := \prod_{j \in \Lambda} \left( \widehat{G}_j \right)^{\lambda_j} \,, \qquad \boldsymbol{\lambda} := \left( \lambda_1, \cdots, \lambda_{2Nn} \right) \,. \tag{5.11}$$

By minimally coupling the bond algebra (5.2a), we obtain the extended bond algebra

$$\mathfrak{B}_{b,b'} := \left\langle \widehat{Z}_j, \qquad \widehat{X}_j \widehat{Z}^{\star}_{j^{\star}} \left( \widehat{X}_{j+1} \right)^{-1} \ \middle| \ j \in \Lambda \right\rangle \,, \tag{5.12}$$

of gauge-invariant operators. As was done in Sec. 2, there exists a unitary operator $\widehat{U}_{b,b'}$ that implements the transformation [51]

$$\begin{aligned} \widehat{U}_{b,b'} \widehat{X}_j \widehat{U}^{\dagger}_{b,b'} &= \widehat{X}_j \,, & \widehat{U}_{b,b'} \widehat{Z}_j \widehat{U}^{\dagger}_{b,b'} &= \widehat{X}^{\star}_{j^{\star}-1} \widehat{Z}_j \left( \widehat{X}^{\star}_{j^{\star}} \right)^{-1} \,, \\ \widehat{U}_{b,b'} \widehat{X}^{\star}_{j^{\star}} \widehat{U}^{\dagger}_{b,b'} &= \widehat{X}^{\star}_{j^{\star}} \,, & \widehat{U}_{b,b'} \widehat{Z}^{\star}_{j^{\star}} \widehat{U}^{\dagger}_{b,b'} &= \left( \widehat{X}_j \right)^{-1} \widehat{Z}^{\star}_{j^{\star}} \widehat{X}_{j+1} \,, \end{aligned} \tag{5.13a}$$

for any $j \in \Lambda$ and $j^{\star} \in \Lambda^{\star}$. In particular, under this transformation, the local Gauss operator (5.11) becomes

$$\widehat{U}_{b,b'} \widehat{G}_j \widehat{U}^{\dagger}_{b,b'} = \widehat{Z}_j \,, \tag{5.13b}$$

for any $j \in \Lambda$. The subspace of physical states is defined to be the one for which the action of all local Gauss operators reduces to the identity. Hence, after the unitary transformation (5.13), the subspace of physical states is defined to be the one for which the action of $\widehat{Z}_j$ for any $j \in \Lambda$ reduces to the identity. It is the $n^{2Nn}$-dimensional gauge-invariant subspace $\mathcal{H}^{\vee}_{b'}$ of $\mathcal{H}_{b,b'}$. The projection of the bond algebra (5.12) to the subspace $\mathcal{H}^{\vee}_{b'}$ delivers the dual bond algebra

$$\mathfrak{B}_{b'} := \left\langle \widehat{X}^{\star\vee}_{j^{\star}-1} \left( \widehat{X}^{\star\vee}_{j^{\star}} \right)^{-1} \,, \qquad \widehat{Z}^{\star\vee}_{j^{\star}} \ \middle| \ j^{\star} \in \Lambda^{\star} \right\rangle \,, \tag{5.14a}$$

which is symmetric under the dual $\mathbb{Z}_n^{z^{\vee}}$-symmetry generated by the unitary operator

$$\widehat{U}^{\vee}_{z^{\vee}} := \prod_{j^{\star} \in \Lambda^{\star}} \widehat{Z}^{\star\vee}_{j^{\star}} \,. \tag{5.14b}$$

The projected Hilbert space $\mathcal{H}_{b'}^{\vee}$ is isomorphic to the Hilbert space (5.1d). It can be decomposed into subspaces with definite eigenvalue of $\widehat{U}_{z_n}^{\vee}$ [as was done in Eq. (5.3)],

$$
\mathcal{H}_{b'}^{\vee} = \bigoplus_{\alpha=0}^{n-1} \mathcal{H}_{b';\alpha}^{\vee}, \qquad \mathcal{H}_{b';\alpha}^{\vee} := \widehat{P}_{b';\alpha}^{\vee} \mathcal{H}_{b'}^{\vee}, \qquad \alpha = 0, \cdots, n-1, \tag{5.15}
$$

where $\widehat{P}_{b';\alpha}^{\vee}$ is the projector to subspace with definite eigenvalue $\omega_n^\alpha$ of $\widehat{U}_{r_n^{z\vee}}^{\vee}$.

Consistency with the pair of twisted boundary conditions (5.1c) and (5.8) requires the identification of the pair of operators

$$
\left( \prod_{j \in \Lambda} \widehat{Z}_j \equiv \widehat{U}_{r_n^z}, \qquad \left( \omega_n \right)^{b'} \widehat{\mathbb{1}}_{\mathcal{H}_{b'}^{\vee}} \right), \tag{5.16a}
$$

on the one hand and the pair of operators

$$
\left( \left( \omega_n \right)^{b} \widehat{\mathbb{1}}_{\mathcal{H}_b}, \qquad \prod_{j^\star \in \Lambda^\star} \widehat{Z}_{j^\star}^{\star\vee} \equiv \widehat{U}_{r_n^z}^{\vee} \right), \tag{5.16b}
$$

on the other hand. This pair of consistency conditions can only be met if the domains of definition of all dual pairs of operators are restricted to the dual pair of subspaces

$$
\mathcal{H}_{b;b'} := \widehat{P}_{b;b'} \mathcal{H}_b, \qquad \mathcal{H}_{b';b}^{\vee} := \widehat{P}_{b';b}^{\vee} \mathcal{H}_{b'}^{\vee}. \tag{5.17}
$$

The boundary conditions $b$ on the Hilbert space $\mathcal{H}_b$ dictates the definite eigenvalue subspace of the dual Hilbert space $\mathcal{H}_{b'}^{\vee}$ and vice versa. This is the $\mathbb{Z}_n$ generalization of the KW duality from Sec. 2.2.

We now turn to obtaining the duals of the crystalline and internal symmetries (5.4) and (5.5), respectively. We impose periodic boundary conditions for both bond algebras (5.2) and (5.14a) by choosing $b = b' = 0$.

As described in Sec. 3.2, the dual crystalline symmetries are obtained by first extending the operators to the Hilbert space (5.9) by demanding covariance of the Gauss operators (5.7). We obtain the duals of operators $\widehat{U}_t$ and $\widehat{U}_r$ defined by their actions on the Hilbert space $\mathcal{H}_{b'=0}^{\vee}$

$$
\widehat{U}_t^{\vee} \widehat{X}_{j^\star}^{\star\vee} \left( \widehat{U}_t^{\vee} \right)^\dagger = \widehat{X}_{t(j^\star)}^{\star\vee}, \qquad \widehat{U}_t^{\vee} \widehat{Z}_{j^\star}^{\star\vee} \left( \widehat{U}_t^{\vee} \right)^\dagger = \widehat{Z}_{t(j^\star)}^{\star\vee}, \qquad t(j^\star) = j^\star + 1, \tag{5.18a}
$$

$$
\widehat{U}_r^{\vee} \widehat{X}_{j^\star}^{\star\vee} \left( \widehat{U}_r^{\vee} \right)^\dagger = \left( \widehat{X}_{r(j^\star)}^{\star\vee} \right)^{-1}, \quad \widehat{U}_r^{\vee} \widehat{Z}_{j^\star}^{\star\vee} \left( \widehat{U}_r^{\vee} \right)^\dagger = \left( \widehat{Z}_{r(j^\star)}^{\star\vee} \right)^{-1}, \quad r(j^\star) = 2Nn - j^\star. \tag{5.18b}
$$

as it should be, these transformation rules reduce to those in Eqs. (3.7c) and (3.7d) when $n = 2$. Since $\widehat{X}_{j^\star}^{\star\vee}$ and $\widehat{Z}_{j^\star}^{\star\vee}$ are akin to electric field $e^{iE}$ and gauge field $e^{iA}$, respectively, they are to be Hermitian conjugated under reflection in Eq. (5.18b).

Due to the projective algebra (5.6a), the global symmetry operator $\widehat{U}_x$ is not gauge invariant. We therefore dualize it by expressing it in terms of products of local operators from the bond algebra (5.2a). This allows us to use the isomorphism between the dual bond algebras (5.2a) and (5.14a). We treat the cases of $n$ even and $n$ odd separately.

**Case of $n$ even.** First, we rewrite the unitary operator $\widehat{U}_x$ defined in Eq. (5.5a) as

$$
\begin{aligned}
\widehat{U}_x = & \left[ \widehat{X}_1 \left( \widehat{X}_2 \right)^{-1} \right] \left[ \widehat{X}_2 \left( \widehat{X}_3 \right)^{-1} \right]^2 \left[ \widehat{X}_3 \left( \widehat{X}_4 \right)^{-1} \right]^3 \cdots \left[ \widehat{X}_{n-1} \left( \widehat{X}_n \right)^{-1} \right]^{n-1} \\
& \times \left[ \widehat{X}_{n+1} \left( \widehat{X}_{n+2} \right)^{-1} \right] \left[ \widehat{X}_{n+2} \left( \widehat{X}_{n+3} \right)^{-1} \right]^2 \cdots \left[ \widehat{X}_{2Nn-1} \left( \widehat{X}_{2Nn} \right)^{-1} \right]^{n-1},
\end{aligned} \tag{5.19a}
$$

where each term inside the square brackets is a generator of the bond algebra (5.2a). The $j$th square bracket on the right-hand side of Eq. (5.19a) becomes gauge invariant upon insertion of $\widehat{Z}_{j^\star}^\star$ between the pair of shift operators on the sites $j$ and $j+1$ of $\Lambda$. We may then use the isomorphism between dual bond algebras (5.2a) and (5.14a) to obtain the dual symmetry generator

$$\widehat{U}_{x^\vee}^\vee = \prod_{j^\star \in \Lambda^\star} \left( \widehat{Z}_{j^\star}^{\star\,\vee} \right)^{j^\star - 1/2} . \tag{5.19b}$$

Note that the dual operator (5.19b) is neither invariant under translations (5.18a) nor under reflection (5.18b). Instead, one verifies the algebra

$$\widehat{U}_t^\vee \, \widehat{U}_{x^\vee}^\vee \left( \widehat{U}_t^\vee \right)^\dagger = \left( \widehat{U}_{z^\vee}^\vee \right)^\dagger \widehat{U}_{x^\vee}^\vee , \qquad \widehat{U}_r^\vee \, \widehat{U}_{x^\vee}^\vee \left( \widehat{U}_r^\vee \right)^\dagger = \widehat{U}_{z^\vee}^\vee \, \widehat{U}_{x^\vee}^\vee . \tag{5.20}$$

Hence, we find that the total symmetry group (5.5b) dualizes to the symmetry group

$$\mathsf{G}_{\text{tot}}^\vee \equiv \mathsf{G}_{\text{spa}}^\vee \ltimes \mathsf{G}_{\text{int}}^\vee , \tag{5.21a}$$

with the internal symmetry group

$$\mathsf{G}_{\text{int}}^\vee \equiv \mathbb{Z}_n^{z\,\vee} \times \mathbb{Z}_n^{x\,\vee} , \tag{5.21b}$$

generated by $z^\vee$ and $x^\vee$ of order $n = 2, 3, \cdots$ that are represented by operators (5.14b) and (5.19b), respectively. The spatial symmetry group

$$\mathsf{G}_{\text{spa}}^\vee \equiv \mathbb{Z}_{2Nn}^t \rtimes \mathbb{Z}_2^r , \tag{5.21c}$$

has two generators $t$ and $r$ that are order $2Nn$ and 2; and represented by operators (5.18a) and (5.18b), respectively. The semi-direct product structure in the dual total symmetry group (5.21a) is due to the non-trivial group action

$$t\,x^\vee\,t^{-1} = (z^\vee)^{-1}\,x^\vee , \qquad r\,x^\vee\,r = z^\vee\,x^\vee , \qquad r\,z^\vee\,r = (z^\vee)^{-1} , \tag{5.21d}$$

of crystalline symmetries $\mathsf{G}_{\text{spa}}^\vee$ on the internal symmetries $\mathsf{G}_{\text{int}}^\vee$.

**Case of $n$ odd.** For the case of $n$ odd, Eq. (5.19a) can be used to reexpress the operator $\widehat{U}_x$. However, there is an alternative expression for $\widehat{U}_x$ which was not available when $n$ is even. We may write

$$\widehat{U}_x = \left[ \widehat{X}_{2Nn} \left( \widehat{X}_1 \right)^{-1} \right]^{\frac{n+1}{2}} \left[ \widehat{X}_1 \left( \widehat{X}_2 \right)^{-1} \right]^{\frac{n+3}{2}} \cdots \left[ \widehat{X}_{N\,n-1} \left( \widehat{X}_{N\,n} \right)^{-1} \right]^{\frac{n-1}{2}}$$

$$\times \left[ \widehat{X}_{2Nn} \left( \widehat{X}_{2Nn-1} \right)^{-1} \right]^{\frac{n+1}{2}} \left[ \widehat{X}_{2Nn-1} \left( \widehat{X}_{2Nn-2} \right)^{-1} \right]^{\frac{n+3}{2}} \cdots \left[ \widehat{X}_{N\,n+1} \left( \widehat{X}_{N\,n} \right)^{-1} \right]^{\frac{n-1}{2}} ,$$

where we have utilized the fact that $n$ is an odd integer when writing the exponents. The $j$th square bracket on the right-hand side of Eq. (5.19a) becomes gauge invariant upon insertion of $\widehat{Z}_{j^\star}^\star$ between the pair of shift operators on the sites $j$ and $j+1$ of $\Lambda$. We may then use the isomorphism between dual bond algebras (5.2a) and (5.14a) to obtain the dual symmetry generator

$$\widehat{U}_{x^\vee}^\vee = \left[ \prod_{j=0}^{Nn-1} \left( \widehat{Z}_{j^\star}^{\star\,\vee} \right)^{\frac{n+1}{2}+j} \right] \left[ \prod_{j=0}^{Nn-1} \left( \widehat{Z}_{2Nn-j^\star}^{\star\,\vee} \right)^{\frac{n-1}{2}-j} \right] . \tag{5.22}$$

While still not invariant under translation symmetry (5.18a), the unitary operator (5.22) is manifestly invariant under the reflection symmetry (5.18b). Therefore, we find the algebra

$$\widehat{U}_t^\vee \, \widehat{U}_{x^\vee}^\vee \, \left(\widehat{U}_t^\vee\right)^\dagger = \left(\widehat{U}_{z^\vee}^\vee\right)^\dagger \, \widehat{U}_{x^\vee}^\vee \,, \qquad \widehat{U}_r^\vee \, \widehat{U}_{x^\vee}^\vee \, \left(\widehat{U}_r^\vee\right)^\dagger = \widehat{U}_{x^\vee}^\vee \,. \tag{5.23a}$$

As opposed to the algebra (5.20), reflection commutes with the dual symmetry (5.22). The total symmetry group

$$\mathrm{G}_{\mathrm{tot}}^{\mathrm{o}\,\vee} \equiv \mathrm{G}_{\mathrm{spa}}^\vee \ltimes \mathrm{G}_{\mathrm{int}}^\vee \,, \tag{5.23b}$$

differs from the group (5.21a) by the group action

$$t \, x^\vee \, t^{-1} = (z^\vee)^{-1} \, x^\vee \,, \qquad r \, x^\vee \, r = x^\vee \,, \qquad r \, z^\vee \, r = (z^\vee)^{-1} \,, \tag{5.23c}$$

of crystalline symmetries $\mathrm{G}_{\mathrm{spa}}^\vee$ on the internal symmetries $\mathrm{G}_{\mathrm{int}}^\vee$.

The fact that the reflection symmetric decomposition (5.22) is only possible for $n$ odd is rooted in the LSM anomaly 4, which only applies when $n$ is an even integer.[22] Importantly, while the reflection symmetry has a nontrivial group action on the generator $z^\vee$ of the dual symmetry group $\mathbb{Z}_n^{z\,\vee}$, LSM anomalies 3 and 4 appear as the incompatibility between the image $\mathbb{Z}_n^{x\,\vee}$ of the ungauged internal symmetry group $\mathbb{Z}_n^x$ and crystalline symmetries.

We observe that, for any $n$, operators $\widehat{U}_{z^\vee}^\vee$ and $\widehat{U}_{x^\vee}^\vee$ implement $\mathbb{Z}_n$-charge and $\mathbb{Z}_n$-dipole symmetries, respectively. As was the case with $n = 2$ in Sec. 3, we find that $\mathbb{Z}_n$ clock Hamiltonians with charge, dipole, and translation symmetries are free from an LSM anomaly for any $n$ and are dual to Hamiltonians with global internal $\mathbb{Z}_n \times \mathbb{Z}_n$ symmetry with an LSM anomaly. A more detailed investigation of dualities between models with multipolar symmetries and models with uniform symmetries and LSM anomalies is left for future works.

# 6 Conclusions

In this paper, we studied the dualization induced by the gauging of global internal sub-symmetries of one-dimensional quantum spin chains with LSM anomalies. We found that when the pre-gauged theory had a non-trivial LSM anomaly, the dual theory was free from an LSM anomaly but had a symmetry structure wherein the crystalline and internal symmetries combined together through non-trivial group extensions. Therefore, the symmetry structure of the gauged theory was shown to serve as a diagnostic for LSM anomalies. Similar phenomena (restricted to only internal symmetries) have been studied extensively in the context of quantum field theory (see for example [66]), where gauging a non-anomalous symmetry participating in a mixed anomaly delivers a dual theory with a symmetry structure involving a group extension controlled by the anomaly of the pre-gauged theory. We exemplified our procedure for a $\mathbb{Z}_2 \times \mathbb{Z}_2$-symmetric quantum spin-1/2 $XYZ$ chain with LSM anomalies due to translation and reflection. We established a triality of models by gauging a $\mathbb{Z}_2 \subset \mathbb{Z}_2 \times \mathbb{Z}_2$ symmetry in two ways, which amount to performing Kramers-Wannier or Jordan-Wigner duality, respectively. We detailed the mapping of the phase diagram of the quantum spin-1/2 $XYZ$ chain under the triality and showed that the deconfined quantum critical transitions between Neel and valence-bond-solid orders of the chain map to either topological transitions or conventional Landau-Ginzburg-type transitions.

There are several future directions that could be pursued. One avenue is the generalization of the approach developed in this work to quantum lattice Hamiltonians with LSM anomalies for higher-dimensional lattices. We expect that in higher dimensions, gauging non-anomalous

---

[22]Our starting point is the projective algebra (5.6a) which corresponds to $[\omega] = 1$. It can be verified for general $[\omega]$ that a reflection symmetric decomposition is possible only when $[\omega] = 2k \bmod n$ for some integer $k$.

subgroups of internal symmetries participating in LSM anomalies delivers dual theories with novel symmetry structures that may involve higher groups or even non-invertible symmetries [41] mixing the dual crystalline and dual internal symmetries. Furthermore, higher-dimensional space accommodates dualities between phase diagrams that support phases that are not allowed when space is one dimensional, namely phases supporting symmetry-enriched (anomalous) topological order or ordered phases with local order parameters that break spontaneously a continuous symmetry group. [51]. Another avenue is to construct fermionic models that support novel deconfined phase transitions by gauging sub-symmetries.

## Acknowledgments

We thank Lakshya Bhardwaj, Lea Bottini, Heidar Moradi and Sakura Schafer Nameki for several useful discussions.

**Funding information** ÖMA is supported by the Swiss National Science Foundation (SNSF) under Grant No. 200021 184637. AT is supported by the Swedish Research Council (VR) through grants number 2019-04736 and 2020-00214. AF is supported by JSPS KAKENHI (Grant No. JP19K03680), JST CREST (Grant No. JPMJCR19T2), and the National Science Foundation (Grant No. NSF PHY-1748958).

## A   Triality of the Majumdar-Ghosh line

### A.1   Definition and properties of the Majumdar-Ghosh line

To study the Majumdar-Ghosh (MG) line (4.4c), we start from the fully SU(2)-symmetric Hamiltonian

$$
\begin{aligned}
\widehat{H}_{\mathrm{SU(2)}} &:= \sum_{j=1}^{2N}\left(\hat{\boldsymbol{\sigma}}_j \cdot \hat{\boldsymbol{\sigma}}_{j+1} + \frac{3}{2}\widehat{\mathbb{1}}_{\mathcal{H}_{b=0}}\right) + \frac{1}{2}\sum_{j=1}^{2N}\hat{\boldsymbol{\sigma}}_j \cdot \hat{\boldsymbol{\sigma}}_{j+2} \\
&= \sum_{j=1}^{2N}\left[\frac{1}{4}\left(\hat{\boldsymbol{\sigma}}_j + \hat{\boldsymbol{\sigma}}_{j+1} + \hat{\boldsymbol{\sigma}}_{j+2}\right)^2 - \frac{3}{4}\widehat{\mathbb{1}}_{\mathcal{H}_{b=0}}\right],
\end{aligned}
\tag{A.1}
$$

where periodic boundary conditions ($b = 0$) have been imposed.

We shall denote with

$$
|[j, j+1]\rangle := \frac{1}{\sqrt{2}}\left(|\rightarrow\rangle_j \otimes |\leftarrow\rangle_{j+1} - |\leftarrow\rangle_j \otimes |\rightarrow\rangle_{j+1}\right),
\tag{A.2}
$$

the singlet state for two spin-1/2 localized on two consecutive sites $j$ and $j+1$ of $\Lambda$ in the basis for which $|\rightarrow\rangle_j$ ($|\uparrow\rangle_j$) is the eigenstate with eigenvalue $+1$ of $\hat{\sigma}_j^x$ ($\hat{\sigma}_j^z$). By inspection, the states

$$
|\mathrm{Dimer_o}\rangle := \bigotimes_{j=1}^{N}|[2j-1, 2j]\rangle,
\tag{A.3a}
$$

and

$$
|\mathrm{Dimer_e}\rangle := \bigotimes_{j=1}^{N}|[2j, 2j+1]\rangle,
\tag{A.3b}
$$

are orthonormal eigenstates of $\widehat{H}_{\mathrm{SU(2)}}$ with the degenerate eigenvalue

$$
E_{\mathrm{Dimer}} = 0.
\tag{A.3c}
$$

Their bonding and anti-bonding linear combinations are defined by

$$|\text{Dimer}\rangle^{\pm} := \frac{1}{\sqrt{2}}\left(|\text{Dimer}_{\text{o}}\rangle \pm |\text{Dimer}_{\text{e}}\rangle\right). \tag{A.3d}$$

Since the square bracket on the right-hand side of the second equality in Eq. (A.1) is positive definite, this energy is that of the ground state. Shastry and Sutherland have shown that these are the gapped ground states of Hamiltonian (4.1) along the MG line (4.4c).

The projectors onto the two dimer states are

$$\widehat{P}^{\text{o}}_{\text{Dimer}} := \prod_{j=1}^{N} \widehat{P}_{[2j-1,2j]}, \tag{A.4a}$$

and

$$\widehat{P}^{\text{e}}_{\text{Dimer}} := \prod_{j=1}^{N} \widehat{P}_{[2j,2j+1]}, \tag{A.4b}$$

where

$$\begin{aligned}\widehat{P}_{[j,j+1]} &:= \frac{1}{4}\left(\widehat{\mathbb{1}}_{\mathcal{H}_{b=0}} - \hat{\boldsymbol{\sigma}}_j \cdot \hat{\boldsymbol{\sigma}}_{j+1}\right)\\ &= \frac{1}{4}\left\{\widehat{\mathbb{1}}_{\mathcal{H}_{b=0}} - \frac{1}{2}\left[\left(\hat{\boldsymbol{\sigma}}_j + \hat{\boldsymbol{\sigma}}_{j+1}\right)^2 - 6\widehat{\mathbb{1}}_{\mathcal{H}_{b=0}}\right]\right\}.\end{aligned} \tag{A.4c}$$

We have the transformation laws

$$\widehat{U}_t\,\widehat{P}^{\text{o}}_{\text{Dimer}}\left(\widehat{U}_t\right)^{\dagger} = \prod_{j=1}^{N}\left[\widehat{U}_t\,\widehat{P}_{[2j-1,2j]}\left(\widehat{U}_t\right)^{\dagger}\right] = \widehat{P}^{\text{e}}_{\text{Dimer}}, \tag{A.5a}$$

$$\widehat{U}_r\,\widehat{P}^{\text{o}}_{\text{Dimer}}\,\widehat{U}_r = \prod_{j=1}^{N}\left(\widehat{U}_r\,\widehat{P}_{[2j-1,2j]}\,\widehat{U}_r\right) = \widehat{P}^{\text{e}}_{\text{Dimer}}, \tag{A.5b}$$

$$\widehat{U}_{r^x_\pi}\,\widehat{P}^{\text{o}}_{\text{Dimer}}\,\widehat{U}_{r^x_\pi} = \prod_{j=1}^{N}\left(\widehat{U}_{r^x_\pi}\,\widehat{P}_{[2j-1,2j]}\,\widehat{U}_{r^x_\pi}\right) = \widehat{P}^{\text{o}}_{\text{Dimer}}, \tag{A.5c}$$

$$\widehat{U}_{r^z_\pi}\,\widehat{P}^{\text{o}}_{\text{Dimer}}\,\widehat{U}_{r^z_\pi} = \prod_{j=1}^{N}\left(\widehat{U}_{r^z_\pi}\,\widehat{P}_{[2j-1,2j]}\,\widehat{U}_{r^z_\pi}\right) = \widehat{P}^{\text{o}}_{\text{Dimer}}, \tag{A.5d}$$

and

$$\widehat{U}_t\,\widehat{P}^{\text{e}}_{\text{Dimer}}\left(\widehat{U}_t\right)^{\dagger} = \prod_{j=1}^{N}\left[\widehat{U}_t\,\widehat{P}_{[2j,2j+1]}\left(\widehat{U}_t\right)^{\dagger}\right] = \widehat{P}^{\text{o}}_{\text{Dimer}}, \tag{A.5e}$$

$$\widehat{U}_r\,\widehat{P}^{\text{e}}_{\text{Dimer}}\,\widehat{U}_r = \prod_{j=1}^{N}\left(\widehat{U}_r\,\widehat{P}_{[2j,2j+1]}\,\widehat{U}_r\right) = \widehat{P}^{\text{o}}_{\text{Dimer}}, \tag{A.5f}$$

$$\widehat{U}_{r^x_\pi}\,\widehat{P}^{\text{e}}_{\text{Dimer}}\,\widehat{U}_{r^x_\pi} = \prod_{j=1}^{2N}\left(\widehat{U}_{r^x_\pi}\,\widehat{P}_{[2j,2j+1]}\,\widehat{U}_{r^x_\pi}\right) = \widehat{P}^{\text{e}}_{\text{Dimer}}, \tag{A.5g}$$

$$\widehat{U}_{r^z_\pi}\,\widehat{P}^{\text{e}}_{\text{Dimer}}\,\widehat{U}_{r^z_\pi} = \prod_{j=1}^{2N}\left(\widehat{U}_{r^z_\pi}\,\widehat{P}_{[2j,2j+1]}\,\widehat{U}_{r^z_\pi}\right) = \widehat{P}^{\text{e}}_{\text{Dimer}}. \tag{A.5h}$$

In words, both translation $t$ and reflection $r$ with the fixed points $N$ and $2N$ interchange $|\text{Dimer}_{\text{o}}\rangle$ and $|\text{Dimer}_{\text{e}}\rangle$. Observe that both rotations $r^x_\pi$ and $r^z_\pi$ map $|\text{Dimer}_{\text{o}}\rangle$ to $(-1)^N\,|\text{Dimer}_{\text{o}}\rangle$

(they do the same for $|\mathrm{Dimer_e}\rangle$),

$$\widehat{U}_{r_\pi^x}|\mathrm{Dimer_o}\rangle = (-1)^N |\mathrm{Dimer_o}\rangle, \qquad \widehat{U}_{r_\pi^z}|\mathrm{Dimer_o}\rangle = (-1)^N |\mathrm{Dimer_o}\rangle, \tag{A.6a}$$

$$\widehat{U}_{r_\pi^x}|\mathrm{Dimer_e}\rangle = (-1)^N |\mathrm{Dimer_e}\rangle, \qquad \widehat{U}_{r_\pi^z}|\mathrm{Dimer_e}\rangle = (-1)^N |\mathrm{Dimer_e}\rangle. \tag{A.6b}$$

The multiplicative phase factor $(-1)^N$ cancels in either one of the projectors $\widehat{P}_{\mathrm{Dimer}}^{\mathrm{o}}$ and $\widehat{P}_{\mathrm{Dimer}}^{\mathrm{e}}$.

It is instructive to compare the dimer states (A.3) with the Neel states

$$\begin{aligned}|\mathrm{Neel}_{\mathrm{o}}^x\rangle &:= |\rightarrow\rangle_1 \otimes |\leftarrow\rangle_2 \otimes \cdots \otimes |\rightarrow\rangle_{2N-1} \otimes |\leftarrow\rangle_{2N} \\ &\equiv |\rightarrow, \leftarrow, \cdots, \rightarrow, \leftarrow\rangle,\end{aligned} \tag{A.7a}$$

and

$$\begin{aligned}|\mathrm{Neel}_{\mathrm{e}}^x\rangle &:= \widehat{U}_{r_\pi^z}|\mathrm{Neel}_{\mathrm{o}}^x\rangle \\ &\equiv |\rightarrow, \leftarrow, \cdots, \rightarrow, \leftarrow\rangle.\end{aligned} \tag{A.7b}$$

The pair of Neel states (A.7) are the two-fold degenerate gapped ground states of Hamiltonian (4.1) at the lower left corner in the reduced coupling space (4.3).

The projectors onto the Neel states are

$$\widehat{P}_{\mathrm{Neel}^x}^{\mathrm{o}} = \prod_{j=1}^{2N} \frac{1}{2}\left[\widehat{\mathbb{1}}_{\mathcal{H}_{b=0}} + (-1)^{j+1}\, \hat{\sigma}_j^x\right], \tag{A.8a}$$

and

$$\begin{aligned}\widehat{P}_{\mathrm{Neel}^x}^{\mathrm{e}} &= \widehat{U}_{r_\pi^z}\, \widehat{P}_{\mathrm{Neel}^x}^{\mathrm{o}}\, \widehat{U}_{r_\pi^z} \\ &= \prod_{j=1}^{2N} \frac{1}{2}\left[\widehat{\mathbb{1}}_{\mathcal{H}_{b=0}} - (-1)^{j+1}\, \hat{\sigma}_j^x\right],\end{aligned} \tag{A.8b}$$

respectively. The Neel projectors corresponding to the orthonormal pair of bonding and anti-bonding linear combinations

$$|\mathrm{Neel}^x\rangle^{\pm} := \frac{1}{\sqrt{2}}\left(|\mathrm{Neel}_{\mathrm{o}}^x\rangle \pm |\mathrm{Neel}_{\mathrm{e}}^x\rangle\right), \tag{A.9a}$$

are

$$\widehat{P}_{\mathrm{Neel}^x}^{\pm} := \frac{1 \pm \widehat{U}_{r_\pi^z}}{2}\left(\widehat{P}_{\mathrm{Neel}^x}^{\mathrm{o}} + \widehat{P}_{\mathrm{Neel}^x}^{\mathrm{e}}\right). \tag{A.9b}$$

The projector $\widehat{P}_{\mathrm{Neel}^x}^{+}$ is a linear combination of string of $\hat{\sigma}^x$'s of even length. The projector $\widehat{P}_{\mathrm{Neel}^x}^{-}$ is a linear combination of string of $\hat{\sigma}^x$'s of odd length.

We have the transformation laws

$$\widehat{U}_t\, \widehat{P}_{\mathrm{Neel}^x}^{\mathrm{o}}\, \left(\widehat{U}_t\right)^\dagger = \widehat{P}_{\mathrm{Neel}^x}^{\mathrm{e}}, \tag{A.10a}$$

$$\widehat{U}_r\, \widehat{P}_{\mathrm{Neel}^x}^{\mathrm{o}}\, \widehat{U}_r = \widehat{P}_{\mathrm{Neel}^x}^{\mathrm{o}}, \tag{A.10b}$$

$$\widehat{U}_{r_\pi^x}\, \widehat{P}_{\mathrm{Neel}^x}^{\mathrm{o}}\, \widehat{U}_{r_\pi^x} = \widehat{P}_{\mathrm{Neel}^x}^{\mathrm{o}}, \tag{A.10c}$$

$$\widehat{U}_{r_\pi^z}\, \widehat{P}_{\mathrm{Neel}^x}^{\mathrm{o}}\, \widehat{U}_{r_\pi^z} = \widehat{P}_{\mathrm{Neel}^x}^{\mathrm{o}}, \tag{A.10d}$$

and

$$\widehat{U}_t\, \widehat{P}_{\mathrm{Neel}^x}^{\mathrm{e}}\, \left(\widehat{U}_t\right)^\dagger = \widehat{P}_{\mathrm{Neel}^x}^{\mathrm{o}}, \tag{A.10e}$$

$$\widehat{U}_r\, \widehat{P}_{\mathrm{Neel}^x}^{\mathrm{e}}\, \widehat{U}_r = \widehat{P}_{\mathrm{Neel}^x}^{\mathrm{e}}, \tag{A.10f}$$

$$\widehat{U}_{r_\pi^x}\, \widehat{P}_{\mathrm{Neel}^x}^{\mathrm{e}}\, \widehat{U}_{r_\pi^x} = \widehat{P}_{\mathrm{Neel}^x}^{\mathrm{e}}, \tag{A.10g}$$

$$\widehat{U}_{r_\pi^z}\, \widehat{P}_{\mathrm{Neel}^x}^{\mathrm{e}}\, \widehat{U}_{r_\pi^z} = \widehat{P}_{\mathrm{Neel}^x}^{\mathrm{e}}. \tag{A.10h}$$

In words, translation $t$ interchanges $|\text{Neel}^x_\text{o}\rangle$ and $|\text{Neel}^x_\text{e}\rangle$. Reflection $r$ with the fixed points $N$ and $2N$ in $\Lambda$ leaves each Neel state unchanged. The same is true of rotation $r^x_\pi$. Observe that rotation $r^z_\pi$ maps $|\text{Neel}^x_\text{o}\rangle$ to $|\text{Neel}^x_\text{o}\rangle$ (it does the same for $|\text{Neel}^x_\text{e}\rangle$),

$$\widehat{U}_{r^x_\pi} |\text{Neel}^x_\text{o}\rangle = (-1)^N |\text{Neel}^x_\text{o}\rangle, \qquad \widehat{U}_{r^z_\pi} |\text{Neel}^x_\text{o}\rangle = |\text{Neel}^x_\text{o}\rangle, \tag{A.11a}$$

$$\widehat{U}_{r^x_\pi} |\text{Neel}^x_\text{e}\rangle = (-1)^N |\text{Neel}^x_\text{e}\rangle, \qquad \widehat{U}_{r^z_\pi} |\text{Neel}^x_\text{e}\rangle = |\text{Neel}^x_\text{e}\rangle. \tag{A.11b}$$

The multiplicative phase factor $(-1)^N$ cancels in either one of the projectors $\widehat{P}^\text{o}_{\text{Neel}^x}$ and $\widehat{P}^\text{e}_{\text{Neel}^x}$.

Define the local order parameters

$$\widehat{O}^\text{o}_{\text{Neel}^\alpha} := \frac{1}{2N} \sum_{j=1}^{2N} (-1)^{j+1} \hat{\sigma}^\alpha_j, \qquad \alpha = x, y, z, \tag{A.12a}$$

$$\widehat{O}^\text{o}_{\text{dimer}} := \frac{1}{N} \sum_{j=1}^{2N} (-1)^j \frac{1}{3} \hat{\boldsymbol{\sigma}}_j \cdot \hat{\boldsymbol{\sigma}}_{j+1}. \tag{A.12b}$$

Define the two-point operator

$$\widehat{C}^\alpha_{j,j+n} := \hat{\sigma}^\alpha_j \hat{\sigma}^\alpha_{j+n}, \qquad \alpha = x, y, z. \tag{A.13}$$

We will replace the staggered magnetization (A.12a) with the two-point operator (A.13) to detect Neel order as the former cannot be dualized when $\alpha = x, y$. We recall the definition of the unitary operator

$$\widehat{U}_r := \prod_{j=1}^{N-1} \frac{1}{2} \left( \widehat{\mathbb{1}}_{\mathcal{H}_{b=0}} + \hat{\boldsymbol{\sigma}}_j \cdot \hat{\boldsymbol{\sigma}}_{r(j)} \right), \tag{A.14a}$$

that implements reflection with the fixed points $N$ and $2N$ (the upper bound is $N-1$ in the product because the two fixed points $N$ and $2N$ must be removed from the product) and the unitary operators

$$\widehat{U}_{r^\alpha_\pi} := \prod_{j=1}^{2N} \hat{\sigma}^\alpha_j, \qquad \alpha = x, y, z, \tag{A.14b}$$

$$\widehat{U}^{\text{o}(2n)}_{r^\alpha_\pi} := \prod_{k=2j-1}^{2j-1+2n-1} \hat{\sigma}^\alpha_k, \qquad \alpha = x, y, z, \qquad j = 1, \cdots, 2N, \qquad n = 1, \cdots, N, \tag{A.14c}$$

$$\widehat{U}^{\text{e}(2n)}_{r^\alpha_\pi} := \prod_{k=2j}^{2j+2n-1} \hat{\sigma}^\alpha_k, \qquad \alpha = x, y, z, \qquad j = 1, \cdots, 2N, \qquad n = 1, \cdots, N, \tag{A.14d}$$

that implement rotations by $\pi$ around the $\alpha$ axis in the Bloch spheres labeled by the lattice sites $j = 1, \cdots, 2N$ on strings of consecutive $2n$ lattice sites. Their expectation values in the four Neel and four Dimer states are given in Table 8. Observe that of the eight states in Table 8, only six are eigenstates of $\widehat{U}_{r^z_\pi}$, namely

$$|\text{Neel}^x\rangle^+, \qquad |\text{Neel}^x\rangle^-, \qquad |\text{Dimer}_\text{o}\rangle, \qquad |\text{Dimer}_\text{e}\rangle, \qquad |\text{Dimer}\rangle^+, \qquad |\text{Dimer}\rangle^-. \tag{A.15}$$

Moreover, we can distinguish $|\text{Dimer}_\text{o}\rangle$ and $|\text{Dimer}_\text{e}\rangle$ from $|\text{Dimer}\rangle^+$ and $|\text{Dimer}\rangle^-$ by using the fact that the two elements of the first pair are interchanged by reflection about the lattice site $N$, while the two elements of the second pair transform like the eigenstates of reflection about the lattice site $N$.

Table 8: The expectation values of nine operators in eight states. The domain of definition of all nine operators is $\mathcal{H}_{b=0}$. The first four Neel states are defined in Eqs. (A.7) and (A.9). The next four dimer states are defined in Eqs. (A.3) All nine operators are defined in Eqs. (A.12), (A.13), and (A.14).

| | $\widehat{C}^x_{j,j+n}$ | $\widehat{O}^o_{\mathrm{dimer}}$ | $\widehat{U}_r$ | $\widehat{U}_{r^x_\pi}$ | $\widehat{U}_{r^z_\pi}$ | $\widehat{U}^{o(2n)}_{r^x_\pi}$ | $\widehat{U}^{e(2n)}_{r^x_\pi}$ | $\widehat{U}^{o(2n)}_{r^z_\pi}$ | $\widehat{U}^{e(2n)}_{r^z_\pi}$ |
|---|---|---|---|---|---|---|---|---|---|
| $\lvert\mathrm{Neel}^x_o\rangle$ | $(-1)^n$ | $0$ | $+1$ | $(-1)^N$ | $0$ | $(-1)^n$ | $(-1)^n$ | $0$ | $0$ |
| $\lvert\mathrm{Neel}^x_e\rangle$ | $(-1)^n$ | $0$ | $+1$ | $(-1)^N$ | $0$ | $(-1)^n$ | $(-1)^n$ | $0$ | $0$ |
| $\lvert\mathrm{Neel}^x\rangle^+$ | $(-1)^n$ | $0$ | $+1$ | $(-1)^N$ | $+1$ | $(-1)^n$ | $(-1)^n$ | $0$ | $0$ |
| $\lvert\mathrm{Neel}^x\rangle^-$ | $(-1)^n$ | $0$ | $+1$ | $(-1)^N$ | $-1$ | $(-1)^n$ | $(-1)^n$ | $0$ | $0$ |
| $\lvert\mathrm{Dimer}_o\rangle$ | $-\delta_{(-1)^j,-1}\,\delta_{n,1}$ | $+1$ | $0$ | $(-1)^N$ | $(-1)^N$ | $(-1)^n$ | $0$ | $(-1)^n$ | $0$ |
| $\lvert\mathrm{Dimer}_e\rangle$ | $-\delta_{(-1)^j,+1}\,\delta_{n,1}$ | $-1$ | $0$ | $(-1)^N$ | $(-1)^N$ | $0$ | $(-1)^n$ | $0$ | $(-1)^n$ |
| $\lvert\mathrm{Dimer}\rangle^+$ | $-\frac{\delta_{n,1}}{2}$ | $0$ | $+(-1)^N$ | $(-1)^N$ | $(-1)^N$ | $\frac{(-1)^n}{2}$ | $\frac{(-1)^n}{2}$ | $\frac{(-1)^n}{2}$ | $\frac{(-1)^n}{2}$ |
| $\lvert\mathrm{Dimer}\rangle^-$ | $-\frac{\delta_{n,1}}{2}$ | $0$ | $-(-1)^N$ | $(-1)^N$ | $(-1)^N$ | $\frac{(-1)^n}{2}$ | $\frac{(-1)^n}{2}$ | $\frac{(-1)^n}{2}$ | $\frac{(-1)^n}{2}$ |

## A.2 Kramers-Wannier dualization of the Majumdar-Ghosh line

The projectors that are the Kramers-Wannier dual to those for the pair of dimer states are built out of

$$\widehat{P}^{o\vee}_{\mathrm{Dimer}} := \prod_{j=1}^{2N} \widehat{P}^\vee_{[(2j-1)^\star,(2j)^\star]}, \tag{A.16a}$$

and

$$\widehat{P}^{e\vee}_{\mathrm{Dimer}} := \prod_{j=1}^{2N} \widehat{P}^\vee_{[(2j)^\star,(2j+1)^\star]}, \tag{A.16b}$$

where

$$\begin{aligned}
\widehat{P}^\vee_{[j^\star,j^\star+1]} &= \frac{1}{4}\left( \widehat{\mathbb{1}}_{\mathcal{H}^\vee_{b'=0}} - \hat{\tau}^{z\vee}_{j^\star} + \hat{\tau}^{x\vee}_{j^\star-1}\hat{\tau}^{z\vee}_{j^\star}\hat{\tau}^{x\vee}_{j^\star+1} - \hat{\tau}^{x\vee}_{j^\star-1}\hat{\tau}^{x\vee}_{j^\star+1} \right)\\
&= \frac{1}{2}\left( \widehat{\mathbb{1}}_{\mathcal{H}^\vee_{b'=0}} - \hat{\tau}^{z\vee}_{j^\star} \right)\frac{1}{2}\left( \widehat{\mathbb{1}}_{\mathcal{H}^\vee_{b'=0}} - \hat{\tau}^{x\vee}_{j^\star-1}\hat{\tau}^{x\vee}_{j^\star+1} \right),
\end{aligned} \tag{A.16c}$$

by restriction to the subspace $\mathcal{H}^\vee_{b'=0}$ from Table 1. As a consequence of the fact that each of $\widehat{P}^\vee_{[2j-1,2j]}$ and $\widehat{P}^\vee_{[2j,2j+1]}$ acts non-trivially on three consecutive sites, we are going to show that each of projectors $\widehat{P}^{o\vee}_{\mathrm{Dimer}}$ and $\widehat{P}^{e\vee}_{\mathrm{Dimer}}$ has two degenerate orthonormal eigenstates with eigenvalue one.

The projector

$$\widehat{P}^{o\vee}_{\mathrm{Dimer}} := \prod_{j=1}^{2N} \widehat{P}^\vee_{[(2j-1)^\star,(2j)^\star]}, \tag{A.17a}$$

has the degenerate pair

$$\lvert 1\rangle = \lvert \downarrow,\rightarrow,\downarrow,\leftarrow;\cdots;\downarrow,\rightarrow,\downarrow,\leftarrow;\downarrow,\rightarrow,\downarrow,\leftarrow;\cdots\downarrow,\rightarrow,\downarrow,\leftarrow\rangle, \tag{A.17b}$$

and

$$\begin{aligned}
\lvert 2\rangle &= \lvert \downarrow,\leftarrow,\downarrow,\rightarrow;\cdots;\downarrow,\leftarrow,\downarrow,\rightarrow;\downarrow,\leftarrow,\downarrow,\rightarrow;\cdots;\downarrow,\leftarrow,\downarrow,\rightarrow\rangle\\
&= \widehat{U}^\vee_{r^z_\pi}\lvert 1\rangle,
\end{aligned} \tag{A.17c}$$

of orthonormal eigenstates with eigenvalue one. The projector

$$\widehat{P}^{e\vee}_{\mathrm{Dimer}} := \prod_{j=1}^{2N} \widehat{P}^\vee_{[(2j)^\star,(2j+1)^\star]}, \tag{A.18a}$$

has the degenerate pair

$$
\begin{aligned}
|3\rangle &= |\rightarrow, \downarrow, \leftarrow, \downarrow; \cdots; \rightarrow, \downarrow, \leftarrow, \downarrow; \rightarrow, \downarrow, \leftarrow, \downarrow; \cdots; \rightarrow, \downarrow, \leftarrow, \downarrow\rangle \\
&= \widehat{U}_r^{\vee} |2\rangle,
\end{aligned}
\tag{A.18b}
$$

and

$$
\begin{aligned}
|4\rangle &= |\leftarrow, \downarrow, \rightarrow, \downarrow; \cdots; \leftarrow, \downarrow, \rightarrow, \downarrow; \leftarrow, \downarrow, \rightarrow, \downarrow; \cdots; \leftarrow, \downarrow, \rightarrow, \downarrow\rangle \\
&= \widehat{U}_{r_\pi^z}^{\vee} |3\rangle \\
&= \widehat{U}_r^{\vee} |1\rangle,
\end{aligned}
\tag{A.18c}
$$

of orthonormal eigenstates with eigenvalue one. Hence, the dual of the dimer phase when periodic boundary conditions ($b = 0$) apply has the two degenerate and orthonormal ground states along the MG line

$$
|\mathrm{Dimer}_o^{\vee}\rangle^+ = \frac{1}{\sqrt{2}} \left( |1\rangle + |2\rangle \right),
\tag{A.19a}
$$

and

$$
|\mathrm{Dimer}_e^{\vee}\rangle^+ = \frac{1}{\sqrt{2}} \left( |3\rangle + |4\rangle \right).
\tag{A.19b}
$$

The two degenerate and orthonormal states

$$
|\mathrm{Dimer}_o^{\vee}\rangle^- = \frac{1}{\sqrt{2}} \left( |1\rangle - |2\rangle \right),
\tag{A.20a}
$$

and

$$
|\mathrm{Dimer}_e^{\vee}\rangle^- = \frac{1}{\sqrt{2}} \left( |3\rangle - |4\rangle \right),
\tag{A.20b}
$$

are the ground states along the MG line when twisted boundary conditions ($b' = 1$) apply. Observe that

$$
\widehat{U}_r^{\vee} |\mathrm{Dimer}_o^{\vee}\rangle^+ = +|\mathrm{Dimer}_e^{\vee}\rangle^+,
\tag{A.21a}
$$

$$
\widehat{U}_r^{\vee} |\mathrm{Dimer}_o^{\vee}\rangle^- = -|\mathrm{Dimer}_e^{\vee}\rangle^-.
\tag{A.21b}
$$

We can dualize all operators entering Eqs. (A.12), (A.13), and (A.14) except for $\hat{\sigma}_j^x$ and $\hat{\sigma}_j^y$. The dimer order parameter (A.12b) dualizes to

$$
\widehat{O}_{\mathrm{dimer}}^{o\,\vee} = \frac{1}{N} \sum_{j^\star \in \Lambda^\star} (-1)^{j^\star - 1/2} \frac{1}{3} \left( \hat{\tau}_{j^\star}^{z\,\vee} - \hat{\tau}_{j^\star - 1}^{x\,\vee} \hat{\tau}_{j^\star}^{z\,\vee} \hat{\tau}_{j^\star + 1}^{x\,\vee} + \hat{\tau}_{j^\star - 1}^{x\,\vee} \hat{\tau}_{j^\star + 1}^{x\,\vee} \right).
\tag{A.22}
$$

The $xx$ two-point operator (A.13) dualizes to the string operator made of $n$ consecutive sites from the dual lattice given by

$$
\widehat{C}_{j^\star, j^\star + n}^{x\,\vee} = \prod_{k=1}^{n} \hat{\tau}_{j^\star + (k-1)}^{z\,\vee}.
\tag{A.23}
$$

The reflection with no fixed point on the dual lattice dualizes to

$$
\widehat{U}_r^{\vee} = \prod_{j=1}^{N} \frac{1}{2} \left( \widehat{\mathbb{1}}_{\mathcal{H}_{b'=0}^{\vee}} + \hat{\tau}_{j^\star}^{x\,\vee} \hat{\tau}_{r(j^\star)}^{x\,\vee} + \hat{\tau}_{j^\star}^{y\,\vee} \hat{\tau}_{r(j^\star)}^{y\,\vee} + \hat{\tau}_{j^\star}^{z\,\vee} \hat{\tau}_{r(j^\star)}^{z\,\vee} \right)
\tag{A.24}
$$

(the upper bound is now $N$ in the product instead of $N - 1$ because there are no invariant dual lattice points under reflection, i.e., we need not remove the invariant fixed points). We

choose to dualize the global rotation by $\pi$ around the $x$ and $y$ axis of the Bloch spheres labeled by $j = 1, \cdots, 2N$ to the rotation by $\pi$ around the $z$ axis of the Bloch spheres labeled by $j^\star = 1 + \frac{1}{2}, 3 + \frac{1}{2}, \cdots, 2N - 3 + \frac{1}{2}, 2N - 1 + \frac{1}{2}$ and $j^\star = 2 + \frac{1}{2}, 4 + \frac{1}{2}, \cdots, 2N - 2 + \frac{1}{2}, 2N + \frac{1}{2}$, respectively, i.e., by

$$\widehat{U}_{r_\pi^x}^\vee = \prod_{j=1}^{N} \widehat{\tau}_{2j-1+\frac{1}{2}}^{z\,\vee} \equiv \widehat{U}_{\mathrm{o}}^\vee, \tag{A.25}$$

and

$$\widehat{U}_{r_\pi^y}^\vee = \prod_{j=1}^{N} \widehat{\tau}_{2j+\frac{1}{2}}^{z\,\vee} \equiv \widehat{U}_{\mathrm{e}}^\vee, \tag{A.26}$$

respectively.

The global rotation by $\pi$ around the $z$ axis of the Bloch spheres labeled by $j = 1, \cdots, 2N$ dualizes to the identity

$$\widehat{U}_{r_\pi^z}^\vee = \widehat{\mathbb{1}}_{\mathcal{H}_{b'=0}^\vee}. \tag{A.27}$$

The rotation by $\pi$ around the $x$ axis of the Bloch spheres labeled by $j = 1, \cdots, 2N$ on a string of $2n$ consecutive sites from the lattice $\Lambda$ starting from an odd site dualizes to the rotation by $\pi$ around the $z$ axis of the Bloch spheres labeled by $j^\star = 1 + \frac{1}{2}, \cdots, 2N + \frac{1}{2}$ on a string of $n$ consecutive odd sites from the dual lattice starting from an odd dual site $\Lambda^\star$, i.e., by

$$\widehat{U}_{r_\pi^x}^{\mathrm{o}(2n)\vee} = \prod_{k=1}^{n} \widehat{\tau}_{2j-1+2(k-1)+\frac{1}{2}}^{z\,\vee} \equiv \widehat{U}_n^{\mathrm{o}\vee}, \qquad j = 1, \cdots, 2N, \qquad n = 1, \cdots, N. \tag{A.28}$$

The rotation by $\pi$ around the $x$ axis of the Bloch spheres labeled by $j = 1, \cdots, 2N$ on a string of $2n$ consecutive sites from the lattice $\Lambda$ starting from an even site dualizes to the rotation by $\pi$ around the $z$ axis of the Bloch spheres labeled by $j^\star = 1 + \frac{1}{2}, \cdots, 2N + \frac{1}{2}$ on a string of $n$ consecutive even sites from the dual lattice $\Lambda^\star$ starting from an even dual site, i.e., by

$$\widehat{U}_{r_\pi^x}^{\mathrm{e}(2n)\vee} = \prod_{k=1}^{n} \widehat{\tau}_{2j+2(k-1)+\frac{1}{2}}^{z\,\vee} \equiv \widehat{U}_n^{\mathrm{e}\vee}, \qquad j = 1, \cdots, 2N, \qquad n = 1, \cdots, N. \tag{A.29}$$

The rotation by $\pi$ around the $z$ axis of the Bloch spheres labeled by $j = 1, \cdots, 2N$ on a string of $2n$ consecutive sites from the lattice $\Lambda$ starting from an odd site dualizes to the rotation by $\pi$ around the $x$ axis of the Bloch spheres labeled by $j^\star = 1 + \frac{1}{2}, \cdots, 2N + \frac{1}{2}$ on the two end points of a string of $2n + 1$ consecutive sites from the dual lattice $\Lambda^\star$ starting from an even dual site, i.e., by

$$\widehat{U}_{r_\pi^z}^{\mathrm{o}(2n)\vee} = \widehat{\tau}_{2j-1+\frac{1}{2}}^{x\,\vee} \widehat{\tau}_{2j-1+2n+\frac{1}{2}}^{x\,\vee}, \qquad j = 1, \cdots, 2N, \qquad n = 1, \cdots, N. \tag{A.30}$$

The rotation by $\pi$ around the $x$ axis of the Bloch spheres labeled by $j = 1, \cdots, 2N$ on a string of $2n$ consecutive sites from the lattice $\Lambda$ starting from an even site dualize to the rotation by $\pi$ around the $z$ axis of the Bloch spheres labeled by $j^\star = 1 + \frac{1}{2}, \cdots, 2N + \frac{1}{2}$ on the two end points of a string of $2n + 1$ consecutive sites from the dual lattice $\Lambda^\star$ starting from an odd dual site, i.e., by

$$\widehat{U}_{r_\pi^z}^{\mathrm{e}(2n)\vee} = \widehat{\tau}_{2j-1+\frac{1}{2}}^{x\,\vee} \widehat{\tau}_{2j+2n-1+\frac{1}{2}}^{x\,\vee}, \qquad j = 1, \cdots, 2N, \qquad n = 1, \cdots, N. \tag{A.31}$$

We seek the duals of the states

$$|\mathrm{Neel}^x\rangle^+, \qquad |\mathrm{Dimer_o}\rangle, \qquad |\mathrm{Dimer_e}\rangle, \tag{A.32}$$

Table 9: The expectation values of nine operators in three states. The domain of definition of all nine operators is $\mathcal{H}^{\vee}_{b'=0}$. The Neel state is defined in Eq. (A.33). The next two dimer states are defined in Eq. (A.19). All nine operators are defined in Eqs. (A.22)-(A.31).

| | $\widehat{C}^{x\vee}_{j,j+n}$ | $\widehat{O}^{o\vee}_{\text{dimer}}$ | $\widehat{U}^{\vee}_{r}$ | $\widehat{U}^{\vee}_{r^x_\pi}$ | $\widehat{U}^{\vee}_{r^z_\pi}$ | $\widehat{U}^{o(2n)\vee}_{r^x_\pi}$ | $\widehat{U}^{e(2n)\vee}_{r^x_\pi}$ | $\widehat{U}^{o(2n)\vee}_{r^z_\pi}$ | $\widehat{U}^{e(2n)\vee}_{r^z_\pi}$ |
|---|---|---|---|---|---|---|---|---|---|
| $\lvert\text{Neel}^{x\vee}\rangle^+$ | $(-1)^n$ | $0$ | $+1$ | $(-1)^N$ | $+1$ | $(-1)^n$ | $(-1)^n$ | $0$ | $0$ |
| $\lvert\text{Dimer}^{\vee}_{o}\rangle$ | $-\delta_{(-1)^j,-1}\,\delta_{n,1}$ | $+1$ | $0$ | $(-1)^N$ | $(-1)^N$ | $(-1)^n$ | $0$ | $(-1)^n$ | $0$ |
| $\lvert\text{Dimer}^{\vee}_{e}\rangle$ | $-\delta_{(-1)^j,+1}\,\delta_{n,1}$ | $-1$ | $0$ | $(-1)^N$ | $(-1)^N$ | $0$ | $(-1)^n$ | $0$ | $(-1)^n$ |

that all have the eigenvalue $+1$ under the global $\pi$ rotation about the $z$ axis of the Bloch spheres labeled by $j = 1, \cdots, 2N$ and are annihilated by either $\widehat{U}^{o(2n)}_{r^x_\pi}$ or $\widehat{U}^{e(2n)}_{r^x_\pi}$ for the dimer states. These are the ground states of the dual Hamiltonian $\widehat{H}^{\vee}_{b'=0}$ defined in Eq. (4.9) with the domain of definition $\mathcal{H}^{\vee}_{b'=0;+}$ with either $J = \Delta = 0$ for the dual to $\lvert\text{Neel}^x\rangle^+$ or $J = 1/2$ for the dual to the dimer states $\lvert\text{Dimer}_o\rangle$ and $\lvert\text{Dimer}_e\rangle$ in the reduced coupling space (4.3). The ground state of the dual Hamiltonian $\widehat{H}^{\vee}_{b'=0}$ with the domain of definition $\mathcal{H}^{\vee}_{b'=0;+}$ when $J = \Delta = 0$ in the reduced coupling space (4.3) is non-degenerate and given by

$$
\begin{aligned}
\lvert\text{Neel}^{x\vee}\rangle^+ &= \lvert\downarrow\rangle_1 \otimes \cdots \otimes \lvert\downarrow\rangle_{2N} \\
&\equiv \lvert\downarrow, \cdots, \downarrow\rangle,
\end{aligned}
\tag{A.33a}
$$

where

$$
\hat{\tau}^{z\vee}_{j+\frac{1}{2}}\lvert\downarrow\rangle_{j+\frac{1}{2}} = -\lvert\downarrow\rangle_{j+\frac{1}{2}}, \qquad \hat{\tau}^{z\vee}_{j+\frac{1}{2}}\lvert\uparrow\rangle_{j+\frac{1}{2}} = +\lvert\uparrow\rangle_{j+\frac{1}{2}}, \quad j = 1, \cdots, 2N.
\tag{A.33b}
$$

The ground state of the dual Hamiltonian $\widehat{H}^{\vee}_{b'=0}$ with the domain of definition $\mathcal{H}^{\vee}_{b'=0;+}$ when $J = 1/2$ in the reduced coupling space (4.3) is two-fold degenerate with the eigenstates $\lvert\text{Dimer}^{\vee}_{o}\rangle^+$ and $\lvert\text{Dimer}^{\vee}_{e}\rangle^+$ defined in Eq. (A.19). The expectation values in the Neel and two dimer states are tabulated in Table 9. These entries agree with the corresponding ones in Table 8 (lines three, five, and six).

## A.3 Jordan-Wigner dualization of the Majumdar-Ghosh line

The projectors that are the Jordan-Wigner dual to those for the pair of dimer states are built out of

$$
\widehat{P}^{o\vee}_{\text{Dimer}} := \prod_{j=1}^{N} \widehat{P}^{\vee}_{[2j-1,2j]},
\tag{A.34a}
$$

and

$$
\widehat{P}^{e\vee}_{\text{Dimer}} := \prod_{j=1}^{N} \widehat{P}^{\vee}_{[2j,2j+1]},
\tag{A.34b}
$$

where

$$
\widehat{P}^{\vee}_{[j,j+1]} := \frac{1}{4}\left(\widehat{\mathbb{1}}_{\mathcal{H}^{\vee}_{f=1}} - i\hat{\beta}^{\vee}_j\,\hat{\alpha}^{\vee}_{j+1} - i\hat{\beta}^{\vee}_{j+1}\,\hat{\alpha}^{\vee}_j - \hat{\beta}^{\vee}_j\,\hat{\beta}^{\vee}_{j+1}\,\hat{\alpha}^{\vee}_j\,\hat{\alpha}^{\vee}_{j+1}\right),
\tag{A.34c}
$$

by restriction to the subspace $\mathcal{H}^{\vee}_{f=1}$ from Table 2. Unlike in the case of Eq. (A.16), each of $\widehat{P}^{\vee}_{[2j-1,2j]}$ and $\widehat{P}^{\vee}_{[2j,2j+1]}$ acts non-trivially on two consecutive sites. This is why each of the projectors $\widehat{P}^{o\vee}_{\text{Dimer}}$ and $\widehat{P}^{e\vee}_{\text{Dimer}}$ has a non-degenerate eigenstate with eigenvalue one.

It is instructive to trade the Majorana operators for fermionic ones. To this end, define for any $j = 1, \cdots, 2N$

$$
\hat{c}^{\vee\dagger}_j := \frac{1}{2}\left(\hat{\alpha}^{\vee}_j - i\hat{\beta}^{\vee}_j\right), \qquad \hat{c}^{\vee}_j := \frac{1}{2}\left(\hat{\alpha}^{\vee}_j + i\hat{\beta}^{\vee}_j\right),
\tag{A.35a}
$$

i.e.,

$$\hat{\alpha}_j^\vee = \hat{c}_j^\vee + \hat{c}_j^{\vee\dagger}, \qquad \hat{\beta}_j^\vee = -\mathrm{i}\left(\hat{c}_j^\vee - \hat{c}_j^{\vee\dagger}\right). \tag{A.35b}$$

There follows the identities

$$
\begin{aligned}
\mathrm{i}\hat{\beta}_j^\vee \hat{\alpha}_j^\vee &= \left(\hat{c}_j^\vee - \hat{c}_j^{\vee\dagger}\right)\left(\hat{c}_j^\vee + \hat{c}_j^{\vee\dagger}\right) \\
&= \hat{c}_j^\vee \hat{c}_j^{\vee\dagger} - \hat{c}_j^{\vee\dagger} \hat{c}_j^\vee \\
&= 1 - 2\,\hat{c}_j^{\vee\dagger} \hat{c}_j^\vee \\
&\equiv 1 - 2\,\hat{n}_j^\vee, \qquad \hat{n}_j^\vee := \hat{c}_j^{\vee\dagger} \hat{c}_j^\vee,
\end{aligned}
\tag{A.36a}
$$

$$
\begin{aligned}
\hat{\beta}_j^\vee \hat{\beta}_{j+1}^\vee \hat{\alpha}_j^\vee \hat{\alpha}_{j+1}^\vee &= \left(\mathrm{i}\hat{\beta}_j^\vee \hat{\alpha}_j^\vee\right)\left(\mathrm{i}\hat{\beta}_{j+1}^\vee \hat{\alpha}_{j+1}^\vee\right) \\
&= \left(1 - 2\,\hat{n}_j^\vee\right)\left(1 - 2\,\hat{n}_{j+1}^\vee\right),
\end{aligned}
\tag{A.36b}
$$

$$
\begin{aligned}
\mathrm{i}\hat{\beta}_j^\vee \hat{\alpha}_{j+1}^\vee &= \left(\hat{c}_j^\vee - \hat{c}_j^{\vee\dagger}\right)\left(\hat{c}_{j+1}^\vee + \hat{c}_{j+1}^{\vee\dagger}\right) \\
&= \hat{c}_j^\vee \hat{c}_{j+1}^\vee + \hat{c}_j^\vee \hat{c}_{j+1}^{\vee\dagger} - \hat{c}_j^{\vee\dagger} \hat{c}_{j+1}^\vee - \hat{c}_j^{\vee\dagger} \hat{c}_{j+1}^{\vee\dagger},
\end{aligned}
\tag{A.36c}
$$

$$
\begin{aligned}
\mathrm{i}\hat{\beta}_j^\vee \hat{\alpha}_{j+1}^\vee + \mathrm{i}\hat{\beta}_{j+1}^\vee \hat{\alpha}_j^\vee &= \hat{c}_j^\vee \hat{c}_{j+1}^\vee + \hat{c}_j^\vee \hat{c}_{j+1}^{\vee\dagger} - \hat{c}_j^{\vee\dagger} \hat{c}_{j+1}^\vee - \hat{c}_j^{\vee\dagger} \hat{c}_{j+1}^{\vee\dagger} \\
&\quad + \hat{c}_{j+1}^\vee \hat{c}_j^\vee + \hat{c}_{j+1}^\vee \hat{c}_j^{\vee\dagger} - \hat{c}_{j+1}^{\vee\dagger} \hat{c}_j^\vee - \hat{c}_{j+1}^{\vee\dagger} \hat{c}_j^{\vee\dagger} \\
&= -2\left(\hat{c}_{j+1}^{\vee\dagger} \hat{c}_j^\vee + \hat{c}_j^{\vee\dagger} \hat{c}_{j+1}^\vee\right),
\end{aligned}
\tag{A.36d}
$$

and

$$\widehat{P}_{[j,j+1]}^\vee := \frac{1}{4}\left[\widehat{\mathbb{1}}_{\mathcal{H}_{f=1}^\vee} + 2\left(\hat{c}_{j+1}^{\vee\dagger} \hat{c}_j^\vee + \hat{c}_j^{\vee\dagger} \hat{c}_{j+1}^\vee\right) - \left(1 - 2\,\hat{n}_j^\vee\right)\left(1 - 2\,\hat{n}_{j+1}^\vee\right)\right]. \tag{A.36e}$$

On the Hilbert space

$$\mathcal{H}_{j+\frac{1}{2},j+1+\frac{1}{2}} := \mathrm{span}\left\{\left(\hat{c}_j^{\vee\dagger}\right)^{n_j}\left(\hat{c}_{j+1}^{\vee\dagger}\right)^{n_{j+1}}|0,0\rangle \;\middle|\; n_j, n_{j+1} = 0,1, \hat{c}_j^\vee|0,0\rangle = \hat{c}_{j+1}^\vee|0,0\rangle = 0\right\}, \tag{A.37a}$$

$\mathrm{i}\hat{\beta}_j^\vee \hat{\alpha}_{j+1}^\vee$ is represented by the matrix

$$
\begin{pmatrix}
0 & 0 & 0 & -1 \\
0 & 0 & -1 & 0 \\
0 & -1 & 0 & 0 \\
+1 & 0 & 0 & 0
\end{pmatrix},
\tag{A.37b}
$$

while $\widehat{P}_{[j,j+1]}^\vee$ is represented by the matrix

$$
\frac{1}{4}\left[
\begin{pmatrix}
+1 & 0 & 0 & 0 \\
0 & +1 & 0 & 0 \\
0 & 0 & +1 & 0 \\
0 & 0 & 0 & +1
\end{pmatrix}
+ 2
\begin{pmatrix}
0 & 0 & 0 & 0 \\
0 & 0 & 1 & 0 \\
0 & 1 & 0 & 0 \\
0 & 0 & 0 & 0
\end{pmatrix}
-
\begin{pmatrix}
+1 & 0 & 0 & 0 \\
0 & -1 & 0 & 0 \\
0 & 0 & -1 & 0 \\
0 & 0 & 0 & +1
\end{pmatrix}
\right]
= \frac{1}{2}
\begin{pmatrix}
0 & 0 & 0 & 0 \\
0 & 1 & 1 & 0 \\
0 & 1 & 1 & 0 \\
0 & 0 & 0 & 0
\end{pmatrix},
\tag{A.37c}
$$

that annihilates the three orthonormal eigenstates

$$|0,0\rangle, \qquad \frac{1}{\sqrt{2}}\left(|1,0\rangle - |0,1\rangle\right), \qquad |1,1\rangle, \tag{A.37d}$$

and projects onto the eigenstate

$$\frac{1}{\sqrt{2}} \left( |1,0\rangle + |0,1\rangle \right). \tag{A.37e}$$

Hence, the two orthonormal states

$$|\text{Bonding}_o^\vee\rangle := \left[ \prod_{j=1}^N \frac{1}{\sqrt{2}} \left( \hat{c}_{2j-1}^{\vee\,\dagger} + \hat{c}_{2j}^{\vee\,\dagger} \right) \right] |0\rangle, \tag{A.38}$$

and

$$|\text{Bonding}_e^\vee\rangle := \left[ \prod_{j=1}^N \frac{1}{\sqrt{2}} \left( \hat{c}_{2j}^{\vee\,\dagger} + \hat{c}_{2j+1}^{\vee\,\dagger} \right) \right] |0\rangle, \tag{A.39}$$

are ground states of $\widehat{H}_{f=1}^\vee$ along the MG line. We set $N$ to be an even integer, so that these states have even fermion parity and belong to the subspace $\mathcal{H}_{f=1;+}^\vee$. Their transformation laws under the symmetry group (3.27a) are

$$\widehat{U}_t^\vee |\text{Bonding}_o^\vee\rangle = |\text{Bonding}_e^\vee\rangle, \qquad \widehat{U}_t^\vee |\text{Bonding}_e^\vee\rangle = |\text{Bonding}_o^\vee\rangle, \tag{A.40a}$$

$$\widehat{U}_r^\vee |\text{Bonding}_o^\vee\rangle = |\text{Bonding}_e^\vee\rangle, \qquad \widehat{U}_r^\vee |\text{Bonding}_e^\vee\rangle = |\text{Bonding}_o^\vee\rangle, \tag{A.40b}$$

$$\widehat{U}_o^\vee |\text{Bonding}_o^\vee\rangle = |\text{Bonding}_o^\vee\rangle, \qquad \widehat{U}_o^\vee |\text{Bonding}_e^\vee\rangle = |\text{Bonding}_e^\vee\rangle, \tag{A.40c}$$

$$\widehat{U}_e^\vee |\text{Bonding}_o^\vee\rangle = |\text{Bonding}_o^\vee\rangle, \qquad \widehat{U}_e^\vee |\text{Bonding}_e^\vee\rangle = |\text{Bonding}_e^\vee\rangle. \tag{A.40d}$$

*Proof.* Without loss of generality, we consider the case of $N = 2$. We do the substitutions

$$\hat{c}_1^{\vee\,\dagger} \to \hat{a}^\dagger, \qquad \hat{c}_2^{\vee\,\dagger} \to \hat{b}^\dagger, \qquad \hat{c}_3^{\vee\,\dagger} \to \hat{c}^\dagger, \qquad \hat{c}_4^{\vee\,\dagger} \to \hat{d}^\dagger, \tag{A.41}$$

to simplify the notation. The basis of the Hilbert space is chosen to be

$$\left. \begin{array}{l} |a,b,c,d\rangle = \left(\hat{a}^\dagger\right)^a \left(\hat{b}^\dagger\right)^b \left(\hat{c}^\dagger\right)^c \left(\hat{d}^\dagger\right)^d |0\rangle \\ \hat{a}|0\rangle = \hat{b}|0\rangle = \hat{c}|0\rangle = \hat{d}|0\rangle = 0 \end{array} \right\} a,b,c,d = 0,1. \tag{A.42}$$

In this basis,

$$|\text{Bonding}_o^\vee\rangle = \frac{1}{2} \left( |1,0,1,0\rangle + |1,0,0,1\rangle + |0,1,1,0\rangle + |0,1,0,1\rangle \right), \tag{A.43a}$$

$$|\text{Bonding}_e^\vee\rangle = \frac{1}{2} \left( |0,1,0,1\rangle + |1,1,0,0\rangle + |0,0,1,1\rangle + |1,0,1,0\rangle \right). \tag{A.43b}$$

Translation $j \mapsto j+1 \bmod 4$ by one lattice spacing corresponds to

$$\hat{a} \mapsto \hat{b}, \qquad \hat{b} \mapsto \hat{c}, \qquad \hat{c} \mapsto \hat{d}, \qquad \hat{d} \mapsto (-1)\hat{a}, \tag{A.44a}$$

under which

$$|\text{Bonding}_o^\vee\rangle \mapsto |\text{Bonding}_e^\vee\rangle, \qquad |\text{Bonding}_e^\vee\rangle \mapsto |\text{Bonding}_o^\vee\rangle. \tag{A.44b}$$

Reflection $j \mapsto 2N - j \bmod 4$ corresponds to

$$\hat{a} \mapsto -\mathrm{i}\hat{c}, \qquad \hat{b} \mapsto -\mathrm{i}\hat{b}, \qquad \hat{c} \mapsto -\mathrm{i}\hat{a}, \qquad \hat{d} \mapsto +\mathrm{i}\hat{d} \tag{A.45a}$$

(and not $\hat{a} \mapsto \hat{b}$, $\hat{b} \mapsto \hat{a}$, $\hat{c} \mapsto \hat{d}$, $\hat{d} \mapsto \hat{c}$), under which

$$|\text{Bonding}_o^\vee\rangle \mapsto |\text{Bonding}_e^\vee\rangle, \qquad |\text{Bonding}_e^\vee\rangle \mapsto |\text{Bonding}_o^\vee\rangle. \tag{A.45b}$$

The representation of

$$\widehat{U}_{\rm o}^{\vee} = \left(\widehat{U}_{\rm o}^{\vee}\right)^{\dagger}, \tag{A.46a}$$

after normal ordering is

$$\widehat{U}_{\rm o}^{\vee} = \left(\hat{a}\,\hat{b} + \hat{b}^{\dagger}\,\hat{a}^{\dagger} - \hat{b}^{\dagger}\,\hat{a} - \hat{a}^{\dagger}\,\hat{b}\right)\left(\hat{c}\,\hat{d} + \hat{d}^{\dagger}\,\hat{c}^{\dagger} - \hat{d}^{\dagger}\,\hat{c} - \hat{c}^{\dagger}\,\hat{d}\right). \tag{A.46b}$$

On the one hand,

$$
\begin{aligned}
\widehat{U}_{\rm o}^{\vee}\,|\text{Bonding}_{\rm o}^{\vee}\rangle &= \frac{1}{2}\left(\hat{a}\,\hat{b} + \hat{b}^{\dagger}\,\hat{a}^{\dagger} - \hat{b}^{\dagger}\,\hat{a} - \hat{a}^{\dagger}\,\hat{b}\right)\left(\hat{c}\,\hat{d} + \hat{d}^{\dagger}\,\hat{c}^{\dagger} - \hat{d}^{\dagger}\,\hat{c} - \hat{c}^{\dagger}\,\hat{d}\right)\\
&\quad \times \left(|1,0,1,0\rangle + |1,0,0,1\rangle + |0,1,1,0\rangle + |0,1,0,1\rangle\right)\\
&= \frac{1}{2}\left(\hat{a}\,\hat{b} + \hat{b}^{\dagger}\,\hat{a}^{\dagger} - \hat{b}^{\dagger}\,\hat{a} - \hat{a}^{\dagger}\,\hat{b}\right)\\
&\quad \times \left[-\hat{d}^{\dagger}\,\hat{c}\,\left(|1,0,1,0\rangle + |0,1,1,0\rangle\right) - \hat{c}^{\dagger}\,\hat{d}\,\left(|1,0,0,1\rangle + |0,1,0,1\rangle\right)\right]\\
&= \frac{1}{2}\left(\hat{a}\,\hat{b} + \hat{b}^{\dagger}\,\hat{a}^{\dagger} - \hat{b}^{\dagger}\,\hat{a} - \hat{a}^{\dagger}\,\hat{b}\right)\\
&\quad \times (-1)\left(|1,0,0,1\rangle + |0,1,0,1\rangle + |1,0,1,0\rangle + |0,1,1,0\rangle\right)\\
&= \frac{1}{2}\left[\hat{b}^{\dagger}\,\hat{a}\,\left(|1,0,0,1\rangle + |1,0,1,0\rangle\right) + \hat{a}^{\dagger}\,\hat{b}\,\left(|0,1,0,1\rangle + |0,1,1,0\rangle\right)\right]\\
&= \frac{1}{2}\left(|0,1,0,1\rangle + |0,1,1,0\rangle + |1,0,0,1\rangle + |1,0,1,0\rangle\right)\\
&= |\text{Bonding}_{\rm o}^{\vee}\rangle. \tag{A.47}
\end{aligned}
$$

On the other hand,

$$
\begin{aligned}
\widehat{U}_{\rm o}^{\vee}\,|\text{Bonding}_{\rm e}^{\vee}\rangle &= \frac{1}{2}\left(\hat{a}\,\hat{b} + \hat{b}^{\dagger}\,\hat{a}^{\dagger} - \hat{b}^{\dagger}\,\hat{a} - \hat{a}^{\dagger}\,\hat{b}\right)\left(\hat{c}\,\hat{d} + \hat{d}^{\dagger}\,\hat{c}^{\dagger} - \hat{d}^{\dagger}\,\hat{c} - \hat{c}^{\dagger}\,\hat{d}\right)\\
&\quad \times \left(|0,1,0,1\rangle + |1,1,0,0\rangle + |0,0,1,1\rangle + |1,0,1,0\rangle\right)\\
&= \frac{1}{2}\left(\hat{a}\,\hat{b} + \hat{b}^{\dagger}\,\hat{a}^{\dagger} - \hat{b}^{\dagger}\,\hat{a} - \hat{a}^{\dagger}\,\hat{b}\right)\\
&\quad \times \left(\hat{c}\,\hat{d}\,|0,0,1,1\rangle + \hat{d}^{\dagger}\,\hat{c}^{\dagger}\,|1,1,0,0\rangle - \hat{d}^{\dagger}\,\hat{c}\,|1,0,1,0\rangle - \hat{c}^{\dagger}\,\hat{d}\,|0,1,0,1\rangle\right)\\
&= \frac{1}{2}\left(\hat{a}\,\hat{b} + \hat{b}^{\dagger}\,\hat{a}^{\dagger} - \hat{b}^{\dagger}\,\hat{a} - \hat{a}^{\dagger}\,\hat{b}\right)\\
&\quad \times (-1)\left(|0,0,0,0\rangle + |1,1,1,1\rangle + |1,0,0,1\rangle + |0,1,1,0\rangle\right)\\
&= \frac{1}{2}\\
&\quad \times (-1)\left(\hat{a}\,\hat{b}\,|1,1,1,1\rangle + \hat{b}^{\dagger}\,\hat{a}^{\dagger}\,|0,0,0,0\rangle - \hat{b}^{\dagger}\,\hat{a}\,|1,0,0,1\rangle - \hat{a}^{\dagger}\,\hat{b}\,|0,1,1,0\rangle\right)\\
&= \frac{1}{2}\left(|0,0,1,1\rangle + |1,1,0,0\rangle + |0,1,0,1\rangle + |1,0,1,0\rangle\right)\\
&= |\text{Bonding}_{\rm e}^{\vee}\rangle. \tag{A.48}
\end{aligned}
$$

We can deduce the action of $\widehat{U}_{\rm e}^{\vee}$ on $|\text{Bonding}_{\rm o}^{\vee}\rangle$ and $|\text{Bonding}_{\rm e}^{\vee}\rangle$ from the facts that

$$\left[\widehat{U}_{\rm o}^{\vee}, \widehat{U}_{\rm e}^{\vee}\right] = 0, \tag{A.49a}$$

$$\widehat{U}_{\rm o}^{\vee}\,\widehat{U}_{\rm e}^{\vee} = \widehat{U}_{\rm e}^{\vee}\,\widehat{U}_{\rm o}^{\vee} = \widehat{P}_{\rm F}^{\vee}, \tag{A.49b}$$

$$\left(\widehat{U}_{\rm o}^{\vee}\right)^{2} = \left(\widehat{U}_{\rm e}^{\vee}\right)^{2} = \widehat{\mathbb{1}}_{\mathcal{H}_{f}^{\vee}}, \tag{A.49c}$$

$$\widehat{P}_{\rm F}^{\vee}\,|\text{Bonding}_{\rm o}^{\vee}\rangle = |\text{Bonding}_{\rm o}^{\vee}\rangle, \tag{A.49d}$$

$$\widehat{P}_{\rm F}^{\vee}\,|\text{Bonding}_{\rm e}^{\vee}\rangle = |\text{Bonding}_{\rm e}^{\vee}\rangle. \tag{A.49e}$$

We then infer that

$$
\begin{aligned}
\widehat{U}_{\mathrm{e}}^{\vee}\,|\mathrm{Bonding}_{\mathrm{o}}^{\vee}\rangle &= \left(\widehat{U}_{\mathrm{e}}^{\vee}\,\widehat{U}_{\mathrm{o}}^{\vee}\right)\left(\widehat{U}_{\mathrm{o}}^{\vee}\,|\mathrm{Bonding}_{\mathrm{o}}^{\vee}\rangle\right)\\
&= \left(\widehat{P}_{\mathrm{F}}^{\vee}\right)|\mathrm{Bonding}_{\mathrm{o}}^{\vee}\rangle\\
&= |\mathrm{Bonding}_{\mathrm{o}}^{\vee}\rangle,
\end{aligned}
\tag{A.50}
$$

and

$$
\begin{aligned}
\widehat{U}_{\mathrm{e}}^{\vee}\,|\mathrm{Bonding}_{\mathrm{e}}^{\vee}\rangle &= \left(\widehat{U}_{\mathrm{e}}^{\vee}\,\widehat{U}_{\mathrm{o}}^{\vee}\right)\left(\widehat{U}_{\mathrm{o}}^{\vee}\,|\mathrm{Bonding}_{\mathrm{e}}^{\vee}\rangle\right)\\
&= \left(\widehat{P}_{\mathrm{F}}^{\vee}\right)|\mathrm{Bonding}_{\mathrm{e}}^{\vee}\rangle\\
&= |\mathrm{Bonding}_{\mathrm{e}}^{\vee}\rangle.
\end{aligned}
\tag{A.51}
$$

$\square$

# B  Triality with open boundary conditions

To treat the case of open boundary conditions, we need to modify the bond algebras

$$
\mathfrak{B}_b\,,\qquad \mathfrak{B}_{b'}\,,\qquad \mathfrak{B}_f\,,
\tag{B.1}
$$

defined in Secs. 2.1, 2.2, and 2.3, respectively. The following changes must be done as one repeats all steps of Sec. 2.

One must remove the term $\hat{\sigma}_{2N}^{x}\,\hat{\sigma}_{2N+1}^{x}$ from $\mathfrak{B}_b$ as the dual lattice

$$
\Lambda^{\star} = \left\{j^{\star}=j+\frac{1}{2}\ \middle|\ j=1,\cdots,2N-1\right\},
\tag{B.2a}
$$

has one less site than the direct lattice

$$
\Lambda = \left\{j=1,\cdots,2N\right\},
\tag{B.2b}
$$

when open boundary conditions are imposed.

One must modify the bond algebras (B.1) according to

$$
\mathfrak{B}_b \to \mathfrak{B}_{\sigma} := \left\langle \hat{\sigma}_i^z,\qquad \hat{\sigma}_j^x\,\hat{\sigma}_{j+1}^x\ \middle|\ i\in\Lambda,\qquad j\in\Lambda\setminus\{2N\}\right\rangle,
\tag{B.3a}
$$

$$
\mathfrak{B}_{b,b'} \to \mathfrak{B}_{\sigma,\tau} := \left\langle \hat{\sigma}_i^z,\qquad \hat{\sigma}_j^x\,\hat{\tau}_{j^{\star}}^z\,\hat{\sigma}_{j+1}^x\ \middle|\ i\in\Lambda,\qquad j\in\Lambda\setminus\{2N\}\right\rangle,
\tag{B.3b}
$$

$$
\mathfrak{B}_{b,f} \to \mathfrak{B}_{\sigma,\beta\alpha} := \left\langle \hat{\sigma}_i^z,\qquad \hat{\sigma}_j^x\left(\mathrm{i}\hat{\beta}_j\,\hat{\alpha}_{j+1}\right)\hat{\sigma}_{j+1}^x\ \middle|\ i\in\Lambda,\qquad j\in\Lambda\setminus\{2N\}\right\rangle,
\tag{B.3c}
$$

with the Hilbert space

$$
\begin{aligned}
\mathcal{H}_{b,b'} := \mathcal{H}_b\otimes\mathcal{H}_{b'} &\to \mathcal{H}_{\sigma,\tau} := \mathcal{H}_\sigma\otimes\mathcal{H}_\tau\\
&\cong \left(\bigotimes_{j\in\Lambda}\mathbb{C}^2\right)\otimes\left(\bigotimes_{j^{\star}\in\Lambda^{\star}}\mathbb{C}^2\right) = \mathbb{C}^{2^{2N}}\otimes\mathbb{C}^{2^{2N-1}} = \mathbb{C}^{2^{4N-1}},
\end{aligned}
\tag{B.3d}
$$

of dimension $2^{4N-1}$ and that

$$
\begin{aligned}
\mathcal{H}_{b,f} := \mathcal{H}_b\otimes\mathcal{H}_f &\to \mathcal{H}_{\sigma,\beta\alpha} := \mathcal{H}_\sigma\otimes\mathcal{H}_{\beta\alpha}\\
&\cong \left(\bigotimes_{j\in\Lambda}\mathbb{C}^2\right)\otimes\left(\bigotimes_{j\in\Lambda}\mathbb{C}^2\right) = \mathbb{C}^{2^{2N}}\otimes\mathbb{C}^{2^{2N}} = \mathbb{C}^{2^{4N}},
\end{aligned}
\tag{B.3e}
$$

of dimension $2^{4N}$ as domain of definition, respectively.

One must modify the local Gauss operators (2.9) and (2.24) according to

$$\widehat{G}_{b,b';j} \to \widehat{G}_{\sigma,\tau;j} := \begin{cases} \hat{\sigma}_1^z \, \hat{\tau}_{1+\frac{1}{2}}^x \,, & j = 1 \,, \\ \hat{\tau}_{j-1+\frac{1}{2}}^z \, \hat{\sigma}_j^z \, \hat{\tau}_{j+\frac{1}{2}}^z \,, & j = 2, \cdots, 2N-1 \,, \\ \hat{\tau}_{2N-1+\frac{1}{2}}^z \, \hat{\sigma}_{2N}^z \,, & j = 2N \,, \end{cases}$$ (B.4a)

and

$$\widehat{G}_{b,f;j} \to \widehat{G}_{\sigma,\beta\alpha;j} := \mathrm{i}\hat{\beta}_j \, \hat{\sigma}_j^z \, \hat{\alpha}_j \,, \qquad j \in \Lambda \,,$$ (B.4b)

respectively.

One must modify the dualized bond algebras (2.16) and (2.31) according to

$$\mathfrak{B}_{b'} \to \mathfrak{B}_\tau := \left\langle \left( \hat{\tau}_{i^\star-1}^{x\,\vee} \right)^{1-\delta_{i,1}} \left( \hat{\tau}_{i^\star}^{x\,\vee} \right)^{1-\delta_{i,2N}} \,, \qquad \hat{\tau}_{j^\star}^{z\,\vee} \;\middle|\; i \in \Lambda \,, \qquad j \in \Lambda \setminus \{2N\} \right\rangle \,,$$ (B.5)

and

$$\mathfrak{B}_f \to \mathfrak{B}_{\beta\alpha} := \left\langle \mathrm{i}\hat{\beta}_j^\vee \, \hat{\alpha}_j^\vee \,, \qquad \mathrm{i}\hat{\beta}_j^\vee \, \hat{\alpha}_{j+1}^\vee \;\middle|\; i \in \Lambda \,, \quad j \in \Lambda \setminus \{2N\} \right\rangle \,,$$ (B.6)

respectively.

One must modify the consistency conditions (2.20a) and (2.33c) by identifying the dual pairs

$$\left( \prod_{j \in \Lambda} \hat{\sigma}_j^z = \widehat{U}_{r_\pi^z} \,, \qquad \widehat{\mathbb{1}}_{\mathcal{H}_\tau^\vee} \right) \,,$$ (B.7a)

and

$$\left( \prod_{j \in \Lambda} \hat{\sigma}_j^z = \widehat{U}_{r_\pi^z} \,, \qquad \prod_{j \in \Lambda} \left( \mathrm{i}\hat{\beta}_j \, \hat{\alpha}_j \right) = -\widehat{P}_{\mathrm{F}}^\vee \right) \,,$$ (B.7b)

respectively. Here, $\mathcal{H}_\tau^\vee$ ($\mathcal{H}_{\beta\alpha}^\vee$) is the projection of $\mathcal{H}_{\sigma,\tau}$ ($\mathcal{H}_{\sigma,\beta\alpha}$) to the subspace on which all local Gauss operators reduce to the identity. Correspondingly, the dual pairs of Hilbert subspaces are

$$\left( \mathcal{H}_\sigma^{\mathrm{dual}}, \qquad \mathcal{H}_\tau^{\vee\,\mathrm{dual}} \right) \,, \qquad \mathcal{H}_\tau^{\vee\,\mathrm{dual}} := \mathcal{H}_\tau^\vee \,,$$ (B.8a)

where $\mathcal{H}_\sigma^{\mathrm{dual}}$ is the $2^{2N}-1$-dimensional subspace of $\mathcal{H}_\sigma$ on which $\widehat{U}_{r_\pi^z}$ reduces to the identity, and

$$\left( \mathcal{H}_\sigma^{\mathrm{dual}}, \qquad \mathcal{H}_{\beta\alpha}^{\vee\,\mathrm{dual}} \right) \,, \qquad \mathcal{H}_\sigma^{\mathrm{dual}} := \mathcal{H}_\sigma \,, \qquad \mathcal{H}_{\beta\alpha}^{\vee\,\mathrm{dual}} := \mathcal{H}_{\beta\alpha}^\vee \,,$$ (B.8b)

respectively.

The Kramers-Wannier dual of the Hamiltonian (4.1) when open boundary conditions are imposed in the reduced coupling space (4.3) is[23]

$$\begin{aligned} \widehat{H}_\tau^\vee = {} & \sum_{j=1}^{2N-1} \hat{\tau}_{j+\frac{1}{2}}^{z\,\vee} - \Delta \left( \underline{\hat{\tau}_{1+\frac{1}{2}}^{z\,\vee} \, \hat{\tau}_{2+\frac{1}{2}}^{x\,\vee}} + \sum_{j=2}^{2N-2} \hat{\tau}_{j-1+\frac{1}{2}}^{x\,\vee} \, \hat{\tau}_{j+\frac{1}{2}}^{z\,\vee} \, \hat{\tau}_{j+1+\frac{1}{2}}^{x\,\vee} + \underline{\hat{\tau}_{2N-2+\frac{1}{2}}^{x\,\vee} \, \hat{\tau}_{2N-1+\frac{1}{2}}^{z\,\vee}} \right) \\ & + J \left\{ \sum_{j=1}^{2N-2} \hat{\tau}_{j+\frac{1}{2}}^{z\,\vee} \, \hat{\tau}_{j+1+\frac{1}{2}}^{z\,\vee} + \Delta \left[ \underline{\left( \hat{\tau}_{1+\frac{1}{2}}^{z\,\vee} \, \hat{\tau}_{2+\frac{1}{2}}^{x\,\vee} \right) \left( \hat{\tau}_{1+\frac{1}{2}}^{x\,\vee} \, \hat{\tau}_{2+\frac{1}{2}}^{z\,\vee} \, \hat{\tau}_{3+\frac{1}{2}}^{x\,\vee} \right)} \right. \right. \\ & \left. \left. + \sum_{j=2}^{2N-3} \left( \hat{\tau}_{j-\frac{1}{2}}^{x\,\vee} \, \hat{\tau}_{j+\frac{1}{2}}^{z\,\vee} \, \hat{\tau}_{j+\frac{3}{2}}^{x\,\vee} \right) \left( \hat{\tau}_{j+\frac{1}{2}}^{x\,\vee} \, \hat{\tau}_{j+\frac{3}{2}}^{z\,\vee} \, \hat{\tau}_{j+\frac{5}{2}}^{x\,\vee} \right) \right. \right. \\ & \left. \left. + \underline{\left( \hat{\tau}_{2N-3+\frac{1}{2}}^{x\,\vee} \, \hat{\tau}_{2N-2+\frac{1}{2}}^{z\,\vee} \, \hat{\tau}_{2N-1+\frac{1}{2}}^{x\,\vee} \right) \left( \hat{\tau}_{2N-2+\frac{1}{2}}^{x\,\vee} \, \hat{\tau}_{2N-1+\frac{1}{2}}^{z\,\vee} \right)} \right] \right\} \,. \end{aligned}$$ (B.9)

---

[23]We emphasize that the dual of the Hamiltonian (4.1) when open boundary conditions are imposed is not the Hamiltonian (4.9) when open boundary conditions are imposed.

The terms that are underlined once only act non trivially on the left boundary. The terms that are underlined twice only act non trivially on the right boundary. These boundary terms break explicitly the global internal symmetry (3.12). These boundary terms gap the zero modes, if present, of the bulk contributions (all terms that are not underlined) to the Hamiltonian.

The Jordan-Wigner dual of the Hamiltonian (4.1) when open boundary conditions are imposed in the reduced coupling space (4.3) is

$$\widehat{H}_{\beta\alpha}^{\vee} = \sum_{j=1}^{2N-1} \left( \mathrm{i}\hat{\beta}_j^{\vee}\,\hat{\alpha}_{j+1}^{\vee} + \Delta\,\mathrm{i}\hat{\alpha}_j^{\vee}\,\hat{\beta}_{j+1}^{\vee} \right) + J \sum_{j=1}^{2N-2} \left( \mathrm{i}\hat{\beta}_j^{\vee}\,\hat{\beta}_{j+1}^{\vee}\,\hat{\alpha}_{j+1}^{\vee}\,\hat{\alpha}_{j+2}^{\vee} + \Delta\,\mathrm{i}\hat{\alpha}_j^{\vee}\,\hat{\alpha}_{j+1}^{\vee}\,\hat{\beta}_{j+1}^{\vee}\,\hat{\beta}_{j+2}^{\vee} \right). \tag{B.10}$$

If we do the unitary transformation

$$\hat{\beta}_j^{\vee} \mapsto +\hat{\alpha}_j^{\vee}\,, \qquad \hat{\alpha}_j^{\vee} \mapsto -\hat{\beta}_j^{\vee}\,, \tag{B.11}$$

we recover Hamiltonian (4.19) in the reduced coupling space (4.3) with open boundary conditions. The two-fold degeneracy of the Neel$_x$ or Neel$_y$ phases is now interpreted by the existence of a single Majorana zero mode localized at the left and right ends of the open chain.

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
