# Peer review of "Lieb-Schultz-Mattis anomalies and web of dualities induced by gauging in quantum spin chains"

_SciPost Physics, doi:SciPost Phys. 16, 022 (2024)_

## Round 1 · Referee Report · Anonymous (Referee 1) · 2023-8-22

Strengths

Detailed analysis of several lattice models and the relations between them using duality or gauging.

Weaknesses

No clear comparison of the main results with the existing statements in the literature.

Report

I suggest to add a comparison of the main results with the existing statements in the literature. This can be added in the Conclusions.

After such an addition, the paper is suitable for publication.

Requested changes

See above.

---

## Round 1 · Referee Report · Anonymous (Referee 2) · 2023-10-22

Report

I have read this paper and I have a few comments. In my opinion this is an important paper which, in some form, deserved to be published. I say "in some form" because the paper is really hard to read. It is much (and unnecessarily) too long. There are too many formal results proven (or argued) one after another without a sense of importance or significance. I am not sure if the authors need all these results. Perhaps a fraction of them (the less important ones) can be presented in an appendix. The applications are very interesting but they get lost after the myriad of (claimed) rigorous results presented in the very long first part of the paper. I would recommend either to split the paper in two, or to present in the body of the paper only the important results. If the authors cannot decide which are the more important results they cannot possibly expect the readers to do it for them. The result is that their important results will be lost to the readership.

Thus, my recommendation is to publish after the authors submit a more readable manuscript.

Requested changes

see above

---

## Round 1 · Referee Report · Anonymous (Referee 3) · 2023-11-13

Strengths

  1. Section 2 was a clear introduction into the triality concept, using the well-known example of $\mathbb{Z}_2$ XYZ Heisenberg chain. I found it illuminating/a good refresher.

  2. The subject of LSM and crystalline symmetry anomalies is of high value to the quantum matter community. Since the concept of gauging crystalline symmetries is complicated, it is nice to see the authors use a more well-known approach (i.e. gauging internal symmetries) to identify LSM type systems.

  3. Results seem correct and are new, although potentially straightforward generalization of previous work. It would help if the authors could more clearly highlight their new contributions versus already known results.

Weaknesses

  1. Although this approach is demonstrative of some of the interesting properties of LSM, I feel it is not generalizable or illuminating to the more general non-onsite significance of translation/crystalline symmetries (for example, LSM-like theories when there is only translation symmetry). However, I understand that this is beyond the scope of this paper - perhaps the authors could describe the limitations of their methods in the conclusion/introduction.

  2. I feel that the presentation could more clearly emphasize the new results versus what is already known in literature. Section 2 (although very nice) is very long, so it distract the readers from the actual flesh of the paper. Perhaps a clearer summary of the new results in the introduction, for example in the form of bullet points (since the introduction is a large block of text).

Report

The manuscript "Lieb-Schultz-Mattis anomalies and web of dualities induced by gauging in quantum spin chains" illuminates the procedure of gauging non-anomalous symmetry subgroups to detect existence of an LSM obstruction. The authors present their work clearly with the general subject being of great interest.

My main reservation is the limitation/novelty of this method to study general LSM type systems and illuminate the significance of non-onsite symmetries such as translation. However, with the appropriate changes, this manuscript certainly deserves publication in SciPost. I thank the author for their thought-provoking and interesting work!

---

## Round 2 · Referee Report · Anonymous (Referee 2) · 2023-12-9

Report

I have read and reviewed the resubmitted version of this paper. I appreciate that the authors decided to include two new subsections in the Introduction, one summarizing the salient results and the other on the relation between this work and other published paper. I am disappointed that the authors did not appreciate my other recommendations as being difficult to implement. It is a pity as the paper is indeed much too long (as are the published papers cite as examples) and this will most likely result in a paper that is not as impactful as it could be. But this is their choice. I recommend that this paper be published.

---

## Round 2 · Referee Report · Anonymous (Referee 1) · 2023-12-13

Report

I find the paper interesting and suitable for publications. The revisions made it clearer.

One could argue that the paper is too long. But as a matter of principle, it is better for a paper to be too long than to be too short and therefore incomprehensible. Personally, I do not think this paper is too long.

---

## Round 2 · Author Response

We would like to thank all three referees for their reports.

Referee 1:

We thank Referee 1 for delivering very quickly their report. We followed their suggestion to ``... add a comparison of the main results with the existing statements in the literature.'' which we placed in Subsec. 1.3 of the introduction.

Referee 2:

We thank Referee 2 for assessing the validity, significance, originality, and grammar of our paper as high, good, high, and perfect, respectively.

We agree with Referee 2 that it is better to ``... present in the body of the paper only the important results.'' This is done with the addition of Subsecs. 1.2 and 1.3. We hope that these changes make our results more transparent and facilitate an easier read.

Our paper is comparable in length and is written in the style of Refs.

[39] M. Cheng and N. Seiberg, ``Lieb-Schultz-Mattis, Luttinger, and ’t Hooft - anomaly matching in lattice systems,'' SciPost Phys. 15, 051 (2023), doi:10.21468/SciPostPhys.15.2.051.

[41] N. Seiberg and S.-H. Shao, ``Majorana chain and Ising model –(non-invertible) trans- lations, anomalies, and emanant symmetries,'' arXiv e-prints arXiv:2307.02534 (2023), doi:10.48550/arXiv.2307.02534, 2307.02534 .

[53] S. Seifnashri, ``Lieb-Schultz-Mattis anomalies as obstructions to gauging (non-on-site) symmetries,'' arXiv e-prints arXiv:2308.05151 (2023), doi:10.48550/arXiv.2308.05151, 2308.05151.

The first is already published, while the other two are submitted to SciPost Physics.

We appreciate the Referee's suggestion to reorganize the entire paper. However, we have decided to maintain the current structure,
as we believe it effectively communicates our ideas. Additionally, both Referees 1 and 3 share our perspective, emphasizing that the earlier sections are valuable in elucidating key concepts crucial for comprehending the results of our work.

Referee 3:

We thank Referee 3 for their report and praise of our Sec. 2. Sections 1.2 and 1.3 address the request of Referee 3 to ``... more clearly highlight their new contributions versus already known results.''

Regarding weakness 1:
Most LSM theorems that we know of mix crystalline and internal symmetries. We do not understand what is the meaning of gauging a crystalline symmetry on its own right and do not claim any breakthrough in this direction.

Regarding weakness 2:
We followed with Sec. 1.2 the suggestion to present ``..., a clearer summary of the new results in the introduction, for example in the form of bullet points (since the introduction is a large block of text).''

---

## Round 2 · List of Changes

1. The main changes are the partitioning of the introduction (Sec. 1) into four subsections. Subsections 1.1 and 1.4 are not new. Subsections 1.2 and 1.3 are new. In Subsec. 1.2, we present in a compact and itemized format our new results. In Subsec. 1.3, we explain the difference between our results and the literature.

  2. In the bulk of the paper, we added new results with Eqs. (3.16), (3.17), (3.28), (3.29), and below Eq. (5.23). This is a reinterpretation of the absence of the LSM anomaly in the KW and JW dual theories as a manifestation of the presence of simultaneous charge, dipole, and crystalline symmetries. Correspondingly, we added a few lines to the abstract.

  3. We made a correction around Eq. (4.14). Other changes are stylistic or typos.

---

## Editorial Decision

published